# Glutamate acts on acid-sensing ion channels to worsen ischaemic brain injury

Ke Lai[1,2,3,4,10], Iva Pritišanac[5,6,7], Zhen-Qi Liu[1,10], Han-Wei Liu[1], Li-Na Gong[1], Ming-Xian Li[1], Jian-Fei Lu[8], Xin Qi[8], Tian-Le Xu[8], Julie Forman-Kay[5,9], Hai-Bo Shi[1✉], Lu-Yang Wang[2,3✉] & Shan-Kai Yin[1✉]

Glutamate is traditionally viewed as the first messenger to activate NMDAR (*N*-methyl-D-aspartate receptor)-dependent cell death pathways in stroke[1,2], but unsuccessful clinical trials with NMDAR antagonists implicate the engagement of other mechanisms[3–7]. Here we show that glutamate and its structural analogues, including NMDAR antagonist L-AP5 (also known as APV), robustly potentiate currents mediated by acid-sensing ion channels (ASICs) associated with acidosis-induced neurotoxicity in stroke[4]. Glutamate increases the affinity of ASICs for protons and their open probability, aggravating ischaemic neurotoxicity in both in vitro and in vivo models. Site-directed mutagenesis, structure-based modelling and functional assays reveal a bona fide glutamate-binding cavity in the extracellular domain of ASIC1a. Computational drug screening identified a small molecule, LK-2, that binds to this cavity and abolishes glutamate-dependent potentiation of ASIC currents but spares NMDARs. LK-2 reduces the infarct volume and improves sensorimotor recovery in a mouse model of ischaemic stroke, reminiscent of that seen in mice with *Asic1a* knockout or knockout of other cation channels[4–7]. We conclude that glutamate functions as a positive allosteric modulator for ASICs to exacerbate neurotoxicity, and preferential targeting of the glutamate-binding site on ASICs over that on NMDARs may be strategized for developing stroke therapeutics lacking the psychotic side effects of NMDAR antagonists.

Ischaemic stroke remains the leading cause of death and disability in the world[8]. During stroke, ischaemia deprives brain infarct area of glucose and oxygen, elevating glutamate release and accumulation by 10–100-fold above physiological concentrations[1,9]. This results in an overactivation of NMDARs, intracellular calcium overload and cell death. NMDAR-dependent excitotoxicity rationalizes NMDARs as a key target for stroke treatments[2]. NMDAR antagonists in in vitro and in vivo models are highly neuroprotective[10,11], but have failed at different stages of clinical trials[3], raising the possibility that NMDARs are not solely accountable for glutamate-induced excitotoxicity in stroke. Acid-sensing ion channels (ASICs) are regarded as a strong candidate for mediating neurotoxicity as local acidosis occurs in ischaemic areas during stroke[4,12–14]. NMDARs and ASICs may be coincidently activated by excessive glutamate and H[+], respectively, and these channels may cross-talk through intracellular signalling to aggravate neuronal death[5]. Glutamate was found to enhance recombinant ASIC1a currents[15]. However, it is unclear whether glutamate directly or indirectly interacts with native ASICs in neurons, and whether this interaction mediates and/or aggravates ischaemic injury.

## Glutamate potentiates $I_{ASICs}$

To investigate whether the function of ASICs is regulated by glutamate (that is, L-glutamic acid, unless otherwise stated hereafter), we first performed patch-clamp recordings of acid-evoked currents ($I_{ASICs}$) from CHO cells expressing human ASIC1a (hASIC1a) channels. Glutamate potentiated $I_{ASICs}$ evoked by extracellular fluid (ECF) with pH values ranging from 6.55 to 7.4 (that is, 0.04 to 0.28 µmol l⁻¹ proton concentrations) and left-shifted proton dose–response curve, with the half-maximum effective concentration (EC$_{50}$) being decreased from 189 nM to 152 nM, while the maximal currents were identical at pH values of less than 6.5 (Fig. 1a and Extended Data Fig. 1a–d). When the test pulses of pH 6.6 solution were preceded by perfusate with increasing acidity (from 7.6 to 6.8), the steady-state desensitization of $I_{ASICs}$ was reduced by co-application of glutamate, producing a rightward shift of EC$_{50}$ from 51 to 61 nM (Fig. 1b), suggesting that glutamate reduces closed-state desensitization of ASIC1a channels. The potentiation of $I_{ASICs}$ by glutamate was highly dependent on the acidity of test solutions as exemplified by EC$_{50}$ values, that is, 463.6 µM at pH 7.0 versus 382.3 µM at pH 6.8 (Fig. 1c). To test whether a direct interaction exists

[1]Department of Otorhinolaryngology, Shanghai Sixth People's Hospital and Shanghai Jiao Tong University School of Medicine, Shanghai, China. [2]Program in Neuroscience and Mental Health, SickKids Research Institute, Toronto, Ontario, Canada. [3]Department of Physiology, University of Toronto, Toronto, Ontario, Canada. [4]Shanghai Center for Brain Science and Brain-Inspired Technology, Shanghai, China. [5]Program in Molecular Medicine, SickKids Research Institute, Toronto, Ontario, Canada. [6]Department of Cell & Systems Biology, University of Toronto, Toronto, Ontario, Canada. [7]Department of Medicinal Chemistry, Otto Loewi Research Center, Medical University of Graz, Graz, Austria. [8]Department of Anatomy and Physiology, Shanghai Jiao Tong University School of Medicine, Shanghai, China. [9]Department of Biochemistry, University of Toronto, Toronto, Ontario, Canada. [10]These authors contributed equally: Ke Lai, Zhen-Qi Liu. ✉e-mail: hbshi@sjtu.edu.cn; luyang.wang@utoronto.ca; skyin@sjtu.edu.cn

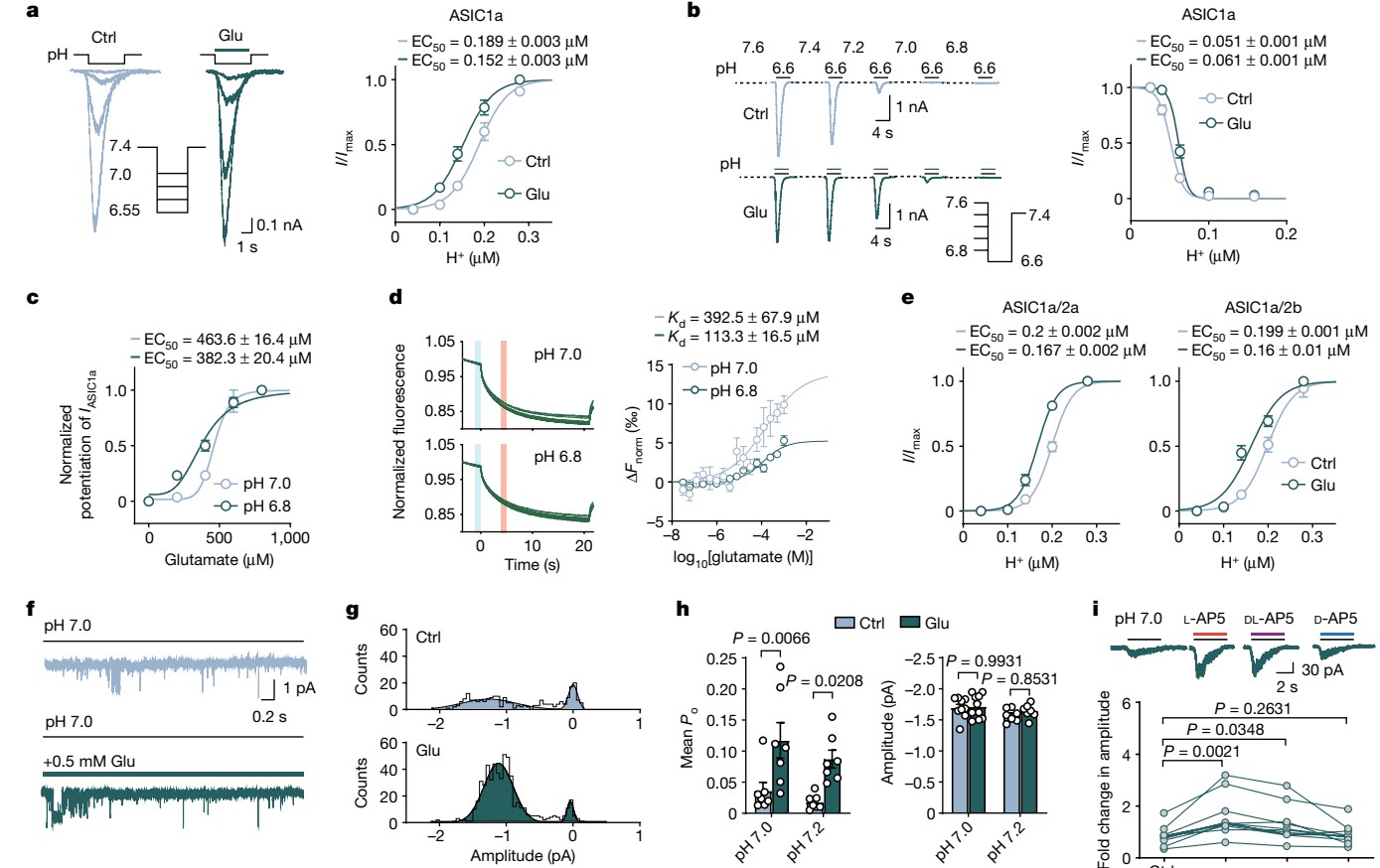

**Fig. 1 | Glutamate and its structural analogues robustly potentiate ASIC currents. a**, Examples of $I_{ASICs}$ evoked in an hASIC1a-transfected CHO cell in the absence and presence of 500 μM glutamate for pooled dose–response curves. $n = 14$. **b**, An example recording showing steady-state desensitization of ASIC1a currents with and without coapplied glutamate for pooled dose–response curves ($n = 11$) for each group. **c**, Dose–response curves for glutamate to potentiate $I_{ASICs}$ at pH 7.0 ($n = 14$) and 6.8 ($n = 8$). **d**, MST imaging traces and dose–response curves showing direct binding between glutamate and GFP-tagged ASIC1a at pH 7.0 ($n = 5$) and 6.8 ($n = 6$). The vertical bars show the cold fluorescence detected at −1–0 s (blue); hot fluorescence detected at 4–5 s (red).

**e**, Dose–response curves for ASIC1a/2a and ASIC1a/2b currents. $n = 13$ and 14 cells, respectively. **f**, Outside-out patch recordings of ASIC single-channel currents evoked by pH 7.0. **g**, All-points amplitude histogram of ASIC single-channel currents from **f**; curves were fitted by double Gaussian components. Bin = 0.05 pA. **h**, The $P_o$ and amplitude of ASIC1a unitary currents evoked at pH 7.0 and 7.2. $n = 7–9$ patches. **i**, The effects of L-, DL- and D-isomers of AP5 (400 μM) on ASIC1a at pH 7.0. $n = 11$ cells. Three to six replicated cultures for patch recordings were tested. Data are mean ± s.e.m. Statistical analysis was performed using two-way analysis of variance (ANOVA) (**h**) and one-way ANOVA (**i**) with Tukey post hoc correction for multiple comparisons.

between glutamate and ASIC1a at different pH values, we used the microscale thermophoresis (MST) in vitro fluorescence binding assay, which detects ligand–receptor binding directly in cell lysates[16,17]. This assay confirmed that glutamate binds to green fluorescent protein (GFP)-tagged hASIC1a in lysates of transfected HEK293T cells with a lower dissociation constant ($K_d$) value at pH 6.8 (113.3 μM) compared with at pH 7.0 (392.5 μM), but does not bind to GFP alone (Fig. 1d and Extended Data Fig. 1e). These results implicated glutamate as a positive allosteric modulator (PAM) for ASIC1a by direct binding. As ASICs can be homomeric ASIC1a channels or heteromeric channels with the ASIC2a or ASIC2b subunit in native central neurons[18], we tested the effect of glutamate on these heteromeric channels. The potentiation by glutamate was preserved (Fig. 1e), indicating that ASIC1a is an integral component of ASICs for glutamate to exert its action. Finally, we performed a series of experiments that ruled out the possibility that the potentiation of $I_{ASICs}$ by glutamate is due to unmasking of the inhibition of $I_{ASICs}$ by $Ca^{2+}$ or $Zn^{2+}$ that is known to block ASIC1a[19,20] (Extended Data Fig. 1f–n).

We next performed single-channel recordings in outside-out patches from ASIC1a-transfected CHO cells and found that glutamate increased the open probability ($P_o$) of ASIC1a at pH 7.0 and even a milder pH 7.2, without affecting the amplitude of ASIC1a unitary currents (Fig. 1f–h).

The current–voltage relationship with and without glutamate showed the same slope values (conductance: 17.3 ± 0.9 pS (control) and 18.4 ± 0.6 pS (glutamate)) and reversal potential (33.15 mV (control) and 28.56 mV (glutamate)) (Extended Data Fig. 1o,p), indicating that the potentiation of $I_{ASICs}$ by glutamate is solely attributed to $P_o$ increase without affecting the single-channel conductance and ion selectivity of ASICs.

Having established the actions of glutamate on $I_{ASICs}$, we tested widely used agonists for glutamate receptors, namely NMDA, AMPA (α-amino-3-hydroxy-5-methyl-4-isoxazole propionic acid), aspartic acid (Asp) and kainic acid, all of which showed similar effects, except for kainic acid (Extended Data Fig. 1q,r). We replicated these results in primary cultured cortical neurons and confirmed that glutamate potentiated $I_{ASICs}$ at pH 7.0 from wild-type mice ($Asic1a^{+/+}$) but not from $Asic1a$-knockout mice ($Asic1a^{−/−}$), indicating that ASIC1a uniquely contains the site for glutamate interaction (Extended Data Fig. 1s). These results prompted us to investigate whether this potentiation could be affected by the classical NMDAR competitive blocker 2-amino-5-phosphonovaleric acid (AP5). Notably, we found no block of $I_{ASICs}$ by three AP5 isomers (L, racemic mixture DL and D; 200 μM) (Extended Data Fig. 2a,b). Instead, the L- and DL-, but not D-isomer of AP5, enhanced $I_{ASICs}$ (Fig. 1i). DL-AP5 increased the $P_o$ of ASIC1a at both pH 7.0 and 7.2 (Extended

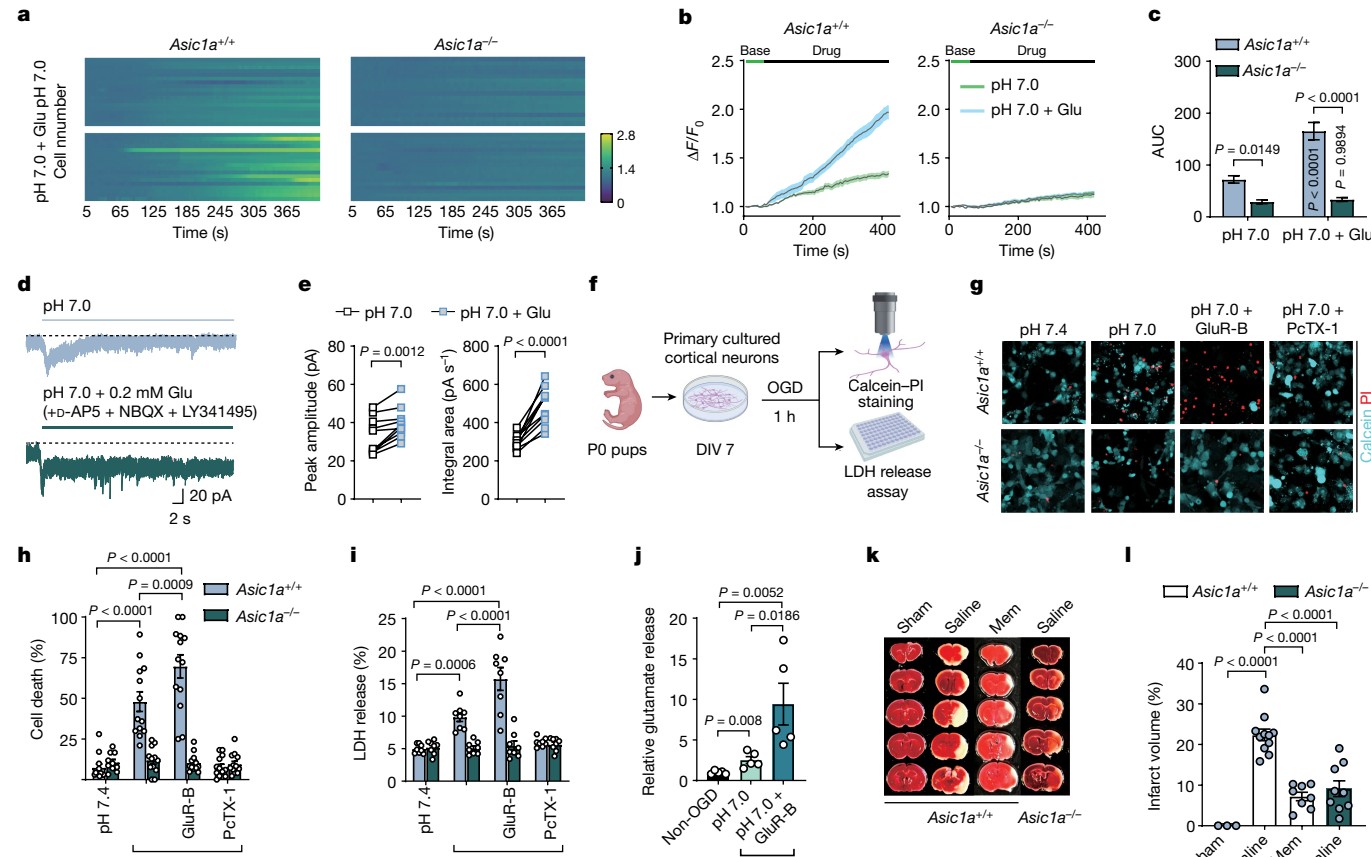

**Fig. 2 | Glutamate aggravates neurotoxicity in vitro and in vivo. a–c**, Examples and summary of changes in $[Ca^{2+}]_i$ levels and time-course imaging in primary cultured cortical neurons from $Asic1a^{+/+}$ ($n = 18$ and 18 cells from 3 cultures per group) and $Asic1a^{-/-}$ ($n = 16$ and 18 cells from 3 cultures per group) mice with and without addition of 200 μM glutamate to the pH 7.0 solution. AUC, area under the curve. **d**, $I_{ASICs}$ evoked at pH 7.0 from cultured cortical neurons for 30 s with and without glutamate. **e**, The peak amplitude and integral area of $I_{ASICs}$ from **d** were pooled from $n = 10$ cells from 3 replicated cultures. 50 μM D-AP5, 10 μM NBQX and 10 μM LY341495 were applied. **f**, Schematic of the cell death experiment. DIV, days in vitro; P0, postnatal day 0. The diagram was created using BioRender (https://biorender.com). **g**, Calcein–propidium iodide (PI) staining of cultured neurons from $Asic1a^{+/+}$ and $Asic1a^{-/-}$ mice under different conditions: pH 7.4 or 7.0 with or without 100 ng ml⁻¹ PcTX-1 or GluR-B; glutamate receptor blockers consisted of 50 μM D-AP5, 10 μM NBQX and 10 μM LY341495. **h**, Summary of the percentage of cell death calculated by calcein (live cells) and propidium iodide (dead cells) counting. $n = 10–13$ images from 3 replicated cultures for each group. **i**, LDH release from cultured neurons. $n = 8$ cultures for each group. **j**, The relative glutamate release with or without 1 h OGD. $n = 5$ cultures per group. **k**, Histology images of brain slices from mice subjected to MCAO. **l**, Quantification of the infarct volume after MCAO mice were injected with physiological saline and memantine (mem; 1 mg per kg). $n = 3, 11, 8$ and 9 mice for each group. Data are mean ± s.e.m. Statistical analysis was performed using two-tailed paired $t$-tests (**e**), and two-way (**c**,**h**,**i**) and one-way ANOVA (**j**,**l**) with Tukey post hoc correction.

Data Fig. 2c–f). Furthermore, an MST assay detected strong binding between L-AP5 and ASIC1a (Extended Data Fig. 2g,h). In contrast to L-glutamate and its sodium or potassium salts, D-glutamate did not potentiate $I_{ASICs}$ (Extended Data Fig. 3). These results showed that the influence of AP5 and glutamate on ASIC1a depends on their specific stereochemistry, with effectiveness observed only in the L-isomer, but not the D-isomer, consistent with their optical activity and rotation properties (Supplementary Table 1). These observations suggested that glutamate and its structural analogues must share a common binding site, most likely on the extracellular domain of ASIC1a.

## Glutamate enhances cell death through ASIC1a

Excessive glutamate release and acidosis are believed to cause ischaemic neuronal death by activating NMDARs and ASICs, respectively[4,21]. Our observations led us to hypothesize that glutamate may cause NMDAR-independent cell death by directly acting on ASICs to drive $Ca^{2+}$ overload. To this end, we performed calcium imaging in cultured cortical neurons from $Asic1a^{+/+}$ and $Asic1a^{-/-}$ mice in the presence of the AMPA receptor blocker NBQX, NMDAR pore blocker MK801 and voltage-gated calcium channel blocker $CdCl_2$ (Extended Data Fig. 4a).

In $Asic1a^{+/+}$ cortical neurons, we found that application of pH 7.0 solution alone led to slow elevation of intracellular $Ca^{2+}$ ($[Ca^{2+}]_i$), which was robustly potentiated by co-application of glutamate, consistent with the idea that glutamate can promote $Ca^{2+}$ overload through ASIC1a, independent of other routes of $Ca^{2+}$ influx. This was reinforced by the same experiments with $Asic1a^{-/-}$ neurons, in which glutamate could elevate $[Ca^{2+}]_i$ only slightly at pH 7.0 (Fig. 2a–c). To rule in/out other sources of elevated $[Ca^{2+}]_i$, we blocked intracellular $Ca^{2+}$ release via/from the metabotropic glutamate receptors (mGluRs, known to trigger intracellular release of $Ca^{2+}$), mitochondria or endoplasmic reticulum[22] with LY341495, CGP37157 and RO2959—inhibitors of the mitochondrial $Na^+/Ca^{2+}$ exchanger and $Ca^{2+}$-release-activated $Ca^{2+}$ (CRAC) channel in the endoplasmic reticulum, respectively. Under such conditions, glutamate remained capable of generating long-lasting $Ca^{2+}$ elevation in $Asic1a^{+/+}$ neurons, but not under conditions with zero-$Ca^{2+}$ ECF, or psalmotoxin-1 (PcTX-1; a blocker of ASICs) (Extended Data Fig. 4b,c). These results indicated that the persistent $Ca^{2+}$ elevation must originate from direct entry through ASICs. With a cocktail of glutamate receptor antagonists (GluR-B), $I_{ASICs}$ fully desensitized in response to pH 7.0 perfusate, but co-application of glutamate significantly attenuated its desensitization to sustain a steady-state inward current and enlarge

charge integrals (Fig. 2d,e), which probably accounts for the cumulative $Ca^{2+}$ overload. In parallel, co-application of glutamate caused a much large decrease in the mitochondrial membrane potential ($\Psi$m) compared with pH 7.0 perfusate alone in $Asic1a^{+/+}$ neurons, but not in $Asic1a^{-/-}$ neurons (Extended Data Fig. 4d–g). Such a decrease in $\Psi$m is regarded as a hallmark of early events en route to neuronal death[23,24].

To address whether glutamate causes cell death by acting on ASIC1a, we subjected primary cultured cortical neurons from $Asic1a^{+/+}$ and $Asic1a^{-/-}$ mice to the oxygen–glucose deprivation (OGD) paradigm for 1 h before performing calcein–propidium iodide staining of dead cells and assaying lactate dehydrogenase (LDH) release due to neuronal injury (Fig. 2f). We found a marked increase in cell death in $Asic1a^{+/+}$ neurons at pH 7.0 compared with $Asic1a^{-/-}$ neurons (Fig. 2g,h). Notably, OGD at lower pH values (that is, pH 6.0 or 6.5) did not increase cell death compared with that at pH 7.0 (Extended Data Fig. 4h–j). To examine the mechanisms underlying these counterintuitive observations, we compared the effects of increasing ECF acidity (pH 7.0, 6.5 and 6.0) on $I_{ASICs}$ with or without glutamate co-application to the same cells in the presence of GluR-B. We found that glutamate significantly attenuated $I_{ASIC}$ desensitization and generated a steady-state tonic current at pH 7.0, which was diminished at pH 6.0/6.5 despite large instantaneous currents being generated (Extended Data Fig. 4k,l), suggesting that the differences in tonic current underlie these unexpected outcomes of cell mortality. Compared with neurons treated at pH 7.4, slight acidification of ECF at pH 7.0 induced a significant increase in LDH release in $Asic1a^{+/+}$ neurons but not in $Asic1a^{-/-}$ neurons (Fig. 2i).

Notably, addition of GluR-B augmented cell death and LDH release in $Asic1a^{+/+}$ but not $Asic1a^{-/-}$ neurons at pH 7.0 (Fig. 2h,i). As none of three reagents in GluR-B potentiated $I_{ASICs}$ (Extended Data Fig. 4m), we postulated that OGD might increase the extracellular concentration of endogenous glutamate in cultured neurons to potentiate $I_{ASIC}$ tonic current after applying glutamate receptor blockers. Indeed, GluR-B led to around a fivefold increase in extracellular glutamate concentration after 1 h OGD (Fig. 2j). Blocking postsynaptic glutamate receptors might also cause a homeostatic upregulation of presynaptic glutamate release and postsynaptic excitability, which in turn amplifies ASIC1a-mediated tonic current to exacerbate cell death at pH 7.0 (Fig. 2h,i). By contrast, the reduced cell death at pH 6.0 (Extended Data Fig. 4h,i) may be attributed to full desensitization of ASICs and minimized tonic current, despite elevated glutamate during OGD. Blocking ASICs with PcTX-1 suppressed cell death and LDH release in $Asic1a^{+/+}$ neurons (Fig. 2h,i), indicating that the OGD paradigm favours acidosis- but not glutamate-receptor-mediated cell death. In ASIC1a-transfected CHO cells without native glutamate receptors, application of glutamate for 1 h at pH 7.0 yielded prominent cell death, which was abolished by PcTX-1 (Extended Data Fig. 4n,o). Thus, glutamate has a vital role in amplifying the level of $[Ca^{2+}]_i$, mitochondrial dysfunction and cell injury/death even under slightly acidic conditions (for example, pH 7.0), whereas blockade of ionotropic and metabotropic glutamate receptors does the opposite to the expected neuroprotection. Instead, glutamate might work through ASIC1a to cause $Ca^{2+}$ overload and neurotoxicity, with mild or moderate acidity being of the most lethal to neurons in the penumbra surround the ischaemic core.

To test the roles of ASICs versus NMDARs in ischaemic brain damage in vivo, we used a mouse stroke model by performing transient middle cerebral artery occlusion (MCAO) for 30 min to induce excessive glutamate release and acidosis during ischaemia and reperfusion. The infarct volume was quantified by post hoc tissue staining with 2,3,5-triphenyltetrazolium chloride (TTC) 24 h later (Extended Data Fig. 4p). MCAO reliably diminished the relative cerebral blood flow (rCBF) and cerebral infarction during surgery. We found significantly smaller infarct volumes in $Asic1a^{-/-}$ mice compared with those in $Asic1a^{+/+}$ mice, while the rCBF during MCAO between the two genotype mice showed no difference (Extended Data Fig. 4q,r and Fig. 2k,l). As a positive control, we administrated memantine[25] (NMDAR open channel

blocker, 1 mg per kg, intraperitoneally) and confirmed that it reduced the infarct volume in $Asic1a^{+/+}$ mice to a level comparable to that in $Asic1a^{-/-}$ mice (Fig. 2k,l and Extended Data Fig. 4s). These data indicate that targeting ASICs can be as effective as blocking NMDARs in reducing ischaemic brain damage in the MCAO mouse model.

## Glutamate directly binds to ASIC1a

To predict the binding pocket on ASIC1a for glutamate and its structural analogues to function as PAMs, we performed in-depth analyses of the properties and conservation of the accessible surface area in the ASIC1a structure as well as computational docking onto the trimeric extracellular domain of the available homomeric chicken ASIC1a (cASIC1a) structure (Protein Data Bank (PDB): 5WKU)[26]. As only a low-resolution cryo-electron structure of the human protein was available at the onset of this work[27], a higher-resolution structure of cASIC1a was used. The structure is expected to closely resemble that of hASIC1a, given a strong conservation between the human and chicken ASIC1a sequences (90% identity; Extended Data Fig. 5). We identified six putative glutamate-binding sites around amino acid residues Arg161, Lys379, Lys383, Gln226, Lys391 and Lys387 (Fig. 3a). Subsequent molecular docking (by HADDOCK; PDB: 5WKU) was performed to examine the plausibility of the proposed binding sites, compute the binding energetics and rank the sites (Supplementary Table 2). These putative sites were mapped back onto the hASIC1a sequence (that is, Arg160, Lys380, Lys384, Gln225, Lys392 and Lys388).

Site-directed mutagenesis was then performed to individually replace each of these six plausible interface residues with alanine or leucine in hASIC1a before these mutants were expressed in CHO cells for patch-clamp experiments. We found that currents mediated by hASIC1a(K380A) mutant or mASIC1a(K378A) mutant displayed diminished sensitivity to glutamate at pH 6.55–7.0 (Fig. 3b and Extended Data Fig. 6a). By contrast, no changes in the magnitude of potentiation were observed in the hASIC1a K384A, Q225L and K388A mutants as compared to the wild type. Glutamate lost its PAM effect on the hASIC1a K392A and R160A mutants (Extended Data Fig. 6b,c). The three-dimensional structural proximity of Lys380 and Lys392 rationalized these two residues orientated towards the same binding pocket. The Arg160 mutant was not pursued further because it generated very little currents with markedly altered kinetics, making it difficult to ascertain whether this mutation perturbed protein structural stability, glutamate binding, and/or proton binding or channel gating. Finally, potentiation of $I_{ASICs}$ by glutamate in CHO cells co-expressing the hASIC1a(K380A) mutant subunit and ASIC2a or 2b subunit was significantly reduced (Fig. 3c). These data indicated that Lys380 in hASIC1a is necessary for glutamate binding, suggesting that it is a key residue within the glutamate-binding pocket that enables its role as a PAM for hASIC1a. The K380A mutant probably did not affect the stability of the protein as its current was comparable to that of the wild type. Predictions of the protein stability after K379A mutation using the program MAESTRO suggested no significant effect on either trimer or monomer stability[28] (Supplementary Table 3).

To gain molecular insights into glutamate interactions within the proposed binding pocket around Lys380 of hASIC1a using an independent method, we performed further molecular docking by placing glutamate near the chain A of cASIC1a Lys379 at the outer vestibule of the ion permeation pathway (Fig. 3d). In Fig. 3e, the optimal glutamate docking pose to the putative binding site is shown with optimized side-chain positions of five interacting residues in the vicinity. Glutamate appears to form four specific hydrogen bonds to Gln227, Pro381, Lys383 and Gln398, two salt bridges to Lys379 and Lys383 (Fig. 3e), giving a very stable glutamate–cASIC1a complex. The putative binding pocket for glutamate is a solvent-exposed and electrostatically positive cavity (Supplementary Table 4), which makes it similar to the proton-binding pocket on ASIC1[26] that resides in its close proximity.

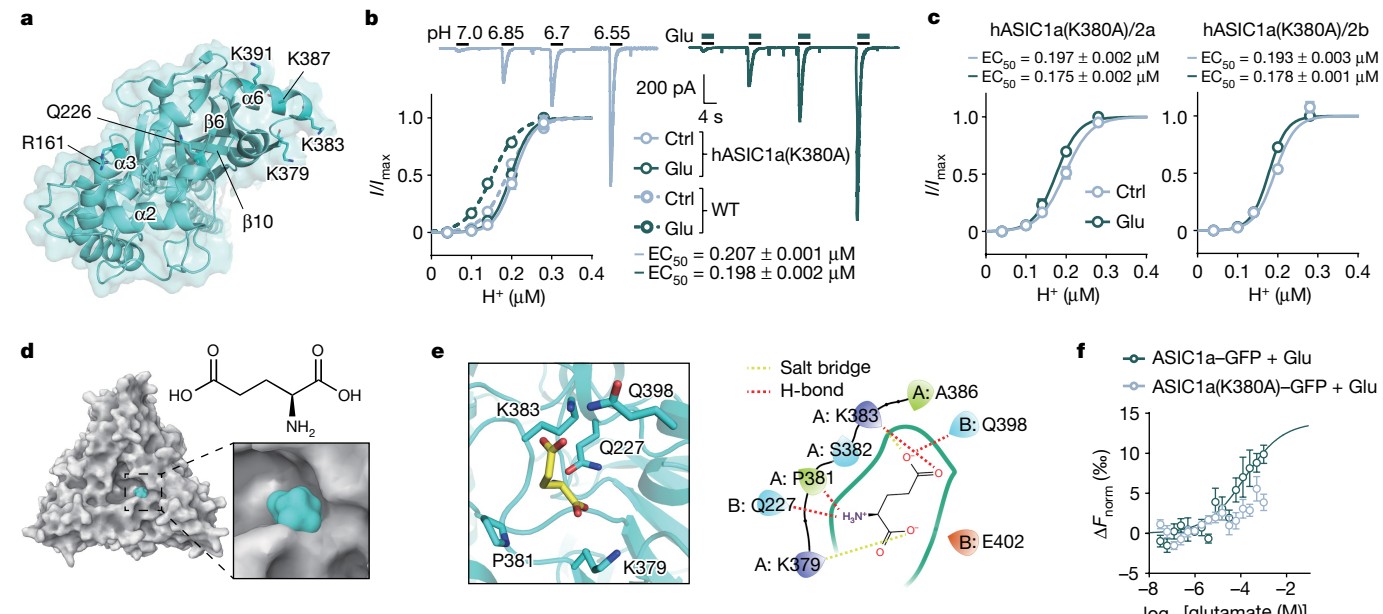

**Fig. 3 | Structure-based determination of the glutamate-binding pocket in the extracellular domain of ASIC1a. a**, A top view of cASIC1a (PBD: 5WKU). Top six scoring binding residues of glutamate by initial docking calculation are shown. For clarity, only chain A is shown here. **b**, $I_{ASICs}$ activated in a human ASIC1a(K380A)-transfected CHO cell in the presence and absence of 500 μM glutamate. Dose–response curves (solid lines) were contrasted to those from wild-type ASIC1a channels as in Fig. 1a (dashed lines). $n = 12$ cells for ASIC1a(K380A). **c**, Dose–response curves for hASIC1a(K380A)/hASIC2a and hASIC1a(K380A)/hASIC2b currents. $n = 10$ cells for each group. **d**, Top view of glutamate-bound cASIC1a. A magnified view of the glutamate-binding pocket is shown at the bottom right. The chemical structure of glutamate is shown at the top right. The surface of cASIC1a and glutamate are coloured white and cyan, respectively. **e**, The putative glutamate-binding pocket near the outer vestibule of the channel pore. **f**, MST assay showing the dose–response curve for glutamate binding with wild-type ASIC1a (data from Fig. 1d) or K380A mutant at pH 7.0. $n = 5$ replicated tests per group. Three to six replicated cultures for patch recordings were tested. Data are mean ± s.e.m.

Notably, the glutamate pocket forms at subunit interfaces stabilized by hydrophobic and polar contacts across the finger, thumb and palm domains. We postulated that, after glutamate binding, the conformation of the thumb domain may undergo rearrangements, which could be transduced to the channel pore through the palm domain to enhance ion permeation.

To determine whether Lys379 of cASIC1a (or Lys380 of hASIC1a) could represent the core binding site for glutamate, we analysed the molecular mechanics/generalized Born surface area (MM/GBSA) binding free energy and docking scores[29] of the glutamate–cASIC1a(K379) and glutamate–cASIC1a(K379A) complex models. We found that both parameters for the wild type were significantly more negative for the wild type compared with for the cASIC1a(K379A)–glutamate complex (Extended Data Fig. 7a), which suggests a destabilization of the complex by Lys379 mutation. Given that glutamate is a natural agonist for NMDARs, we constructed a glutamate–NMDAR complex model to identify any differences in comparison to our glutamate–cASIC1a complex model (Extended Data Fig. 7b). Although no difference was found in the docking scores between glutamate–NMDAR and glutamate–cASIC1a(K379) (wild type) complexes, the MM/GBSA of the glutamate–NMDAR complex was higher than that of glutamate–cASIC1a(K379) complex (Extended Data Fig. 7c), suggestive of higher glutamate-binding affinity to NMDARs than to ASIC1a. Molecular dynamics simulations of the glutamate–cASIC1a(K379) (wild type) and glutamate–cASIC1a(K379A) mutant complexes provide additional support for Lys379 of cASIC1a (that is, homologous site Lys380 of hASIC1a) being important for glutamate binding to ASIC1a channels (Extended Data Fig. 7d–j).

The binding of glutamate to the ASIC1a channel was validated by MST assays in which increasing concentrations of glutamate induced dose-dependent changes in the fluorescence of ASIC1a–GFP in the absence or presence of amiloride (100 μM, an open channel blocker of ASIC1a, non-competitive for glutamate binding) at pH 7.0 (Fig. 3f and Extended Data Fig. 7k,l). By contrast, weaker and unstable binding was observed between glutamate and hASIC1a(K380A) mutant while no conspicuous binding was detectable between glutamate and ASIC1a–GFP at pH 7.4 (Fig. 3f and Extended Data Fig. 7m). Thus, glutamate directly binds to a bona fide cavity at Lys380 of the ASIC1a channel.

## The glutamate binding site is druggable

Our structural findings raised the possibility of developing therapeutics to alleviate neuronal injury by targeting the glutamate binding site on ASIC1a. We tested the effects of a series of candidate chemicals on glutamate-mediated potentiation of $I_{ASICs}$ (Extended Data Fig. 8). Among these, L-aminoadipic acid (L-AA) did not block $I_{ASICs}$, but markedly abolished the potentiation of $I_{ASICs}$ by glutamate (Extended Data Fig. 8g–i). However, L-AA may exert an agonist activity for NMDAR[30] and mGluRs[31], making it unsuitable for ischaemic stroke therapy. CGS19755, a rigid analogue of AP5 with water solubility and blood–brain barrier permeability[32], significantly attenuated glutamate-induced potentiation of $I_{ASICs}$ in CHO cells and culture neurons without affecting $I_{ASICs}$ itself (Fig. 4a–c and Extended Data Fig. 9a). The effect of CGS19755 was dose dependent with a half-maximal inhibition concentration ($IC_{50}$) of $7.7 \pm 2.6$ μM (Fig. 4d). Computational docking showed that the affinity of CGS19755 binding to the cASIC1a(K379) pocket was much higher than that of glutamate (best pose of docking scores: $-6.05$ kcal mol$^{-1}$ (CGS19755 to cASIC1a) and $-3.605$ kcal mol$^{-1}$ (glutamate to ASIC1a)) (Extended Data Fig. 9b and Supplementary Table 4). An MST assay revealed strong binding between CGS19755 and ASIC1a ($K_d = 1.1 \pm 0.2$ μM) (Fig. 4e). These data suggest that CGS19755 might be a potent competitive blocker of the glutamate-binding site on both ASICs and NMDARs.

Our experiments in vitro validated the efficacy of CGS19755 in neuroprotection (Extended Data Fig. 9c–i), raising the hope for drug

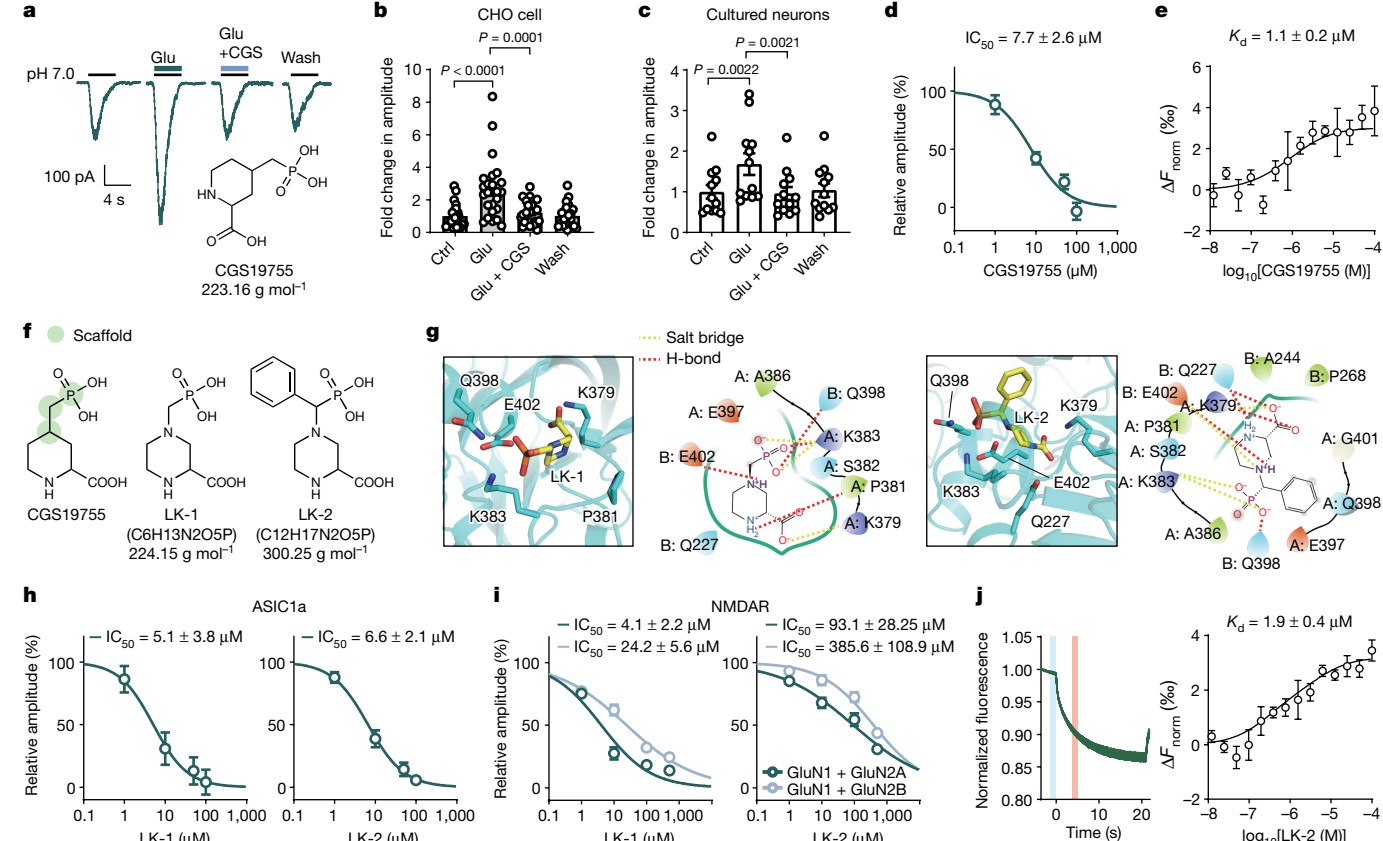

**Fig. 4 | Identification of selective compounds for the glutamate-binding site on ASIC1a. a**, Representative traces showing the effects of glutamate and CGS19755 on $I_{ASICs}$ from an ASIC1a-transfected CHO cell. Inset: the chemical structure of CGS19755. **b**,**c**, Summary plots of CGS19755 on glutamate-induced potentiation of $I_{ASICs}$ from ASIC1a-transfected CHO cells (**b**; $n = 24$ cells) and cultured neurons (**c**; $n = 12$ cells). **d**, Dose–response curve showing that CGS19755 blocked the potentiation of ASIC1a currents by glutamate. $n = 15$ cells. **e**, MST assay showing the dose–response curve for binding between CGS19755 and ASIC1a at pH 7.0. $n = 5$ replicates. **f**, The chemical structures of CGS19755, LK-1 and LK-2. Green coded circles represent the scaffold for virtual screening. **g**, LK-1- and LK-2-bound pockets. **h**, Dose–response curves showing that

glutamate-induced potentiation of $I_{ASICs}$ was inhibited by LK-1 ($n = 14$ cells) and LK-2 ($n = 12$ cells) at pH 7.0. **i**, Dose–response curves showing that LK-1 ($n = 9$ and 17 cells for GluN2A and GluN2B) and LK-2 ($n = 15$ and 21 cells for GluN2A and GluN2B) inhibited NMDAR currents evoked by 100 µM NMDA and 10 µM glycine. NR1 (GluN1) plus NR2A (GluN2A) or NR2B (GluN2B) subunits were co-expressed in CHO cells. **j**, Representative MST traces and dose–response curve showing direct binding between LK-2 and GFP-tagged ASIC1a at pH 7.0. The vertical bars show cold fluorescence detected at around −1 to 0 s (blue) and hot fluorescence detected at 4–5 s (red). $n = 6$ replicated tests. Three to six replicated cultures for patch recordings were tested. Data are mean ± s.e.m. Statistical analysis was performed using one-way ANOVA with Tukey post hoc correction (**b**,**c**).

candidates that selectively target the glutamate-binding site on ASICs but not on NMDARs[32]. This is important because NMDARs are instrumental for brain function under physiological conditions and are even pro-survival during stroke recovery[33], and the inhibition of NMDAR by its high-efficacy antagonists such as CGS19755 (also known as selfotel) may lead to unexpected side effects, such as psychosis shown in previously terminated clinical trials[34,35]. To search for small molecules that could largely block glutamate-dependent enhancement of ASIC activity with minimal effects on NMDAR activity, we performed a CGS19755 structure-based computational drug screen. Two candidate compounds, LK-1 and LK-2, were identified as prototypes of a new class of neuroprotective small molecules (Fig. 4f,g and Extended Data Fig. 10a) on the basis of their molecular docking scores (best pose of docking scores: −6.371 kcal mol⁻¹ (LK-1 to ASIC1a), −4.32 kcal mol⁻¹ (LK-1 to NMDAR), −7.039 kcal mol⁻¹ (LK-2 to ASIC1a) and −4.686 kcal mol⁻¹ (LK-2 to NMDAR)). To verify that LK-1 and LK-2 are pharmacologically effective, we recorded $I_{ASICs}$ by co-applying glutamate and either compound, and found that both compounds reduced glutamate-dependent potentiation of $I_{ASICs}$ in a dose-dependent manner without affecting the basal currents of ASIC1a themselves (Fig. 4h and Extended Data Fig. 10b–d). Importantly, LK-1 and LK-2 were much less effective in attenuating NMDAR currents in acute isolated cortical neurons and

CHO cells that express recombinant NMDARs (Fig. 4i and Extended Data Fig. 10e). NMDARs comprising NR1 (GluN1) plus NR2A (GluN2A) subunits were more sensitive to LK-1 and LK-2 compared with those comprising NR1 plus NR2B (GluN2B) subunits (Fig. 4i). Moreover, LK-2 was found to be a weaker antagonist for NMDARs compared with LK-1 (Fig. 4i), with an IC₅₀ two orders of magnitude higher, implicating its potential to preferentially block the glutamate-binding site on ASICs over that on NMDARs. Differential pharmacological effects of LK-1 and LK-2 on ASIC1a versus NMDARs were consistent with in silico docking results showing higher affinity of LK-1 and LK-2 for ASIC1a than NMDAR, but LK-2 is more likely to spare NMDARs than LK-1. An MST assay showed that LK-2 could directly bind to ASIC1a with a $K_d$ of 1.9 ± 0.4 µM (Fig. 4j). These results rationalized LK-2 as a potent competitive blocker that preferentially targets the glutamate-binding site on ASIC1a channels over that on NMDARs.

## LK-2 provides neuroprotection

To test whether small-molecule LK-2 could provide neuroprotection, we performed cell death and LDH-release assays at pH 7.0 after 1 h OGD in vitro, and found that LK-2 yielded protective effects comparable to CGS19755. This implicated glutamate-mediated enhancement of ASIC

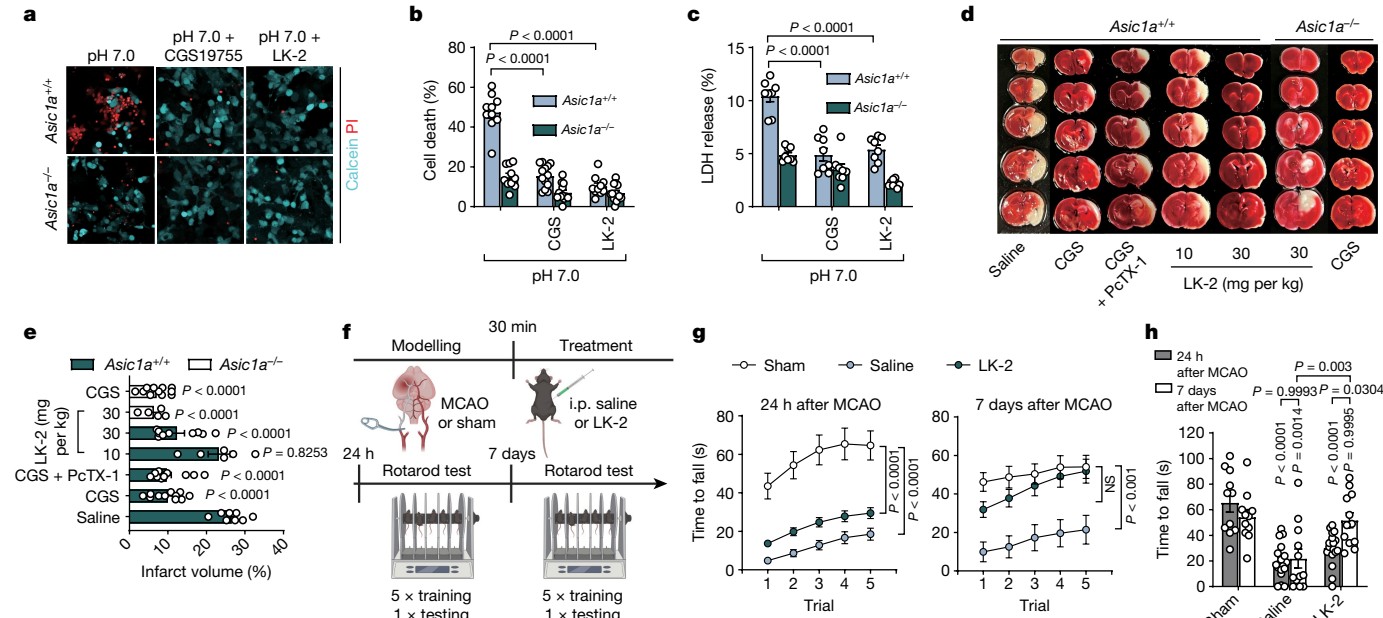

**Fig. 5 | Targeting the glutamate-binding site on ASICs with compound LK-2 is neuroprotective. a**, Calcein–propidium iodide staining of cultured neurons with different treatments at pH 7.0 after 1 h OGD with 10 μM CGS19755 or 10 μM LK-2. **b**, The percentage of cell death from the experiment in **a** was calculated by counting the number of calcein-stained (live cells) and propidium iodide-stained (dead cells) stained cells. *n* = 10–13 images from three replicated cultures for each group. **c**, LDH release of cultured neurons with different treatments after 1 h OGD. *n* = 8 cultures for each group. **d**,**e**, Images of brain slices (**d**) and quantification of the infarct volume (**e**) after MCAO in mice administrated with CGS19755 (1 mg per kg, intraperitoneally (i.p.)), PcTX-1 (100 ng per kg, i.n.) and LK-2 (10 mg per kg and 30 mg per kg, i.p.). *n* = 8, 12, 14, 6, 10, 5 and 11 mice for

each group. **f**, The experimental timeline of the MCAO mouse modelling and behaviour tests. One dose of 30 mg per kg LK-2 was applied. The diagram was created using BioRender. **g**, Motor learning performance was assessed by plotting the time for mice to fall off a rotarod over 5 trials 24 h and 7 days after MCAO in the sham-, saline- and LK-2-treated groups. **h**, Summary of the time for mice to fall off the rotarod was assessed by 1 test trial 24 h and 7 days after MCAO. For the behaviour tests, *n* = 11, 18 and 19 (each group 24 h after MCAO), and *n* = 11, 12 and 12 (each group 7 days after MCAO) mice. Statistical analysis was performed using one-way ANOVA (**e**) and two-way ANOVA (**b**,**c**,**g**,**h**) with Tukey post hoc correction. NS, not significant.

activity rather than NMDAR overactivation as the main contributor to cell injury under ischaemic conditions (Fig. 5a–c). LK-2 can protect ASIC1a-transfected CHO cells after prolonged (1 h) exposure to glutamate at pH 7.0, further demonstrating its efficacy to alleviate cell injury (Extended Data Fig. 10f,g).

To directly investigate the effects of LK-2 in a mouse stroke model in vivo, we first tested whether LK-2 could pass through the blood–brain barrier by measuring its pharmacokinetics in mice. Injection of LK-2 at a dose of 30 mg per kg (intraperitoneally) led to a maximal concentration of 92,660 μg l⁻¹ in the plasma and 236 μg l⁻¹ in the brain tissue, demonstrating that LK-2 could penetrate the blood–brain barrier and potentially reach a micromolar concentration in cerebrospinal fluid to block the glutamate-binding site on ASICs, if considering a dilution factor of around 50 for fluid extracted from the wet brain tissue[36] and a local increase in blood–brain barrier permeability around the infarct (Supplementary Table 5). Moreover, LK-2 had a terminal elimination half-life ($t_{1/2}$) of 2.297 h, presuming its neuroprotective effects within the critical window of intervention after the onset of stroke (Supplementary Table 5 and Extended Data Fig. 10h). Thus, LK-2 displays favourable properties for neuroprotection against ischaemic brain injury in vivo.

Using the mouse MCAO model, we assessed the infarct volume with and without injection of LK-2 at the time of reperfusion (that is, 30 min after artery occlusion). We observed a significant reduction in brain damage by LK-2 at 30 mg per kg (intraperitoneally) compared with in the saline group in *Asic1a⁺/⁺* mice (Fig. 5d,e). And the protective effect provided by LK-2 (30 mg per kg) for brain damage was comparable to that provided by CGS19755 (1 mg per kg, intraperitoneally) (*P* = 0.8288). *Asic1a⁻/⁻* mice were resistant to brain injury by the MCAO paradigm and further administration of CGS19755 (1 mg per kg, intraperitoneally) added little to further neuroprotection (Fig. 5e and Supplementary

Table 6), reinforcing the indispensable role of glutamate-dependent potentiation of ASICs in mediating ischaemic brain injury.

As the MCAO model mainly results in infarction of the ipsilateral frontal cortex (including the motor cortex) and part of the striatum in the middle cerebral artery territory[37], on which sensorimotor behaviours depend, we examined mice using the rotarod task, a commonly used rodent motor behaviour paradigm[38]. The time for mice to fall off the rotating rod was regarded as the readout, 24 h and 7 days after MCAO, to evaluate the short-term and long-term effects of LK-2 on motor learning (a part of cognitive function) and movement coordination ability (Fig. 5f). We found that LK-2-treated mice showed only marginal improvement in their latency to stay on the rotarod 24 h after surgery. But, by day 7 after MCAO, they displayed substantially superior motor learning and coordination ability compared with the saline-treated mice, as reflected by the steep slope of motor learning plots over five test trials with their final latency comparable to that of the sham group (Fig. 5g,h). No differences in improvement of motor behaviour were found between LK-2-treated male and female mice (Extended Data Fig. 10i), indicating its sex-independent efficacy for facilitating long-term recovery. We conclude that LK-2 functions as a competitive antagonist for the glutamate-binding site on ASICs and provides neuroprotection independent of its action on NMDARs, ultimately improving neurobehavioural outcomes after stroke.

## Discussion

Here we demonstrate that glutamate and its structural analogues, including agonists for NMDARs and even competitive NMDAR antagonist L-AP5 function as PAMs to potentiate currents mediated by ASICs. NMDAR antagonists including CGS19755, which are effective in animal

models of stroke, have failed in clinical trials due to severe side effects, including neuropsychiatric and tolerance issues[34,39,40]. NMDAR antagonists not only block pro-survival functions of NMDARs during stroke recovery but also induce severe psychosis[41], making NMDARs difficult to target. Previous studies showed that TRPM2, TRPM4 and ASIC1a channels directly interact with NMDARs[5–7]. The mapping of their respective interfaces provided means to disrupt NMDAR signalling without direct perturbations to the multifaceted functions of NMDARs in synaptic transmission and plasticity that are critical for cognitive functions. These studies all converged onto decoupling NMDARs from cell-death signalling. Our study showed that glutamate at clinically relevant concentrations potentiated $I_{ASICs}$ and largely mediated neurotoxicity in vitro and in vivo. Indeed, ASIC1a is responsible for acidosis-mediated ischaemic brain injury and cell death of ischaemic core at a low pH 6.5–6.0 (refs. 4,42,43). The potentiation of ASIC1a currents by glutamate at a mild pH range (for example, 7.0–6.5) sensitizes neurons to acidosis relevant to clinical conditions of ischaemic cell injury and death in penumbra. Synergistic actions of protons and glutamate on ASICs at a pH range of around 7.0 may maximally boost $Ca^{2+}$ influx, mitochondrial dysregulation, membrane excitability and downstream signalling, driving cell death to expand infarct volume after ischaemic insult and reperfusion. Previous research showed that the activation of NMDAR potentiates ASIC1a currents through intracellular signalling cascades (that is, after 5 min OGD), indicating that acidosis neurotoxicity is secondary to NMDAR activation[5]. Our study argues that glutamate and protons, co-released from synaptic vesicles[44], can act as the first messengers for both NMDARs and ASICs and their downstream cell-death signalling cascade at the very early onset of ischaemic insult. Neuroprotective effects of LK-2 against ischaemic injury favour the working model that glutamate binding to ASIC1a may be involved after the onset of ischaemia, rather than being secondary. Aside from this pathological link to neurotoxicity, our findings implicate physiological roles of glutamate binding to ASIC1a in synaptic transmission and plasticity underlying learning and memory and other behaviours[45–48].

We demonstrated that glutamate binds directly to a cavity near the outer vestibule of the ASIC1a channel pore, with Lys380 being a critical amino acid residue for glutamate binding. Potent neuroprotection by the classical competitive NMDAR antagonist CGS19755 is probably accounted for by its dual bindings to this cavity on ASICs and that on NMDARs. LK-2 stemmed from the structure of CGS19755 probably spares NMDARs if properly dosed. In contrast to classical ASIC blockers with poor selectivity or blood–brain barrier permeability in drug delivery[4,49], LK-2 at low micromolar concentrations preferentially decouples glutamate from ASICs, showing a promising neuroprotective efficacy and neurobehavioural recovery after stroke. Together, these findings build a conceptual paradigm for mechanistic understanding of neurotoxicity in ischaemic stroke and for strategizing new stroke therapeutics independent of NMDARs.

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

## Methods

### Animals

Wild-type C57BL/6J mice were purchased from Vital River company (China) and housed in groups, in the animal facility at Shanghai Sixth People's Hospital, Shanghai Jiaotong University. *Asic1a*$^{-/-}$ (with congenic C57BL/6J background) mice were provided by the T.-L. Xu laboratory. All mice were kept in standard cages (15 cm × 21 cm × 13.5 cm) under a 12 h–12 h light–dark cycle with ad libitum access to food, water and nesting material. The mice were checked daily and their weight was monitored during the experiments. Mice were randomly allocated to treatment groups.

### Ethical approval

Throughout the study, all efforts were carried out to minimize animal suffering and reduce the number of animals. Experiments were conducted in conformity with the institutional guidelines for the care and use of animals, and experimental protocols that were approved by the Ethics Committee of Shanghai Sixth People's Hospital.

### Electrophysiology analysis of cultures

**Whole-cell recordings.** Whole-cell patch clamp recordings were taken from either neurons or CHO cells in a recording chamber (Corning) mounted onto a fixed-stage upright microscope (Nikon). Patch electrodes (4–6 MΩ) were made from 1.5 mm borosilicate glass (World Precision Instruments). Whole-cell currents were recorded using an EPC-10 patch-clamp amplifier (HEKA). Data were acquired at 10–20 kHz and filtered at 1–3 kHz using a computer equipped with the Pulse 6.0 software (HEKA, Lambrecht). Cells were recorded at a holding potential of −60 mV unless otherwise described. For NMDAR current recordings, the holding potential was −70 mV. A multi-barrel perfusion system (SF-77B, Warner Instruments) and pressure regulator system (ALA-VM8, Scientific Instrument) were used to achieve a rapid exchange of extracellular solutions. The perfusion protocol was set and controlled by the PatchMaster software. To avoid use-dependent desensitization or rundown, ASICs were repeatedly activated by acidic solution at minimal interpulse intervals of >1 min. During each experiment, a voltage step of −10 mV from the holding potential was applied periodically to monitor the cell capacitance and access resistance.

Dose–response curves were fitted to the Hill equation: $a = I/I_{max} = 1/[1 + 10^{n(b - EC_{50})}]$, where $a$ is the normalized amplitude of the $I_{ASICs}$, $b$ is the concentration of proton in external solution ([H$^+$]), $EC_{50}$ is the proton concentration or [H$^+$] yielding half of the maximal peak current amplitude and $n$ is the Hill coefficient. The $IC_{50}$ values for blocker dose–response curves were fitted using the following equation: $I/I_{max} = 1/\{1 + [IC_{50}/(\text{blocker concentration})]^n\}$, where $n$ is the Hill coefficient and $IC_{50}$ is the concentration of blocker producing 50% of the maximal block ($I_{max}$).

**Single-channel recordings.** Unitary currents were recorded using the outside-out configuration of the patch-clamp technique in ASIC1a transfected CHO cells. Channels were activated by rapidly moving squared-glass tubes delivering solutions of desired pH in front of the tip of the patch pipette. The delivery device achieves complete solution changes within 20 ms (SF-77B). When filled with solutions, pipettes had resistances of 6–10 MΩ. Single-channel currents were recorded using the EPC-10 patch-clamp amplifier (HEKA). The data were collected at 10 kHz and gain value 200 mV pA$^{-1}$, filtered at 1 kHz, and stored on a computer for analysis. Data were filtered off-line with a digital Gaussian filter to 1 kHz. All points amplitude histograms of ASIC unitary currents are fitted to the two-exponential equation:

$$a = A1 \times \exp\{-0.5 \times [(b - M1)/SD1]^2\}$$

where $a$ is the count of ASIC unitary currents, $b$ is the amplitude of ASIC unitary currents. A1 and A2 are the heights of the centre of the distribution, and M1 and M2 are the amplitude of ASIC unitary currents at the centre of the two distributions. SD1 and SD2 are measures of the widths of the distributions, in the same units as $b$ (pA).

**Solutions and chemicals.** The artificial cerebrospinal fluid (aCSF) contained 124 mM NaCl, 5 mM KCl, 1.2 mM KH$_2$PO$_4$, 1.3 mM MgCl$_2$, 2.4 mM CaCl$_2$, 24 mM NaHCO$_3$ and 10 mM glucose, with pH 7.4 (300–330 mOsm). The ECF for culture cells or neurons contained 140 mM NaCl, 5 mM KCl, 1 mM CaCl$_2$, 1 mM MgCl$_2$, 10 mM glucose and 25 mM HEPES. For solutions with pH ≤ 6.8, MES was used instead of HEPES for stronger pH buffering. For NMDAR current recordings, MgCl$_2$ was removed. The intracellular solution for voltage-clamp recordings contained 130 mM K-gluconate, 2 mM MgCl$_2$, 1 mM CaCl$_2$, 10 mM HEPES, 0.5 mM EGTA, 4 mM Mg-ATP, with pH 7.3, and the osmolality was adjusted to 290–300 mOsm. For isolated neuron and NMDAR current recordings, K-gluconate was replaced by CsCl. All experiments were performed at room temperature (21–26 °C).

D-AP5, CGS19755, amiloride and PcTX-1 were purchased from Alomone Labs; L-AP5 and CGP39551 were purchased from Tocris; NBQX, LY341495, CGP37157 and RO2959 were purchased from MedChemExpress; and other chemicals were purchased from Sigma-Aldrich or Macklin. Note that the pH of all ECF in this study with any chemicals added was adjusted to the indicated values (the pH changes less than 0.02 after adding chemicals).

**Cell culture and transfection.** Chinese hamster ovary (CHO) K1 and HEK293T cells (provided by Stem Cell Bank, Chinese Academy of Sciences) were cultured in F12 and DMEM medium with 10% FBS (Gibco), respectively. Penicillin–streptomycin (Invitrogen) was added to the medium for preventing bacteria contamination at a final concentration of 1%. For NMDAR expression, 500 µM D-AP5 was added in the cultured medium to avoid glutamate-induced toxicity through activating NMDAR[50]. 0.25% Trypsin-EDTA (Gibco) was used for cell passage. Cells were incubated at 37 °C in a humidified CO$_2$ incubator. Cells were transfected with plasmids described in detail previously[51]. In brief, at a cell density of 50–70%, a total 3.0 µg cDNA mixed with the Lipofectamine 3000 transfection kit (Invitrogen) was added to 35 mm dishes. For co-expression of ASIC1a plus 2a or 2b, equal amounts of cDNA for both subunits were used. For NMDAR expression, plasmids of GluN1, GluNR2A or GluNR2B and PSD-95 at a ratio of 1:4:0.5 were used. cDNA of green fluorescent protein (GFP) was linked at the N terminus of those proteins. All experiments were performed 24–48 h after transfection, all dishes were washed three times using ECF before experiments, and GFP-positive CHO cells were viewed under a fluorescence microscope for patch-clamp recordings. Human and mouse ASIC subunits were used for constructing all wild types and mutants in this study.

**Primary cortical neuron culture.** Primary cortical neurons were prepared and maintained as previously described[52]. In brief, cerebral cortices from 24 h postnatal *Asic1a*$^{+/+}$ or *Asic1a*$^{-/-}$ mice were dissected in DMEM high-glucose solution and dissociated by 0.05% trypsin for 15 min. Cells were plated (~2 × 10$^5$ cells per 35 mm dish for electrophysiology and cell death experiments) on Matrigel-coated (Corning) cover glasses or dishes. Cultures were maintained in Neurobasal-A medium (Gibco) containing 2% B27 (Gibco) and 1% GlutaMax supplements (Gibco) at 37 °C under a 5% CO$_2$ humidified atmosphere.

**Acute isolation of cortical neurons.** Acute dissociation of mouse cortical neurons was performed as described previously[53]. In brief, mice from postnatal day 15 to 18 were anaesthetized with isoflurane. Cortical tissues were dissected and incubated in oxygenated ice-cold aCSF (with half concentration of CaCl$_2$). Transverse cortical slices (500 µm) were cut with a microtome (Leica VT1200) followed by incubation in

aCSF containing 3.5 mg ml$^{-1}$ papain (Sigma-Aldrich) at 37 °C for 30 min. The slices were then washed three times and incubated in enzyme-free ECF solution for at least 15 min before mechanical dissociation. For dissociation, slices were triturated using a series of fire-polished Pasteur pipettes with decreasing tip diameters. Recording began 15 min after the mechanical dissociation. Only the neurons that retained their pyramidal shape and dendrites were used for recordings.

## Site-directed mutagenesis

All ASIC1a mutants were obtained through PCR mutagenesis using wild-type *Asic1a* gene, inserted into the pEGFPC3 vector (Clontech) using the Seamless Cloning kit (Sunbio). For single-site mutations, the PCR (50 µl) reaction contained 1 µl DNA template, 2 µl primer pair, 4 µl dNTPs and 0.5 µl of high-fidelity DNA polymerase (PrimeSTAR, Takara). The PCR cycles were initiated at 98 °C for 3 min, followed by 30 amplifications (98 °C for 10 s, 55 °C for 15 s and 72 °C for 1 min). The last step was performed at 72 °C for 10 min. Each mutant cDNA was transformed into *Escherichia coli* chemically competent cells. All mutants were sequenced by PCR sequence analysis to confirm the presence of the desired mutation and the absence of undesired mutations.

## Ca$^{2+}$ imaging

Primary cortical neurons were incubated in a confocal microscopy (Zeiss 710) specialized dish (35 × 35 mm, Cellvis) with ECF (saturated with 95% O$_2$) and 1 µM fluo-3 AM (Beyotime) for 20 min at room temperature, followed by three washes and additional incubation in normal ECF for 15 min. The dish was then transferred onto the stage of the confocal microscope. Neurons were illuminated using a xenon lamp and observed with a ×40 UV fluor objective lens. The shutter and filter wheel were set to allow for 488 nm excitation wavelength. Images were analysed every 5 s in circumscribed regions of cells in the field of view. Digital images were acquired, stored and analysed by the ZEN software (Zeiss). For quantification, intracellular calcium levels were plotted as $\Delta F/F_0$ ratios over time, where $F_0$ is the initial fluorescence intensity of each cell.

## Imaging of mitochondrial membrane potential

The mitochondrial membrane potential ($\Psi$m) was measured using JC-1 (Beyotime). Primary cortical neurons were loaded with JC-1 working solution in for 20 min, then washed and incubated in JC-1 wash buffer for 5 min before recording. JC-1 was imaged with 488 nm and 546 nm excitation wavelengths using a 40× objective. For quantification, the $\Psi$m was measured as the fluorescence intensity ratio of red and green (R/G), then the ratio was normalized to the initial fluorescence intensity ratio of each cell.

## OGD analysis

Cultured neurons were washed three times and incubated with glucose-free ECF (glucose was replaced by an equal concentration of sucrose) at pH 7.4 or 7.0 in an anaerobic chamber with an atmosphere of 95% N$_2$ and 5% CO$_2$ at 35 °C (controlled by an OxyCycler oxygen profile, BioSpherix). OGD was terminated after 1 h by replacing the glucose-free ECF with normal ECF.

## Calcein–propidium iodide cell death assay

For cultured neurons, the calcein–propidium iodide cell death assay was performed after 1 h OGD and neurons were exposed to 1 µM calcein-AM and 2 µM propidium iodide (Solarbio) for 15 min (ECF was saturated with 95% O$_2$ and 5% CO$_2$). For ASIC1a–GFP-transfected CHO cells, the assay was performed after 1 h different treatments and the cells were exposed only to 2 µM propidium iodide for 15 min. All cells were then washed three times with ECF. After fixation in 4% paraformaldehyde (Solarbio) for 30 min, images were acquired shortly after staining by confocal microscopy. The live cells and apoptotic nuclei were determined by 488 nm and 546 nm excitation wavelengths using

microscopy examination at ×40 magnification. The number of staining nuclei was counted using the Image J software. The percentage of cell death was calculated as: cd% = [$n_{red}/(n_{red} + n_{green})$] × 100, where cd% is the percentage of cell death, $n_{red}$ and $n_{green}$ are the number of PI-stained (dead cells) and calcein-stained (live cells) cells or GFP-labelled cells, respectively.

## Cell injury assay with LDH measurement

LDH release for cultured neurons was measured after 1 h OGD using the LDH assay kit (Beyotime). After incubation, culture medium (60 µl) was transferred to 96-well plates and mixed with 30 µl reaction solution provided by the kit for 30 min (37 °C). The optical density was measured at 490 nm and the background absorbance was subtracted at 620 nm, using a microplate photometer (Multiskan MK3, Thermo Fisher Scientific). The maximal releasable LDH in each group was obtained by incubation with 1% Triton X-100 for 30 min.

## Glutamate-release assay

Glutamate production was measured using the Glutamate Assay Kit (Sigma-Aldrich, MAK004). In brief, the supernatants of cultured neurons were deproteinized with a 10 kDa MWCO spin filter before addition to the reaction. Glutamate standard was prepared by diluting 0.1 M glutamate (provided in the kit) to a different concentration gradient. The glutamate standard and each sample were added into a 96-well plate (50 µl per well), and 100 µl reaction solution was added to each well and mixed thoroughly. The mixture was incubated at 37 °C for 60 min. The absorbance of each well was measured on a microplate reader using 450 nm as the primary wavelength. The glutamate concentration of each sample was calculated by glutamate standard curve.

## MCAO

Transient focal ischaemia was induced by suture occlusion of the MCAO in *Asic1a*$^{+/+}$ and *Asic1a*$^{-/-}$ mice (the same number of male and female mice beyond 6 weeks was used). Animals were anaesthetized using an animal mini anaesthesia machine (RWD). During anaesthesia induction, the mice were put into a chamber (15 cm × 25 cm × 30 cm) with 2% isoflurane; the mice were put on a mask with 1% isoflurane during MCAO operation. Adequate ischaemia was confirmed by continuous laser Doppler flowmetry (moor FLPI-2). Mice that did not have a significant decrease in blood flow to less than 50% baseline values during MCAO were excluded. Rectal and temporalis muscle temperature was maintained at 33 ± 0.5 °C with a thermostatically controlled heating pad (Supplementary Table 6). Intraperitoneal injection was performed immediately after removing the suture occlusion. Mice were killed with isoflurane overdose 24 h after ischaemia.

Brains were removed and dissected coronally at 1 mm intervals, and stained with the vital dye TTC and the normal area was stained with TTC. Infarct volume (%) was calculated by summing infarction areas of all sections and multiplying by slice thickness, then dividing the whole volume of the brain. Manipulations and analyses were performed by individuals blinded to treatment groups. Depending on the experimental design, 30 min MCAO was performed for moderate ischaemic model.

## Laser speckle imaging

Mice were anaesthetized by 1% isoflurane and their head was restrained in a stereotaxic cylinder frame to minimize breathing motion. The scalp and the skull fascia were gently incised down the midline and peeled to the side. Saline was titrated onto the skull to keep it moist. Laser speckle images were recorded using a CMOS camera before MCAO, 15 min after occlusion and 15 min after reperfusion. For each animal, three sets of raw speckle images were acquired in <15 s (250 frames in each set; image width, 752 pixels; image height, 580 pixels; exposure time, 20 ms). A speckle contrast image was calculated from each raw speckle image using a sliding grid of 2.5 mm × 2.5 mm. A mean speckle contrast image was calculated for each set and used to calculate the rCBF.

The rCBF in the ipsilateral (ischaemic) hemisphere was normalized to the mean rCBF in the contralateral (non-ischaemic) hemisphere. Speckle images were obtained and processed using mFLPI2Meas (v.2.0), rCBF data from all pooled hemispheres were obtained using moorFLPI-Review (v.50). All analyses were randomized.

## Molecular docking and Prime-MM/GBSA binding free-energy calculation

The structures of cASIC1a (PDB ID: 5WKU; resting state) and NMDAR (PDB: 5IOU) were obtained from the PDB. Structures of chemicals (that is, L-glutamate, AP5) were obtained from the PubChem compound or ChemBioDraw Ultra 14 software. Initial docking studies involved preparation of the cASIC1a and glutamate using the High-Ambiguity Driven protein-protein DOCKing (HADDOCK) software. The entire accessible extracellular surface area of the protein was considered as a plausible interface with glutamate. After docking calculations, the highest scoring docking poses were analysed. The results revealed several plausible glutamate-binding sites (Supplementary Table 2), which were subsequently validated by site-directed mutagenesis.

After identification of a most probable binding site, additional docking poses were generated using the Schrödinger Maestro software suite (Schrödinger, 2020-3). Before docking, protein was processed using the Protein Preparation Wizard to add missing residues, optimize side-chain positioning, remove bound waters, optimize H bonds and minimize energy (using OPLS3e force field). Ligands were optimized using the OPLS3e force field in LigPrep module. Protein and ligand protonation states at pH 7.0 ± 0.2 were sampled using Epik.

Ligands were docked to an identified residue (for example, Lys379) in a grid box with dimensions of $25 \times 25 \times 25$ Å$^3$. Extra-precision docking (Glide XP) was performed with flexible ligand sampling, and post-docking minimization was performed to generate a maximum of ten poses per ligand within the Glide program. The docking conformation with a highest docking score was further analysed. The binding free energies of all different poses from XP docking outputs were carried out using the Prime-MM/GBSA module[29]. The binding energy proxy was calculated by the software according to the following equation:

$$\Delta G = E_{\text{complex (minimized)}} - [E_{\text{ligand (minimized)}} + E_{\text{receptor (minimized)}}]$$

## Molecular dynamics simulations

For molecular dynamics simulation, the pose with highest docking score of each ligand–protein complex was selected from docking results and the ligand-bound protein systems were built in 150 mM NaCl aqueous solution. To investigate the stability of the docked ligand–protein poses, 50 ns simulations were performed. After 25,000 steps of minimization, the systems were equilibrated using isothermal-isobaric (NPT) ensembles at a constant temperature of 303.15 K, followed by 50 ns production runs. All simulations used the program GROMACS 2020.3. CHARMM36m force field was used for the protein, GROMOS 54A7 force field for ligand[54] and the SPCE model for water. The simulation trajectories were analysed for structural fluctuations using root-mean-square deviation and root-mean-square fluctuation calculations. MM-PBSA calculations on molecular dynamics simulation trajectories were performed with a modified gmx_mmpbsa bash script using solvent-accessible surface area as the model for non-polar solvation energy.

## CGS19755-binding pocket analysis and drug design

The pose with the highest docking score of the CGS19755–cASIC1a complex was selected from molecular docking for binding pocket analysis using Fpocket 2.0 software. A dpocket program analysis was performed to produce pocket parameters using the default settings. We used a scaffold-replacement method based on the CGS19755 structure to screen the ZINC20 database (fragment, lead-like and drug-like

molecules) using the Molecular Operating Environment software (v.2015.10). The three-dimensional conformations of the remaining about 222 compounds were generated by the ligPrep module of Maestro (Schrödinger) with the OPLS3e force field. Possible ionization states of each compound were generated in the pH range of 7.0 ± 0.2 using Ionizer. Possible tautomer forms were also generated for each ligand. Compounds were screened using the high-throughput virtual screening module followed by the extra-precision docking module in Glide. The Glide docking score was used to rank the results list. Finally, six hits were selected for the electrophysiological assay.

## MST assay for protein- and compound-binding assays

The MST measurement for the binding of compounds to ASIC1a was performed using Monolith NT.115 (NanoTemper)[16]. In brief, the HEK293T cell was collected after 48 h transfection with eGFP-tagged wild-type or mutant ASIC1a plasmid and lysed by M-PER mammalian protein extraction reagent (Thermo Fisher Scientific). Protease/phosphatase inhibitor (1%, Cell Signaling) was added to the lysate to avoid protein degradation. The level of ASIC1a was verified by measuring the total fluorescence. For binding studies, the lysate was diluted fourfold using ECF (with 0.1% Tween-20) to provide the optimal level of the fluorescent protein in the binding reaction. Compounds were titrated by ECF at a 1:1 ratio, diluted 16 times. Subsequently, 5 µl cell lysate was mixed with 5 µl compounds at different concentrations. After 5 min incubation at room temperature, all of the samples were loaded into MST NT.115 standard glass capillaries and measurement was performed at 70–100% excitation power to control the fluorescence value between 300 and 400 using the MO control software (v.1.6.1). The thermophoresis time ($t$) was 23 s, and the experimental temperature ($T$) was 25 °C. At least three independent experiments were repeated and then the data were imported into MO affinity analysis software (v.2.3) of NanoTemper was used to calculate the $K_d$ value using the $K_d$ model. The cold and hot fluorescence were measured at −1–0 s and 4–5 s to avoid thermally induced protein configuration change, respectively. The ECF pH was 7.0 unless otherwise described.

## Chemical synthesis

LK-1, LK-2 and other screened compounds were custom synthesized by Simcere Pharmaceutical for this study (Supplementary information). For electrophysiology, stock solutions of two compounds were prepared in water and diluted in the ECF solutions before use for in vitro experiments or in saline for animal studies in vivo.

## Pharmacokinetics study

Pharmacokinetics of LK-2 was analysed in male C57BL/6J mice ($n = 22$). Plasma and brain concentrations were determined using LC–MS/MS methods after a single intraperitoneal injection dose (30 mg per kg) of compound as a clear solution in 0.9% saline at a concentration of 1 mg ml$^{-1}$. Blood samples were collected into an EDTA-coated test tube at the time points of 0.083 h, 0.25 h, 0.5 h, 1.0 h, 2.0 h and 4.0 h, and then centrifuged at 2,000$g$ for 15 min to generate plasma samples. Brain samples were collected after intraventricle perfusion with normal saline and prepared by homogenizing tissue with 5 volumes (w:v) of 0.9% NaCl. LC–MS/MS methods to quantify LK-2 in plasma and brain samples were developed with the LC-MS/MS-T API 4000 instrument. General sample processing procedure was performed as follows: (1) an aliquot of 30 µl plasma sample, calibration standard, quality control, single blank and double blank sample was added to the 1.5 ml tube. An aliquot of 40 µl brain homogenate, calibration standard, quality control, single blank and double blank sample was added to the 96-well plate respectively. (2) Each sample (except the double blank) was quenched with 150 µl (for plasma samples) or 200 µl (for brain homogenates) of IS 1 (6 in 1 internal standard in methanol (labetalol, tolbutamid verapamil, dexamethasone, glyburide and celecoxib, 100 ng ml$^{-1}$ for each) with 40 mM DBAA) (the double blank sample

was quenched in methanol with 40 mM DBAA), and the mixture was then vortex-mixed well (at least 15 s) and centrifuged for 15 min at 12,000$g$ (for plasma samples) or 3,220$g$ (for brain homogenates) at 4 °C. (3) An aliquot of 65 μl supernatant was transferred to the 96-well plate and centrifuged for 5 min at 3,220$g$ at 4 °C. All of the processes were performed on the wet ice. All of the supernatants were directly injected for LC–MS/MS analysis. The column used was an ACQUITY UPLC BEH C18 2.1 × 100 mm, 1.7 μm column. The column temperature was 40 °C. The flow rate was 0.4 ml min$^{-1}$. The mobile phase consisted of A: 0.001% $NH_3 \cdot H_2O$ with 0.18 mM DBAA in water; and B: 10 mM DMHA and 3 mM $NH_4OAc$ in ACN/Water (v:v, 50:50). Standard curves were prepared by spiking compounds into control plasma and brain and these were used to determine drug concentrations. Pharmacokinetic parameters were calculated by non-compartmental analysis using DAS v.2.0 with the mean concentration at each timepoint.

## Rotarod test

Motor behaviours were tested using the rotarod test (RWD). The rotarod measures the ability of mice to maintain balance on a motor pressure-driven rotating rod. In this test, mice were put on a rotating rod, and the time to fall off was recorded. Mice were trained five times to learn moving on the rod before testing, in which the speed was accelerated from 4 to 40 rpm in 3 min. The rod was cleaned with alcohol before each test.

## Statistics

All data are reported as mean ± s.e.m. The number of biological replicates has been reported as nested data. Two-tailed paired and unpaired Student's $t$-tests were used where appropriate to examine the statistical significance of the difference between groups of data. Comparisons among multiple groups were analysed using one-way and two-way ANOVA followed by Tukey multiple-comparison tests for post hoc analysis. Prism 8 was used to analyse all data.

## Reporting summary

Further information on research design is available in the Nature Portfolio Reporting Summary linked to this article.

## Data availability

All data supporting the findings of this study are provided in the Article and its Supplementary Information. Source data are provided with this paper.

## Code availability

The bash script used in MM-PBSA calculations on molecular dynamics simulation trajectories is available at GitHub (https://github.com/Jerkwin/gmxtools/tree/master/gmx_mmpbsa). Graphs were generated using Prism 8.

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

**Acknowledgements** This work was supported by National Key R&D Program of China (2021ZD0201900) to S.-K.Y.; Shanghai Municipal Education Commission-Gaofeng Clinical Medicine Grant Support (20152233), International Cooperation and Exchange of the National Natural Science Foundation of China (82020108008) and Shanghai Jiao Tong University School of Medicine Multicenter Clinical Research Program (DLY201823) to H.-B.S.; the National Natural Science Foundation of China (82101234), China Postdoctoral Science Foundation (2020M681335) and Shanghai Sixth people's Hospital (ynqn202108) to K.L.; and Canadian Institutes of Health Research Project Grants (PJT-156034, PJT-156439 and PJT-191780), Natural Science and Engineering Research Council Discovery Grant (RGPIN-2017-06665) and Tier 1 Canada Research Chair Program (CRC-95-2324) to L.-Y.W. We thank F. Tang and L. Liu for synthesis of all candidate compounds, and A. Fekete, B. Tawfik and A. Wang for comments and edits on an earlier version of the manuscript.

**Author contributions** H.-B.S., L.-Y.W. and S.-K.Y. conceived and designed this study. K.L. performed electrophysiology, in silico simulations and drug screen. I.P. and J.F.-K. performed initial computational modelling for predicting glutamate-binding sites. Z.-Q.L. performed electrophysiology and imaging. L.-N.G. performed cell death assays. H.-W.L and M.-X.L. performed animal modelling and tissue histology. J.-F.L. and X.Q. performed cell culture and transfection. T.-L.X. provided materials and intellectual inputs. L.-Y.W. supervised the study. K.L. and L.-Y.W. wrote the manuscript with edits and revisions by all of the other authors.

**Competing interests** The authors declare no competing interests.

**Additional information**
**Correspondence and requests for materials** should be addressed to Hai-Bo Shi, Lu-Yang Wang or Shan-Kai Yin.

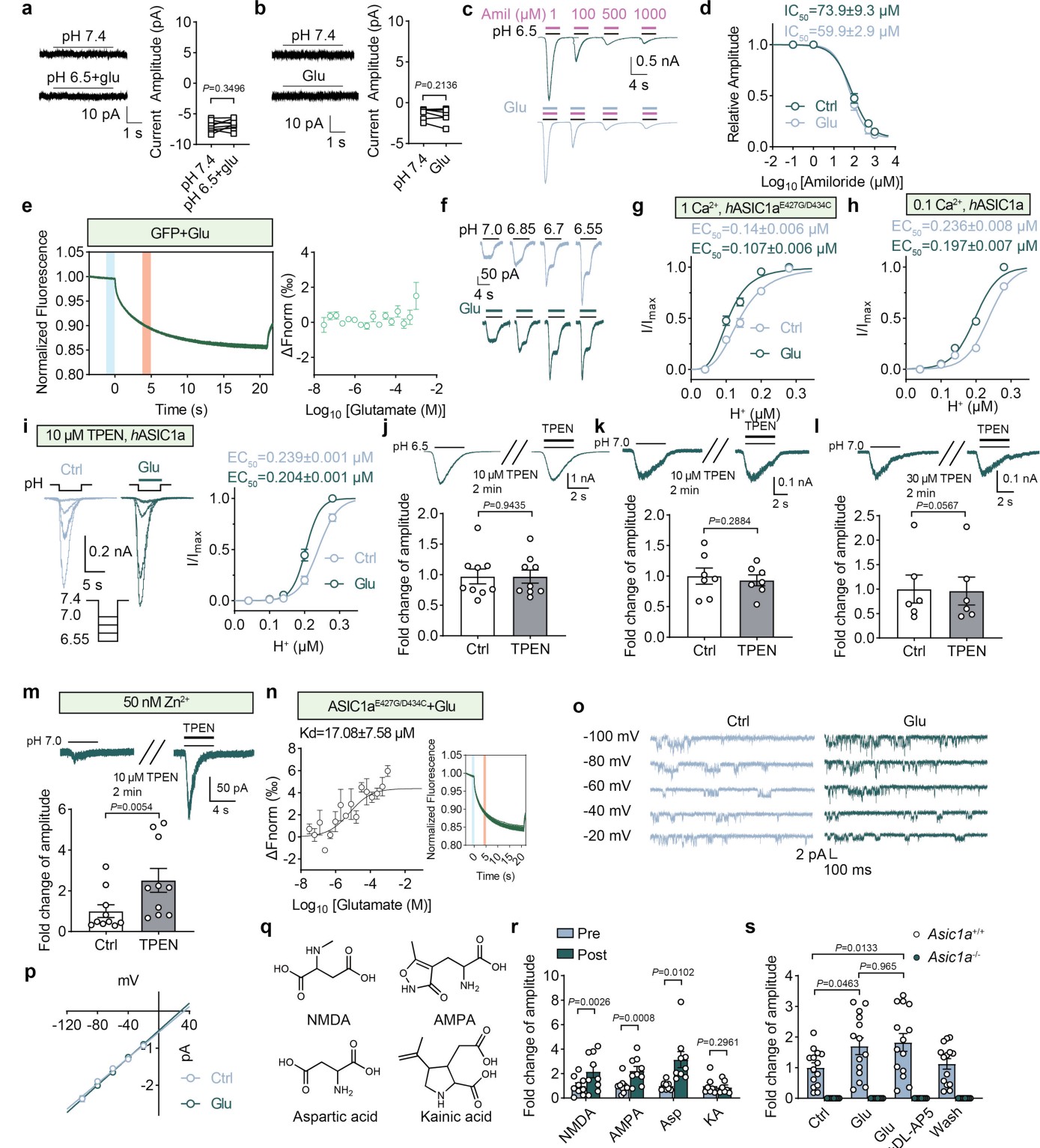

**Extended Data Fig. 1** | See next page for caption.

**Extended Data Fig. 1 | Glutamate-dependent potentiation of $I_{ASICs}$ is independent of $Ca^{2+}$, $Zn^{2+}$ binding sites and does not change ion selectivity.** **a**, Glutamate (500 µM) could not activate currents in blank CHO cells even at pH 6.5. $n$ = 8 cells. **b**, Glutamate (500 µM) could not activate currents in ASIC1a transfected CHO cells at pH 7.4. $n$ = 6 cells. **c, d**, Inhibition of $I_{ASICs}$ by amiloride at a dose-response manner was not affected by 1 mM glutamate. $n$ = 6 cells. **e**, MST assay showing no direct binding between glutamate and GFP at pH 7.0 ($n$ = 4 tests). Vertical bars: blue, cold fluorescence detected at −1 - 0 s; red, hot fluorescence detected at 4 - 5 s. **f, g**, Representative traces and dose-response curves showing glutamate potentiated $I_{ASICs}$ mediated by mutant hASIC1a$^{E427G/D434C}$ devoid of the $Ca^{2+}$ binding sites. $n$ = 7 cells. **h**, Dose-response curves showing glutamate potentiated $I_{ASICs}$ in ECF containing 0.1 mM $Ca^{2+}$. $n$ = 9 cells. **i**, Representative traces and dose-response curves showing glutamate potentiated $I_{ASICs}$ in ECF containing 10 µM TPEN. $n$ = 8 cells. **j**, Representative traces and summary data showing perfusion of 10 µM TPEN for 2 min did not potentiate $I_{ASICs}$ at pH 6.5. $n$ = 9 cells. **k, l**, Representative traces and summary data showing perfusion of 10 µM or 30 µM TPEN for 2 min did not potentiate

$I_{ASICs}$ at pH 7.0. $n$ = 7 and 6 cells, respectively. **m**, Representative traces and summary data showing perfusion of 10 µM TPEN for 2 min potentiated $I_{ASICs}$ at pH 7.0 in the presence of 50 nM $Zn^{2+}$ in ECF. $n$ = 10 cells. **n**, Representative MST traces and dose-response curve showing direct binding between glutamate and GFP-tagged $h$ASIC1a$^{E427G/D434C}$. Vertical bars: blue, cold fluorescence detected at −1 - 0 s; red, hot fluorescence detected at 4 - 5 s. $n$ = 3 replicated tests. **o, p**, Representative traces and current-voltage relationship showing ASIC1a unitary currents recorded at holding potential from −100 mV to −20 mV (20 mV increment) in the presence and absence of 500 µM glutamate at pH 7.0. $n$ = 7 cells. **q**, Chemical structure of glutamate analogs. **r**, $I_{ASICs}$ recorded before and after 500 µM NMDA, AMPA, aspartic acid (Asp) and kainic acid (KA) treatment at pH 7.0. **s**, $I_{ASICs}$ recorded from cultured neurons from *Asic1a$^{+/+}$* ($n$ = 14 cells) and *Asic1a$^{-/-}$* mice ($n$ = 8 cells) at pH 7.0. 500 µM glutamate and 200 µM DL-AP5 were used. Three to six replicated cultures for patch recordings were tested. Data are mean±s.e.m.; two-tailed paired Student's $t$-test (**a,b,j,k,l,m**); two-way ANOVA with Tukey post hoc correction for multiple comparisons (**r,s**); $P$ values are indicated.

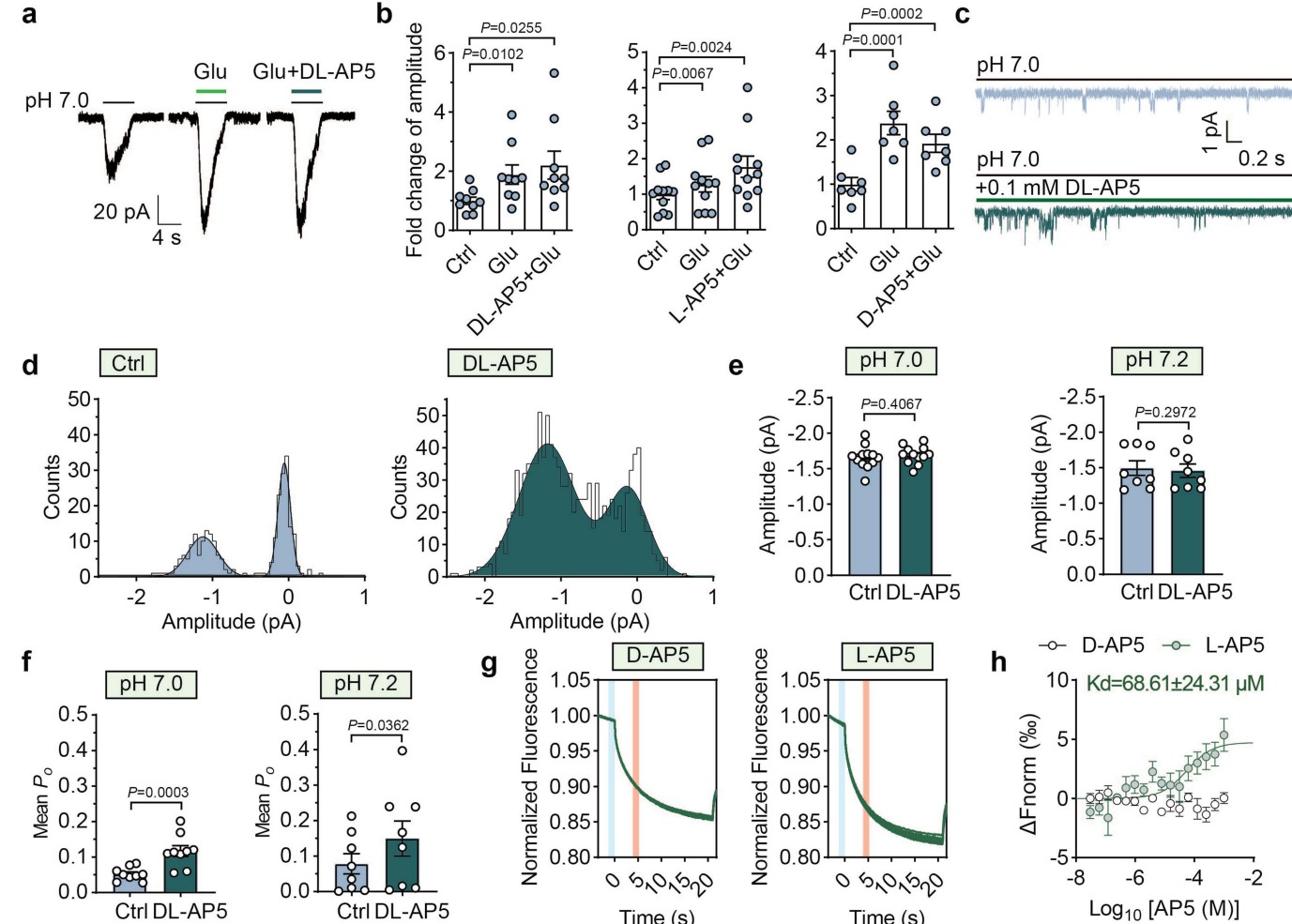

**Extended Data Fig. 2 | NMDAR antagonist AP5 directly binds the ASIC1a channel to enhance its open probability. a**, Typical traces showing the effects of glutamate and DL-AP5 on $I_{ASICs}$. **b**, Summary data showing DL-, L- and D- isomers of AP5 (200 µM) did not block glutamate-enhanced $I_{ASICs}$. $n$ = 9, 11 and 7 cells. **c**, Outside-out patch recordings of ASIC unitary currents in an ASIC1a transfected CHO cell in the presence and absence of 100 µM DL-AP5 at pH 7.0. Currents were recorded at −60 mV. **d**, All points amplitude histogram of of ASIC unitary currents was constructed from (**c**), curves were fitted by two Gaussian components. Bin=0.05 pA. **e**,**f**, Quantification of amplitude and mean open probability ($P_o$) of ASIC1a unitary currents evoked in the presence and absence of 500 µM glutamate at pH 7.0 and 7.2. $n$ = 12 and 8 cells for amplitude; $n$ = 9 and 8 cells for $P_o$. **g**, **h**, Representative MST traces and dose-response curve showing direct binding between ASIC1a and L-AP5 ($n$ = 4 replicated tests) rather than D-AP5 ($n$ = 4 replicated tests) at pH 7.0. Vertical bars: blue, cold fluorescence detected at −1 - 0 s; red, hot fluorescence detected at 4 - 5 s. Three to six replicated cultures for patch recordings were tested. Data are mean±s.e.m.; one-way ANOVA with Tukey post hoc correction for multiple comparisons (**b**); two-tailed paired Student's $t$-test (**e**,**f**); $P$ values are indicated.

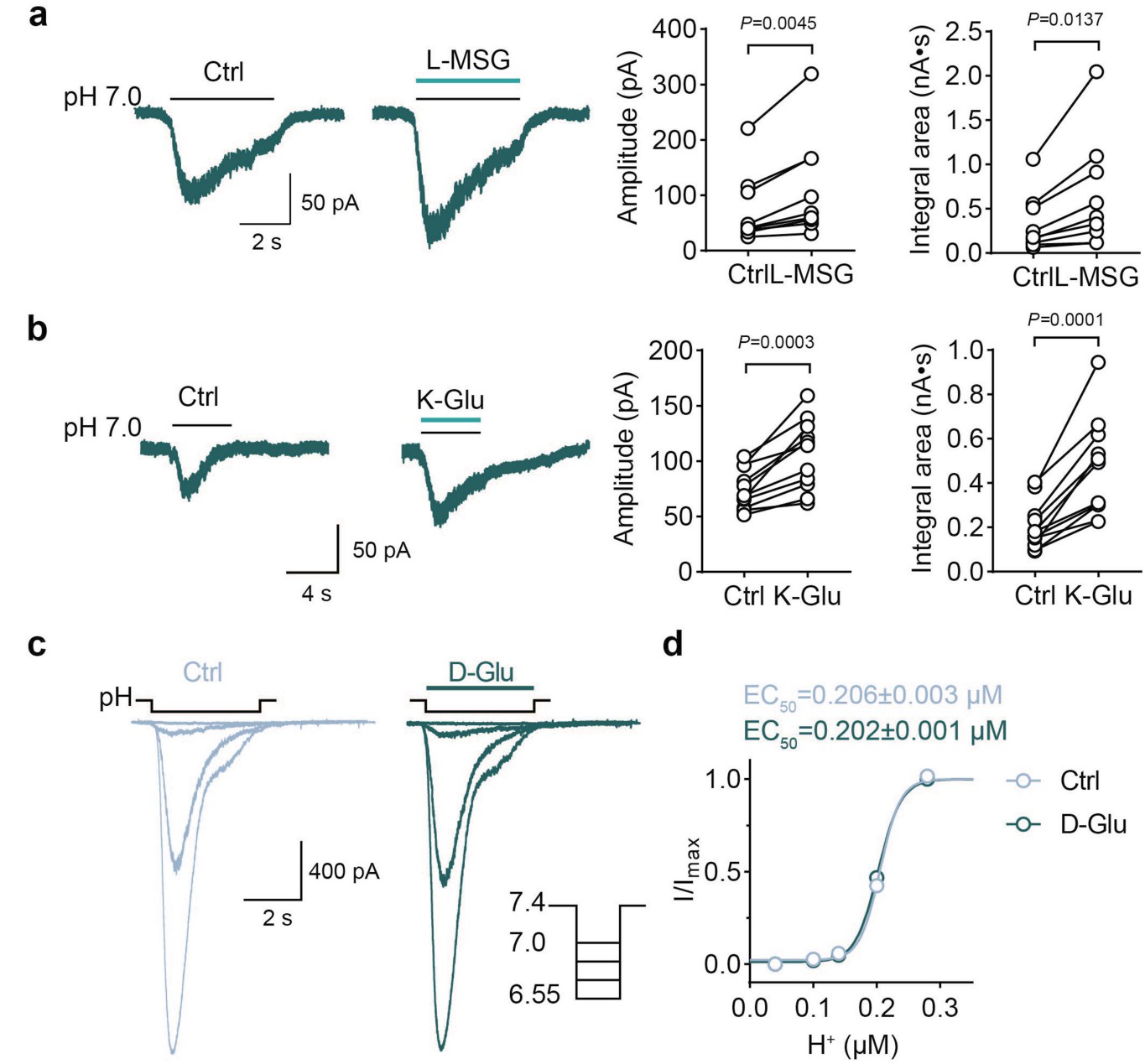

**Extended Data Fig. 3 | The effect of L-MSG, monopotassium glutamate (K-Glu) and D-glutamate on ASIC1a. a**, L-MSG potentiated the peak amplitude and integral area of $I_{ASIC1a}$ at pH 7.0. $n$ = 9 cells. **b**, K-Glu potentiate the peak amplitude and integral area of $I_{ASIC1a}$ at pH 7.0. $n$ = 11 cells. **c**, Representative traces showing the effect of D-glutamate (500 µM) on $I_{ASIC1a}$ at different pH. **d**, Dose-response curve for ASIC1a activation with or without D-glutamate. $n$ = 6 cells. Three replicated cultures for patch recordings were tested. Data are mean±s.e.m.; two-tailed paired t-test (**a**, **b**). $P$ values are indicated.

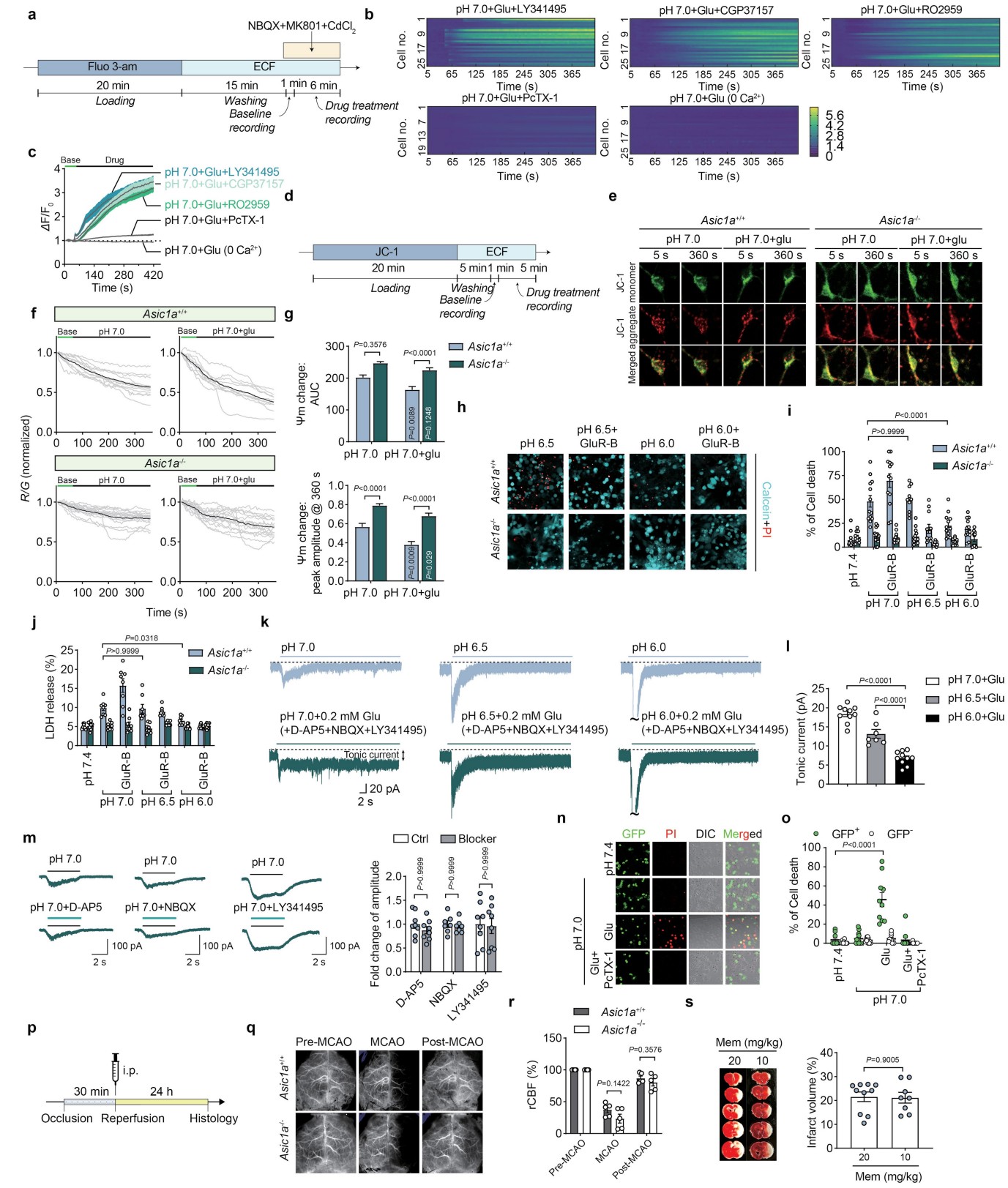

**Extended Data Fig. 4** | See next page for caption.

**Extended Data Fig. 4 | Assessment of cultured neuron and brain injury from** ***Asic1a*** **and** ***Asic1a*** **mice. a**, Schematic of the experimental protocol of $Ca^{2+}$ imaging. Cultured cortical neurons were loaded with Fluo 3-AM for 20 min followed by a 15 min washing step in ECF. After 1 min baseline recording, neurons were imaged for 6 min following pH 7.0 solution with or without glutamate in the presence of 10 μM NBQX, 1 μM MK801 and 100 μM $CdCl_2$ to block AMPA receptors, NMDA receptors and voltage-gated calcium channels, respectively. **b**, **c**, $[Ca^{2+}]_i$ changes of primary cultured cortical neurons imaged from $Asic1a^{+/+}$ mice ($n$ = 20–26 cells from 3 cultures per group). 200 μM glutamate, 10 μM LY341495 (mGluRs blockers), 10 μM CGP37157 (inhibitor of $Na^+/Ca^{2+}$ exchanger in mitochondria), 10 μM RO2959 (selective $Ca^{2+}$ release-activated $Ca^{2+}$ channel inhibitor) and 100 ng/ml PcTX-1 were applied. **d**, Schematic flow of the experimental protocol of mitochondrial potential assay. Cultured cortical neurons were loaded with JC-1 for 20 min followed by a 5 min washout. After 1 min baseline imaging, neurons were recording for 5 min following pH 7.0 solution with or without glutamate. **e**, Representative images showing the mitochondrial potential changes with different treatment from wildtype and $Asic1a^{-/-}$ mice. With mitochondrial potential depolarization, fluorescence intensity of JC-1 monomer (green) enhanced, while JC-1 aggregate (red) faded. First frame at 5 s and last frame at 360 s are shown here for comparisons. **f**, Fluorescence changes in response to different treatments imaged from $Asic1a^{+/+}$ ($n$ = 12 and 10 cells from 3 cultures per group) and $Asic1a^{-/-}$ ($n$ = 16 and 15 cells from 3 cultures per group) cultured cortical neurons. The ratio of red and green fluorescence density was normalized to their initial value. Grey lines represented responses of individual cells while black line was the mean of all cells. **g**, Summary data for the AUC (left panel) and peak response amplitude (right panel) of mitochondrial membrane potential (Ψm) changes during treatment. **h**, Representative images showing the calcein-PI staining of cultured

neurons from $Asic1a^{+/+}$ and $Asic1a^{-/-}$ mice with different treatments. GluR-B: glutamate receptor blockers consisted of 50 μM D-AP5, 10 μM NBQX and 10 μM LY341495. **i**, Summary data showing the percentage of cell death calculated by calcein (live cells) and PI (dead cells) counting with different treatments. $n$ = 10–13 images from 3 replicated cultures for each group. **j**, Summary data showing LDH release of cultured neurons from $Asic1a^{+/+}$ and $Asic1a^{-/-}$ mice with different treatments. $n$ = 8 cultures for each group. **k**, Representative traces showing $I_{ASICs}$ evoked by pH 7.0, 6.5 and 6.0 solution from primary cultured cortical neurons were recorded for 30 s with and without glutamate. 50 μM D-AP5, 10 μM NBQX and 10 μM LY341495 were applied to block NMDARs, AMPARs and mGluRs, respectively. Dashed lines: baselines. Tonic current was indicated. **l**, Summary data showing amplitude of tonic current. $n$ = 10, 7 and 10 cells from 3 replicated cultures per group. **m**, GluR-B (200 μM D-AP5, 10 μM NBQX and 10 μM LY341495) did not affect $I_{ASIC1a}$ at pH 7.0. $n$ = 8, 7, 8 cells, respectively. **n**, PI staining in ASIC1a-GFP transfected CHO cells after 1 h different treatments. 500 μM glutamate and 100 ng/ml PcTX-1 were applied. **o**, The percentage of cell death calculated by PI, $GFP^+$ and $GFP^-$ counting. $n$ = 10–13 images from 3 replicated cultures for each group. **p**, Schematic illustration of the timeline of MCAO treatment. **q**, Laser speckle imaging at 5 min pre-MCAO, 15 min post-occlusion (MCAO) and 15 min post-reperfusion (post-MCAO) in $Asic1a^{+/+}$ and $Asic1a^{-/-}$ mice. **r**, Summary data showing the relative cerebral blood flow (rCBF) changes during MCAO in $Asic1a^{+/+}$ and $Asic1a^{-/-}$ mice. $n$ = 5 mice per group. **s**, Representative images and summary data showing infarct volume after MCAO from wildtype mice injected with NMDAR blocker, memantine (i.p.). $n$ = 10, 8 mice. Three replicated cultures for patch recordings were tested. Data are mean±s.e.m.; one-way (**l**) and two-way (**g**,**i**,**j**,**m**,**o**,**r**) ANOVA with Tukey post hoc correction for multiple comparisons; two-tailed unpaired t-test (**s**). $P$ values are indicated.

# a

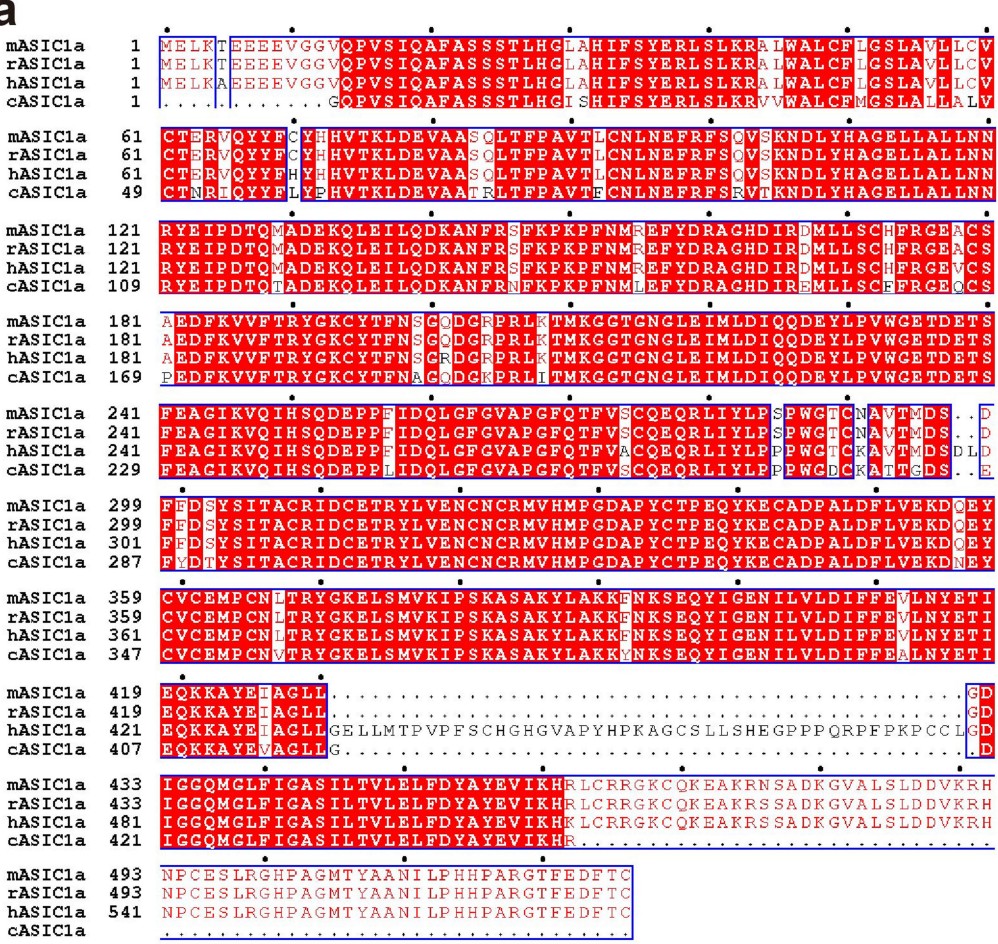

# b

| Species | Identity |
|---|---|
| *Mus musculus* | 100% |
| *Rattus norvegicus* | 99% |
| *Homo sapiens* | 90% |
| *Gallus gallus* | 90% |

**Extended Data Fig. 5 | Alignment of ASIC1a among different species shows highly conserved sequences in *Aves* and *Mammalia*. a**, Sequence alignment of the *m*ASIC1a (mouse), *r*ASIC1a (rat), *h*ASIC1a (human) and *c*ASIC1a (chicken). Homologous regions are coloured red background. **b**, Table showing percentage of ASIC1a amino acid sequence identity among different species.

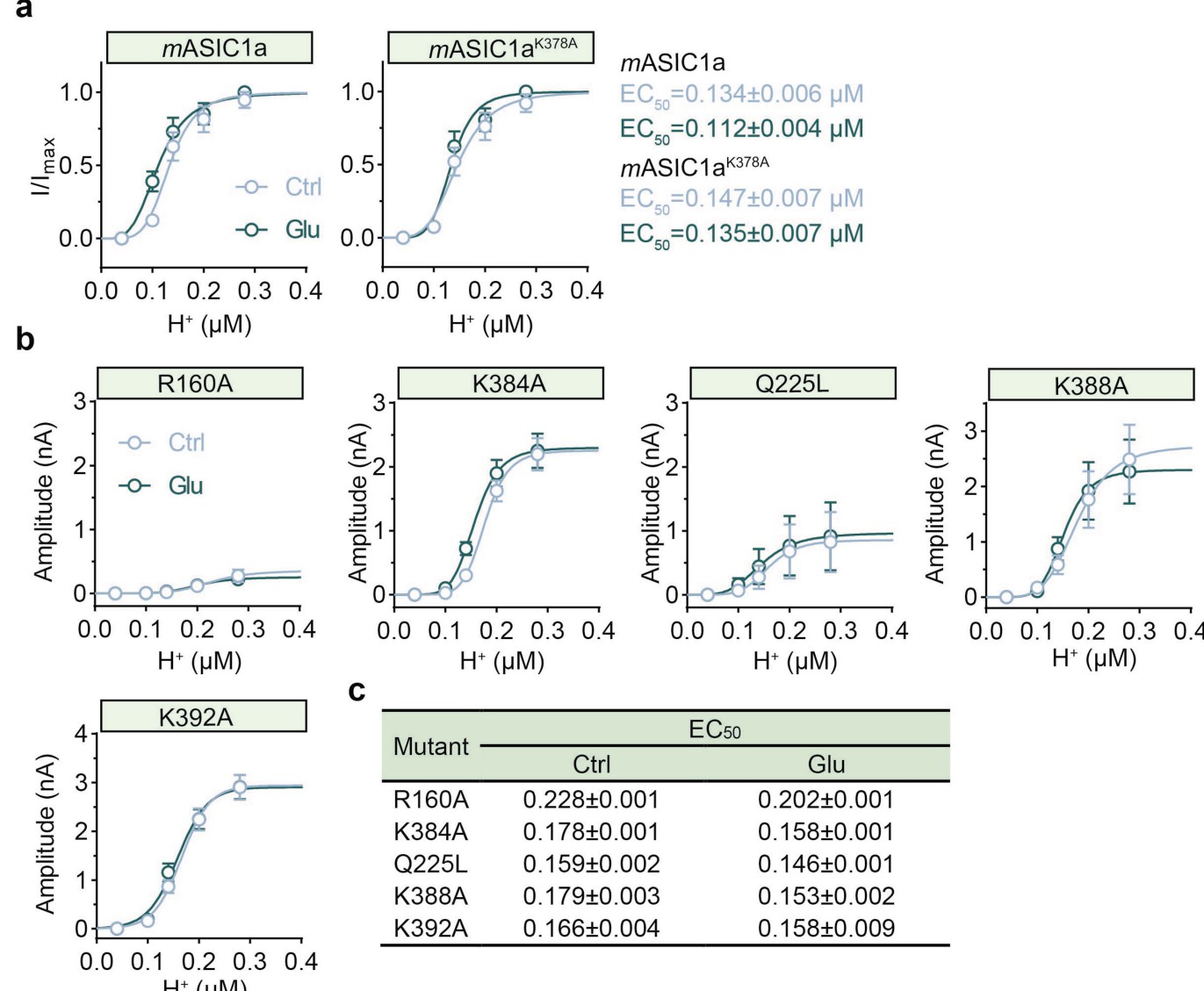

**Extended Data Fig. 6 | Screening glutamate-binding sites by site-directed mutagenesis. a**, Dose-response curves showing the effects of 500 μM glutamate on *m*ASIC1a (*n* = 5 cells) and *m*ASIC1a$^{K378A}$ (*n* = 6 cells) currents in CHO cells. **b**, Dose-response curves showing the effects of 500 μM glutamate on I$_{ASICs}$ in R160A (*n* = 5 cells), K384A (*n* = 7 cells), Q225L (*n* = 3 cells), K388A (*n* = 9 cells) and K392A (*n* = 15 cells) mutants of *h*ASIC1a transfected CHO cells.

Glutamate enhances I$_{ASICs}$ in K384A, Q225L and K388A mutants transfected CHO cells but not in R160A and K392A mutants. However, I$_{ASICs}$ in R160A mutant significantly decreased in amplitude when compared to that by *m*ASIC1a, making it unlikely to be a glutamate-binding site. **c**, Table showing EC$_{50}$ for each mutant in (**b**). Three replicated cultures for patch recordings were tested. Data are mean±s.e.m.

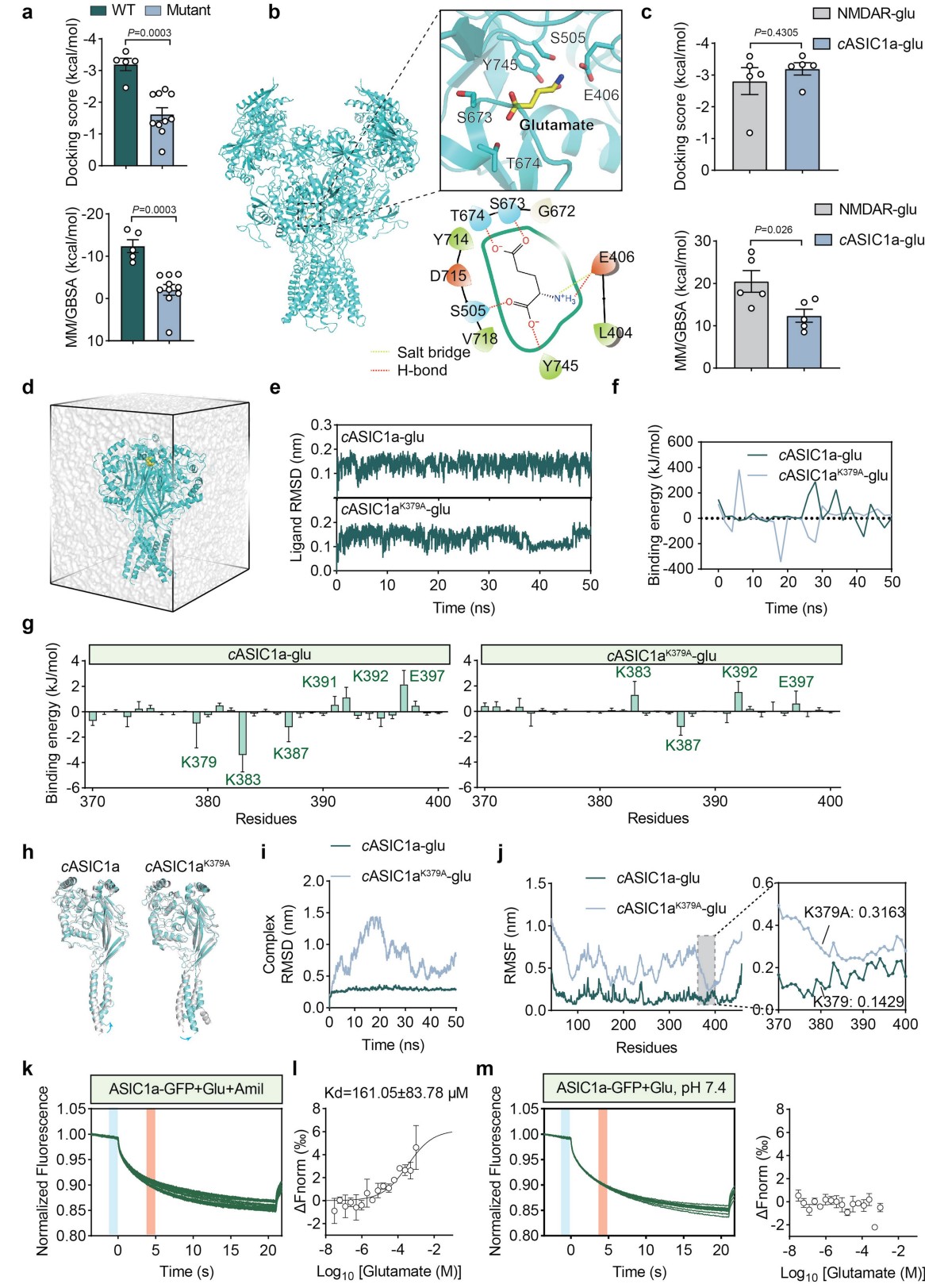

**Extended Data Fig. 7** | See next page for caption.

**Extended Data Fig. 7 | In silico and MST analyses validate the glutamate binding pocket on ASIC1a. a**, Plots of docking scores and MM/GBSA values of glutamate binding to cASIC1a. Mutant indicates K379A. $n$ = 5 and 10 poses, respectively. **b**, Overall structure of glutamate-NMDAR complex (PDB: 5IOU) and a close-up view of glutamate-binding pocket. The interactions between glutamate and surrounding residues are shown as yellow (salt bridge) and red (hydrogen-bond) dash lines. **c**, Summary data by computational calculation showing docking scores and MM/GBSA of glutamate binding to NMDAR and cASIC1a. $n$ = 5 and 5 poses, respectively. **d**, A snapshot of the simulation box of the glutamate-cASIC1a complex in 150 mM NaCl solution. Glutamate is shown as yellow spheres, cASIC1a is shown as cyan cartoon, water is shown as transparent surface. For clarity, ions are omitted. **e**, Structural stability of ligand in wildtype and mutant conformations was measured as the RMSD (unit: nm) over a 50-ns time course. **f**, Binding energy for glutamate-cASIC1a complex was calculated by the MM-PBSA method. **g**, Binding energy of glutamate-cASIC1a complex for amino acid residues from 370 to 400 over a 50-ns time course. Several residues with high binding energy were labelled.

**h**, Conformations of wildtype and mutant cASIC1a before (white) and after (cyan) 50 ns MD simulation. Structures were aligned. For clarity, only chain A is shown here. Blue arrows indicated the direction of conformational change. **i**, RMSD (unit: nm) of glutamate-cASIC1a complex in wildtype and mutant conformations over a 50-ns time course. **j**, RMSF (unit: nm) of glutamate-cASIC1a complex in wildtype and mutant conformations throughout chain A residues. Zoom-in view of RMSF from residues 370 to 400 is shown in the right panel. **k**, **l**, Representative MST traces and dose-response curve showing direct binding between glutamate and GFP-tagged ASIC1a in the presence of 100 μM amiloride, a pore blocker of ASIC1a. Vertical bars: blue, cold fluorescence detected at −1 - 0 s; red, hot fluorescence detected at 4 - 5 s. $n$ = 3 replicated tests. **m**, Representative MST traces and dose-response curve showing no direct binding between ASIC1a and glutamate at pH 7.4 ($n$ = 3 replicated tests). Vertical bars: blue detected at −1 - 0 s, cold fluorescence; red, hot fluorescence detected at 4 - 5 s. Data are mean±s.e.m.; two-tailed unpaired $t$-test (**a**,**c**). $P$ values are indicated.

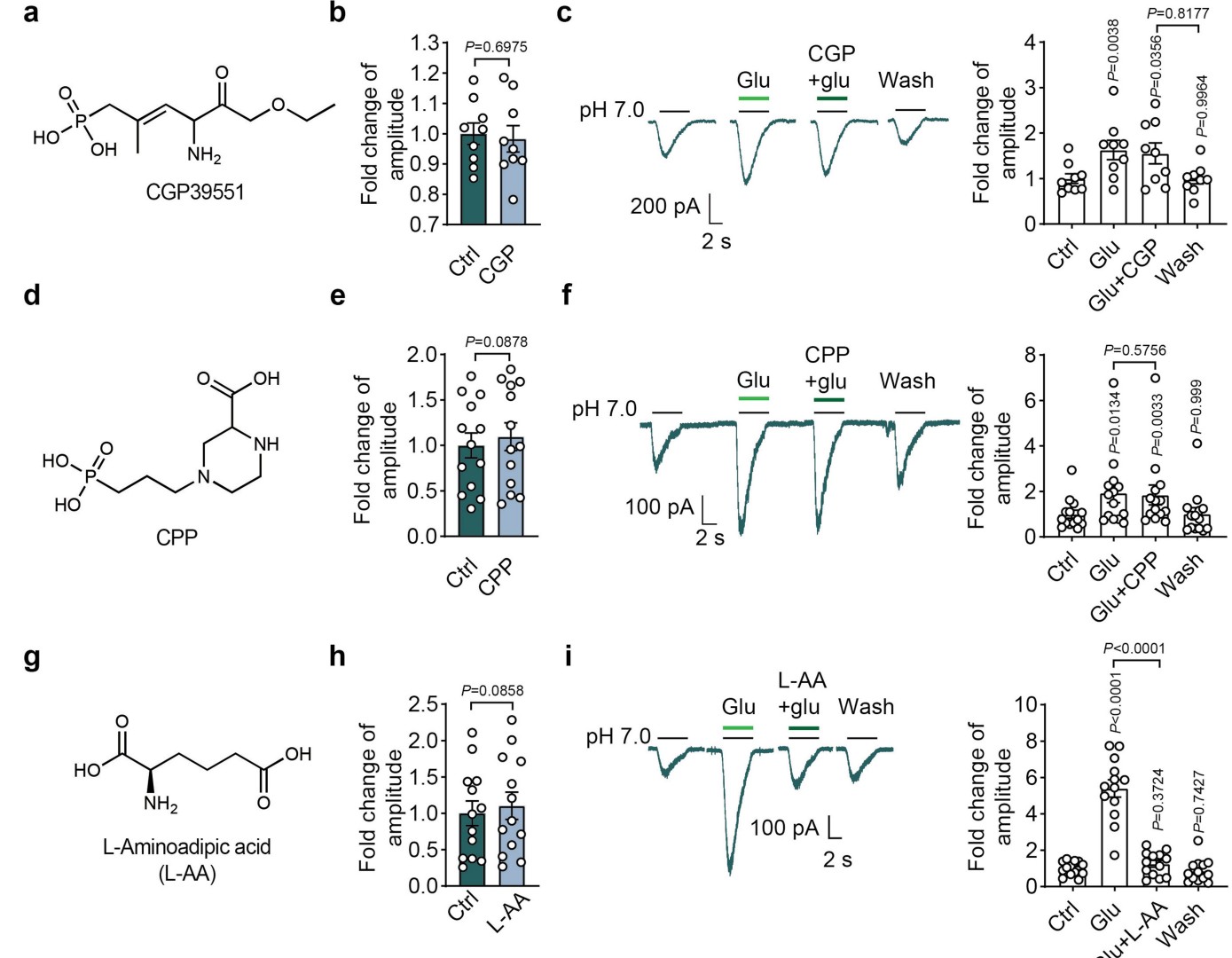

**Extended Data Fig. 8 | Pharmacological effects of glutamate receptor antagonists and agonist on I$_{ASICs}$ in CHO cells. a**, **d**, **g**, Chemical structure of CGP39551, CPP and L-Aminoadipic acid (L-AA). **b**, **c**, **e**, **f**, 100 μM CGP39551 and CPP, competitive antagonists of NMDA receptors, had no effect on I$_{ASICs}$ and could not block glutamate-induced potentiation of I$_{ASICs}$. $n$ = 9 cells (**b**), 9 cells (**c**), 13 cells (**e**) and 14 cells (**f**). **h**, **i**, 100 μM L-AA, an agonist for NMDAR and metabotropic glutamate receptors (mGluRs), had no effect on I$_{ASICs}$, however, could eliminate glutamate-induced potentiation of I$_{ASICs}$. $n$ = 13 cells (**h**) and 13 cells (**i**). Three to six replicated cultures for patch recordings were tested. Data are mean±s.e.m.; two-tailed paired $t$-test (**b**, **e**, **h**); one-way ANOVA with Tukey post hoc correction (**c**, **f**, **i**). $P$ values are indicated.

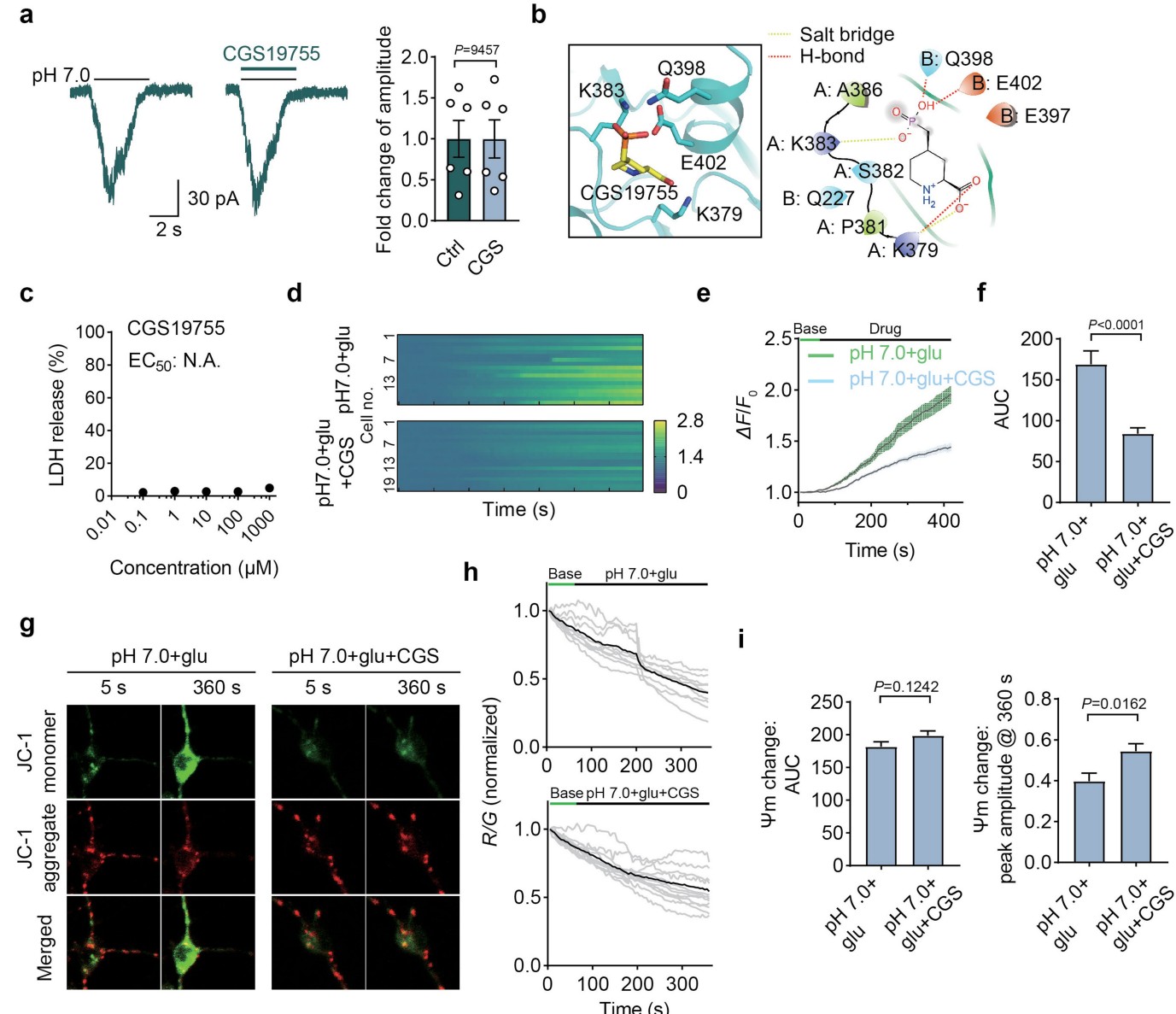

**Extended Data Fig. 9 | CGS19755 acts as a promising neuroprotectant against cell injury in vitro. a**, CGS19755 (100 μM), a competitive antagonists of NMDA receptors, had no effect on $I_{ASICs}$ at pH 7.0. $n$ = 6 cells. **b**, Three-dimensional and two-dimensional images showing the glutamate binding pocket shared by CGS19755. The interactions between CGS19755 and surrounding residues are shown as yellow (salt bridge) and red (hydrogen-bond) dash lines. **c**, Determination of CGS19755 for basal cell death (1 h of exposure to CGS19755) in primary cortical neurons by LDH release assay. $n$ = 5 wells for each concentration. N.A., not applicable. **d**, **e**, Calcium imaging of cultured cortical neurons recorded from wildtype mice with treatment of 500 μM glutamate alone or in combination with 100 μM CGS19755 in pH 7.0 solution. **f**, Quantification of AUC from Ca²⁺ imaging under two treatment conditions.

$n$ = 18 and 20 cells for each group. **g**, **h**, Representative images and traces showing changes of the mitochondrial potential of individual cells (grey) and their means (black) with different treatments of neurons from wildtype mice. When mitochondrial potential (absolute value) decreasing, fluorescence intensity of JC-1 monomer (green) enhanced, while JC-1 aggregate (red) faded. First frame at 5 s and last frame at 360 s are shown here (**g**). **i**, Quantifications of AUC of drug treatment (left panel) and peak amplitude (right panel) showing changes in mitochondrial membrane potential (Ψm) in pH 7.0 solution with glutamate alone or in combination with CGS19755. $n$ = 9 and 13 cells for each group. Three replicated cultures for patch recordings were tested. Data are mean±s.e.m.; two-tailed paired $t$-test (**a**); two-tailed unpaired $t$-test (**f**, **i**). $P$ values are indicated.

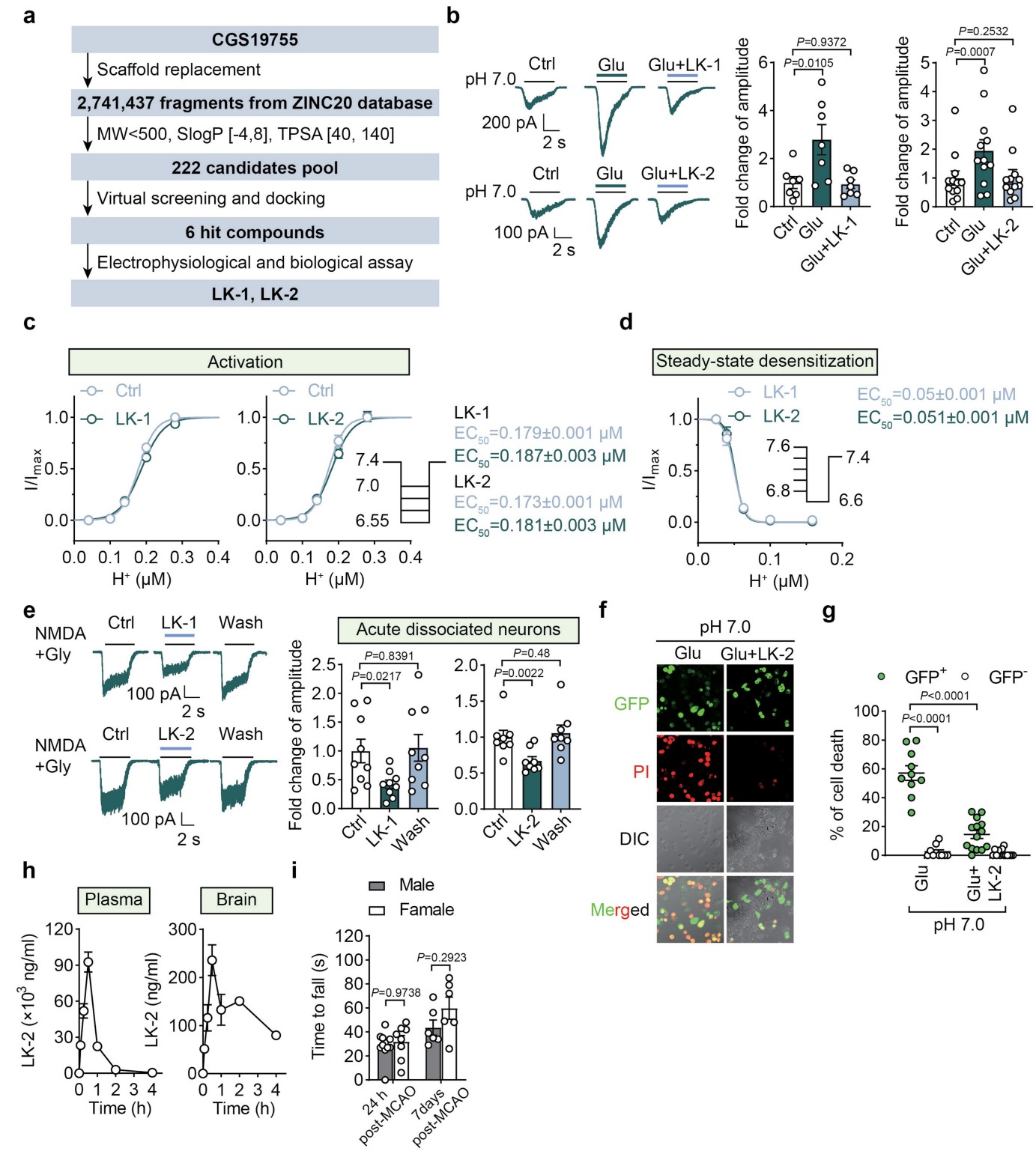

**Extended Data Fig. 10 | In silico screening identifies LK-2 over LK-1 with more favourable pharmacological and pharmacokinetics profiles as a candidate for stroke therapy. a**, Flow chart showing strategy for computational virtual screening of LK-1 and LK-2 based on the backbone structure of CGS19755. **b**, Glutamate-dependent enhancement of $I_{ASICs}$ was inhibited by 100 μM LK-1 ($n$ = 7 cells) or 100 μM LK-2 ($n$ = 12 cells) in ASIC1a transfected CHO cells. **c**, **d**, 10 μM LK-1 and 10 μM LK-2 did not affect activation ($n$ = 7 and 7 cells) and desensitization ($n$ = 6 and 7 cells) of $I_{ASICs}$ in ASIC1a transfected CHO cells. **e**, 100 μM LK-1 ($n$ = 9 cells) or 100 μM LK-2 ($n$ = 8 cells) inhibited NMDAR currents in acute isolated cortical neurons. 100 μM NMDA and 1 μM glycine were applied. **f**, PI staining in ASIC1a-GFP transfected CHO cells after 1 h glutamate (500 μM)

treatment with or without 10 μM LK-2. **g**, The percentage of cell death calculated by PI, GFP$^+$ and GFP$^-$ counting. $n$ = 10–14 images from 3 replicated cultures for each group. **h**, Pharmacokinetic test in vivo showing the concentration of LK-2 in plasma and brain detected by LC-MS/MS. $n$ = 4 samples for each time point. **i**, Quantification of time to fall from rod in rotarod tests in LK-2 injected male and female mice (24 h post-MCAO, $n$ = 11 and 8 mice for male and female mice; 7 days post-MCAO, $n$ = 6, and 6 mice for male and female mice). Three to six replicated cultures for patch recordings were tested. Data are mean±s.e.m.; two-tailed paired $t$-test (**c**); one-way (**b**,**e**) and two-way (**g**,**i**) ANOVA with Tukey post hoc correction. $P$ values are indicated.

# Reporting Summary

## Statistics

For all statistical analyses, confirm that the following items are present in the figure legend, table legend, main text, or Methods section.

| n/a | Confirmed | |
|---|---|---|
| ☐ | ☒ | The exact sample size (*n*) for each experimental group/condition, given as a discrete number and unit of measurement |
| ☐ | ☒ | A statement on whether measurements were taken from distinct samples or whether the same sample was measured repeatedly |
| ☐ | ☒ | The statistical test(s) used AND whether they are one- or two-sided *Only common tests should be described solely by name; describe more complex techniques in the Methods section.* |
| ☒ | ☐ | A description of all covariates tested |
| ☐ | ☒ | A description of any assumptions or corrections, such as tests of normality and adjustment for multiple comparisons |
| ☐ | ☒ | A full description of the statistical parameters including central tendency (e.g. means) or other basic estimates (e.g. regression coefficient) AND variation (e.g. standard deviation) or associated estimates of uncertainty (e.g. confidence intervals) |
| ☐ | ☒ | For null hypothesis testing, the test statistic (e.g. *F*, *t*, *r*) with confidence intervals, effect sizes, degrees of freedom and *P* value noted *Give P values as exact values whenever suitable.* |
| ☒ | ☐ | For Bayesian analysis, information on the choice of priors and Markov chain Monte Carlo settings |
| ☒ | ☐ | For hierarchical and complex designs, identification of the appropriate level for tests and full reporting of outcomes |
| ☒ | ☐ | Estimates of effect sizes (e.g. Cohen's *d*, Pearson's *r*), indicating how they were calculated |

*Our web collection on statistics for biologists contains articles on many of the points above.*

## Software and code

Policy information about availability of computer code

Data collection    Electrophysiology was acquired using HEKA Patchmaster software (v2x90.5); Fluorescent images were acquired using Zen Black (Zeiss). Optical density was acquired using Gen5 v3.05(BioTek); Laser speckle imagings were acquired using mFLPI2Meas v2.0 software (Moor). MST traces were acquired using MO. Control software (Ver.1.6.1).

Data analysis    Analysis of electrophysiology data was performed using Clampfit (v10.5). Images were analyzed by Zen Blue (Zeiss) and ImageJ (v1.51). Statistical tests were performed by Prism 8. Pymol (v2.5) were used for structural model determination, refinement. Maestro (v12.5) and HADDOCK (v2.4) were used for molecular docking. ChemBioDraw Ultra (v14.0) was used for chemical structures. MOE software (v2015.10) and Maestro (v12.5) were used for compounds screening. Molecular dynamic (MD) simulation and analysis of its results were performed by Gromacs (v2020.3) and VMD (v1.9.4). Binding pocket analysis was performed by Fpocket 2.0 software. MM-PBSA calculations on MD simulation trajectories were performed with a modified gmx_mmpbsa bash script (available at Github). MST data were analyzed by MO. Affinity analysis software (Ver.2.3) of NanoTemper. Animal behavior data were collected and analyzed by Smart software (v3.0.06) of Panlab.

For manuscripts utilizing custom algorithms or software that are central to the research but not yet described in published literature, software must be made available to editors and reviewers. We strongly encourage code deposition in a community repository (e.g. GitHub). See the Nature Portfolio guidelines for submitting code & software for further information.

## Data

Policy information about availability of data

All manuscripts must include a data availability statement. This statement should provide the following information, where applicable:
- Accession codes, unique identifiers, or web links for publicly available datasets
- A description of any restrictions on data availability
- For clinical datasets or third party data, please ensure that the statement adheres to our policy

Source data are provided with this paper.

# Field-specific reporting

Please select the one below that is the best fit for your research. If you are not sure, read the appropriate sections before making your selection.

☒ Life sciences          ☐ Behavioural & social sciences          ☐ Ecological, evolutionary & environmental sciences

For a reference copy of the document with all sections, see nature.com/documents/nr-reporting-summary-flat.pdf

# Life sciences study design

All studies must disclose on these points even when the disclosure is negative.

| | |
|---|---|
| Sample size | Sample sizes were empirically determined to match or exceed typical sample sizes reported in previous studies on the topic (Xiong et al, 2004, Cell; Gao et al, 2005, Neuron). |
| Data exclusions | No data was excluded from analysis. |
| Replication | Electrophysiology data for ASIC current recordings in mice was acquired from multiple neurons in at least 3 animals of each genotype. For calcium imaging and mitochondrial membrane potential test, neurons were pooled from at least 5 mice of each genotype. For cell death experiments, data were acquired from at least 3 independent cultures for calcein-PI staining and LDH release test. For MST binding assay, at least 3 independent tests were performed. For MCAO and animal behavior studies, more than 10 animals were used for each group except the sham group (only 3 mice were used) due to the '3R' principle of animal studies. All attempts at replication were successful. |
| Randomization | Experiments were not randomized. |
| Blinding | The screening of glutamate binding site on ASIC1a and electrophysiology recordings were blind to researchers until mutations of binding residues resulted in ablation of the effect that could be readily observed by electrophysiology recordings. The biological tests of synthesized compounds in MCAO model and behavior experiments were blind to the experimenter. No other blinding was performed. |

# Reporting for specific materials, systems and methods

We require information from authors about some types of materials, experimental systems and methods used in many studies. Here, indicate whether each material, system or method listed is relevant to your study. If you are not sure if a list item applies to your research, read the appropriate section before selecting a response.

### Materials & experimental systems

| n/a | Involved in the study |
|---|---|
| ☒ | ☐ Antibodies |
| ☐ | ☒ Eukaryotic cell lines |
| ☒ | ☐ Palaeontology and archaeology |
| ☐ | ☒ Animals and other organisms |
| ☒ | ☐ Human research participants |
| ☒ | ☐ Clinical data |
| ☒ | ☐ Dual use research of concern |

### Methods

| n/a | Involved in the study |
|---|---|
| ☒ | ☐ ChIP-seq |
| ☒ | ☐ Flow cytometry |
| ☒ | ☐ MRI-based neuroimaging |

## Eukaryotic cell lines

Policy information about cell lines

| | |
|---|---|
| Cell line source(s) | The CHO K1 cells and HEK293T cells were provided by Stem Cell Bank, Chinese Academy of Sciences. |
| Authentication | The cells were routinely maintained in our laboratory. They were not authenticated for these studies. |

| | |
|---|---|
| Mycoplasma contamination | The cell lines were tested negative for mycoplasma contamination. |
| Commonly misidentified lines (See ICLAC register) | No commonly misidentified cell lines were used. |

# Animals and other organisms

Policy information about studies involving animals; ARRIVE guidelines recommended for reporting animal research

| | |
|---|---|
| Laboratory animals | Homozygous Asic1a knockout mice were originally generated from Jackson Laboratory, wild-type mice of the same background (C57BL/6j) were purchased from Jiesijie company (China) and housed in our laboratory. |
| Wild animals | This study did not involve wild animals. |
| Field-collected samples | This study did not involve samples collected from the field. |
| Ethics oversight | All experimental protocols and animal use were approved by the Ethics Committee of Shanghai Sixth People's Hospital. |

Note that full information on the approval of the study protocol must also be provided in the manuscript.

