## [Peer Review File · Nature]

Manuscript Title: Glutamate acts on acid-sensing ion channels to worsen ischemic brain damage

Reviewer Comments & Author Rebuttals

Reviewer Reports on the Initial Version:

Referees' comments:

Referee #1 (Remarks to the Author):

In this study, the authors made a very novel discovery: glutamate and its structural analogs, including NMDAR antagonist AP5, potentiated currents mediated by acid-sensing ion channels (ASICs). Site-directed mutagenesis and structure-based *in silico* molecular docking and simulations uncovered a glutamate binding cavity in the extracellular domain of ASIC1a. Computational drug screening of NMDAR competitive antagonist analogs identified a small molecule, LK-2, that binds to this cavity and abolishes glutamate-dependent potentiation of ASIC currents but largely spares NMDARs, providing clear neuroprotection *in vitro* and *in vivo*. The authors conclude that glutamate serves as a co-agonist for ASIC1a to exacerbate neurotoxicity, and that selective blockage of glutamate binding sites on ASICs without affecting NMDARs may be of strategic importance for developing effective stroke therapeutics devoid of the side effects of NMDAR antagonists. The findings are original and represent immediate interest to the audience in field of ASICs and stroke. Having said that, more scrutinized analysis and stringent data need to be provided for injury experiments to clearly separate the contribution of glutamate receptors per se and that of ASICs potentiated by glutamate.

1. P3 line 69-70: "We found that glutamate, but not glutamic acid monosodium, potently potentiated IASICs". Source and purity of glutamate should be described and some explanation of why glutamate but not monosodium glutamate potentiates ASICs. Wouldn't we expect to have glutamate when monosodium glutamate is dissolved in the solution?
2. Figure 2f and 4f. Baseline cell death over 60% (e.g., at 7.4) is unacceptably high. With this high baseline cell death, the system is unreliable for detecting small/moderate changes of cell death. In contrast, small baseline LDH release was detected but the changes in LDH release after LK-1 and LK-2 were marginal (e.g., 16% vs. 13%, Figure 4g), even though statistical analysis showed high significance.
3. On the other hand, in Figure 2g, glutamate appears to potentiate LDH release by 50% in ASIC1a^{-/-} cells (9% to 14%?) yet the difference was not significant. Therefore, more stringent data and scrutinized analysis are required in cell culture experiments to increase the confidence in these results.
4. Figure 2i: Memantine does not show any effect on infarct volume in ASIC^{+/+} mice. This is contradictory to the well-known contribution of glutamate toxicity to stroke damage. How is this finding compared to those of others who studied the effects of glutamate antagonists in stroke

models? At what time was memantine applied?

5. P4, line 86-87: "Thus, glutamate appears to act as a co-agonist to allosterically potentiate ASICs, reminiscent of glycine as a co-agonist for NMDARs". This description is not accurate since NMDAR does not function without the presence of a co-agonist but ASIC1a does not require a co-agonist to open.

6. P4, line 93-94: "reversal potential (control: 33.15 mV; glutamate: 28.56 mV) (Extended Data Fig. 1h, i)". Were these values expected for ASICs? In most whole-cell recordings, the reversal potential for ASICs is close to +60 mV (near the sodium equilibrium potential). The authors need to provide some explanation for the different reversal potentials they recorded.

7. P6, line 137-139: "We found that glutamate markedly increased the percentage of cell death in *Asic1a*^{+/+} slices, whilst in contrast, *Asic1a*^{-/-} slices had decreased cell death and seemed resistant to glutamate insults (Fig. 2e,f)". Since glutamate can cause cell death through NMDA receptors, these experiments should be repeated in the presence of glutamate antagonist that does not block the binding site on ASIC1a but abolish the neuronal injury through NMDA receptors directly (e.g., MK801).

8. P7, line 160-162: "Together, these data indicated that ischemic brain damage in an MCAO mouse model was NMDAR-independent and largely ASIC1a dependent". Again, this finding is inconsistent with previous studies that showed neuroprotective effect of glutamate antagonists including those that do not interact with the glutamate binding site, e.g., MK801.

9. P11-12, line 299-302: "in cell death and LDH release assays under the acid condition with glutamate treatment in vitro, LK-2 remarkably reduced cell death and LDH release to an extent close to CGS19755, indicating glutamate-enhanced ASIC activity rather than NMDAR overactivation was the main contributor to cell injury in stroke conditions (Fig. 4e-g)". There are three issues associated with the data presented in Fig 4e-9. One: the reduction in cell death or LDH release by LK-2 is not remarkable (~10-15% in cell live/dead counting or LDH release). Two: the concentration used for LK-1 and LK-2 (100 μ M) may not differentiate between the direct effect on NMDA reporters and indirect effect on ASICs. Three: MK801, an open channel blocker of NMDA receptor that does not interact with the glutamate binding site, shows a better or comparable effect in reducing LDH release compared with that of LK-1, LK-2, and CGS, arguing strongly against the claim that glutamate-enhanced ASIC activity rather than NMDAR overactivation was the main contributor to cell injury in stroke conditions!

10. Extended data Figure 1g: the representative traces do not show appreciable potentiation of ASIC current by glutamate and the dose response curves do not show a significant shift/separation.

Referee #2 (Remarks to the Author):

Lai and colleagues have presented an interesting and novel manuscript where they suggest that glutamate is a positive allosteric modulator of acid sensitive ion channel 1a with relevance to neuronal death during stroke. They use a breadth of techniques (electrophysiology, mutagenesis, molecular dynamics simulations and modelling). Perhaps most impactful is the discovery and evaluation of a new small molecule therapeutic, LK-2. In brief, they show that glutamate augments ASIC1a currents at acidic pH, but not at physiological pH, which is likely highly relevant to stroke. They demonstrate that relevant EC50 for glutamate and acidosis that occurs during stroke. It is demonstrated that glutamate enhances neuronal death at pH 7 in vitro and in vivo and describe the contribution of ASIC1a-mediated calcium influx to mitochondrial dysfunction. Using in silico molecular docking and site-directed mutagenesis, they identify the apparent binding pocket for glutamate and several NMDAR antagonists. The latter are used to rationally design an inhibitor with minimal activity at NMDARs and demonstrate neuroprotection with this compound. Statistical methods and experimental power are appropriately applied.

Overall the conclusions are important and likely to be of importance to a broad range of neuroscience. Furthermore, the discovery of LK-2 as a starting point for a new class of putative neuroprotective drugs could greatly impact clinical stroke care. There are however some inconsistencies in the data as well as a conceptual issue that raise concerns:

- 1) Conceptually, I'm having difficulty understanding how ASIC1a channels make such a large contribution to Ca²⁺ overload. The channels inactivate rapidly as shown by others and in the whole-cell recordings presented in this manuscript. The channels are closed within about 1s and yet the Ca²⁺ dysregulation persists for minutes (figure 2). Are the channels causing release from stores? For example, the loss of mitochondrial membrane potential will likely cause these organelles to release Ca²⁺
- 2) The title, Pg 4 line 86, and line 196 it is stated that glutamate is a co-agonist for ASIC1a. This idea is compared to glycine for NMDARs. ASIC1a channels open in the absence of glutamate and glutamate alone does not cause channel gating. This is more consistent with a role for glutamate as a positive allosteric modulator, not a co-agonist. This does not detract from the impact of the discovery, but should be corrected for consistency.
- 3) It is reported (line 69-72) that glutamate but not glutamic acid monosodium is a PAM at ASIC1a. This seems impossible to me. Glutamate and glutamic acid have the same structure and both are going to be protonated in the pH ranges used in the study. The only clear difference is the addition of Na⁺. Is Na⁺ inhibiting the binding of glutamate to ASIC1a? This could be evaluated with addition of equimolar NaCl to the glutamate condition. If not, why are these molecules behaving differently?
- 4) Figure 1b, what pH was the dose-response curve determined at? Is the dose response the same at different pH levels? It could be predicted that pH increases affinity of glutamate for the channels to more negatively influence cell health.
- 5) Part of understanding the interaction of glutamate with the channels extracellular domains was determining if glutamate interacts with the known Ca²⁺ binding site. However, the potentiation of hASIC1aE427G/D434C by glutamate, as suggested in Extended Fig 1g, is not convincing. The leftward shift in the curve is modest at best. How was potentiation in these conditions statistically supported? It does appear that mutagenesis of the Ca²⁺ binding site may be important in some way for glutamate binding, whether it is a general structural modification that influences the glutamate pocket/.

- 6) D-APV will potently inhibit NMDARs but not block the glutamate potentiation of ASIC. Memantine and MK801 require channel opening and could allow for significant calcium influx prior to block. Further, NMDARs are known to enhance ASIC1a via nitrosylation (Gao, Neuron 2005), it is possible that this Ca²⁺-dependent cross talk is occurring prior to block by MK801. Key experiments (i.e. cell death assay, mitochondrial assay) should be done in the presence of D-APV instead of MK801 to completely rule out contributions of NMDARs?
- 7) Calcein / PI staining. How was % cell death determined? The basal rate of (apparent) cell death is ~50% in the slices. This seems very high and raises concerns about overall health of the slices or that images were collected too close to the cut surface of the slice. At what depth were the cells imaged?
- 8) It is very surprising that memantine showed no significant neuroprotection in the MCAO model. Numerous groups have shown potent protection by memantine. I see that body temperature was maintained at 33C, which is quite hypothermic and could account for the apparent lack of contribution by NMDARs.
- 9) Were MD simulations done using a pH7 or lower model? This is important because there is little to no potentiation by glutamate at pH7.4.
- 10) Line 240. Not clear why memantine was evaluated for blocking glutamate effects on ASIC1a. memantine is a non-competitive, pore blocker and does not interact with the glutamate / APV sites on NMDARs.
- 11) Test LK2 in the ASIC1a k.o animals in addition to selfotel. This could help address some potential off target effects.
- 12) Importantly, the effective concentration of LK-2 in the brain appears too low to have a significant effect on ASIC1a. The authors report a brain level of 235 ug/L. The formula weight of LK-2 (C₁₂H₁₇N₂O₅P) is 300.25 (this should be reported in the paper), which is a concentration of ~0.7uM. This would appear to inhibit the channels by about 10-20%. The neuroprotective effect in vivo is at odds with this.
- 13) Reductions in lesion size are an interesting measure of neuroprotection, but do not always relate directly to behavioural protection or recovery. The LK-2 drug should be evaluated in behavioural assays relevant to stroke.

Referee #3 (Remarks to the Author):

This manuscript describes a novel potentiation of ASIC1a proton-gated currents by glutamate. Specifically, the authors show pH 7.0 currents potentiated in whole cell and single channel configurations and enhanced ASIC1a-dependent changes in calcium and mitochondrial dysfunction. The authors confirm previous work that ASIC1a is involved in cell death in cultured slices and show potentiation of cell death with glutamate that is limited in ASIC1a knockout animals. The authors identify an area of ASIC1a that, when mutated appears to lower glutamate-induced potentiation. They also identify compounds which interact with this site that can apparently inhibit glutamate-induced potentiation of ASIC1a and (for some) the NMDA receptor and show these molecules limit MCAO-mediated death. The authors provide a compelling and linear interpretation of their data. The idea that there is a glutamate binding site in ASIC1a that potentiates currents is exciting. Further, the discovery of LK-2 as an inhibitor of this phenomenon and the initial effect of this compound in the MCAO model is extremely promising. However, numerous weaknesses related to interpretation of

data, the limited characterization of the effect and compounds and mutant channels, and a disconnection between the written results section and data included (especially within the extended data suggestion) reduce enthusiasm for the manuscript.

There have been multiple extracellular modulators or “co-agonists” reported for ASIC function such as lactate, spermine, calcium, zinc, peptides, ammonium, etc. Although the shift in the p_{H50} for ASIC1a activation with glutamate is not reported, it appears that glutamate acts like many of these compounds to shift the apparent proton response for ASIC1a activation. The authors imply that this is a new mechanism of modulation, and state that it is independent of calcium binding. This is important as several exciting reports of gating modulators and alternate agonists/co-agonists were later found to affect the apparent proton sensitivity of ASIC1a by altering the availability zinc or calcium... or the channels response to calcium and, thus, indirectly affect channel gating. The involvement of calcium and zinc (which are likely present in the recording solutions) was not investigated through the use of chelators or ion substitution. The authors do investigate glutamate response in a mutant channel likely insensitive to calcium (with altered expression and proton response curve) and state that glutamate still potentiates the current. However, this potentiation is NOT evident in the extended data (Figure 1g) as stated in the results section. In fact, it appears that this channel is unaffected by glutamate in the proton concentration response curve. This is significant as it speaks to the mechanism of action of glutamate, it’s novelty, and the mechanism of action of the newly identified compounds.

ASIC1 knockout animals have been shown to have altered signaling cascades and CamKII levels. The authors rely on the sole use of these animals to determine ASIC-specific effects for cell death and mitochondrial function . The use of ASIC1a-dependent inhibitors (such as PcTx1 and other venom peptides) was not tested in most experiments and models.

The ASIC1 KO slices showed an increase in LDH release with pH 7.0 and glutamate (Figure 2G) suggesting there are ASIC1a-independent effects. This result is minimized in the text of the results.

Binding of glutamate to the channel is not shown, but inferred through modeling and the reduced effect on a mutant channel that also appears to have an alteration to the pH dependence of activation (which is not investigated or characterized). Further, solid measurements of the concentration response of glutamate in these models is sorely needed to better ensure that a reduction of effect is due to a reduction in binding (and not secondary to an effect on channel gating).

The manuscript in multiple places states that glutamate causes NMDAR-independent neurotoxicity. Previously reports (Neuron 2005) show that activation of NMDA enhances ASIC1a toxicity through a mechanism involving signaling cascades. Further, glutamate can activate metabotropic glutamate receptors. Alternate mechanisms to the one proposed by the authors are not explored.

The authors show that MK801 did not prevent MCAO-mediated cortical injury and therefore conclude that all effects in the MCAO model are independent of the NMDA receptor. Yet, there the methods reported are superficial and whether different doses were tested/used is not discussed. It would be great to see that the methods used could replicate the effect of MK801 in MCAO before

stating that all effects are independent of the NMDA receptor.

Other more minor Issues:

- (1) The number of cells recorded is reported without reporting the number of independent transfections/cultures. The number of slices is reported, but not the number of animals. Thus, the number of biological replicates is uncertain. Further, are these nested data as well?
- (2) The change in the EC50 for pH activation of ASIC with glutamate is not reported. The effect of glutamate on the channel's desensitization or calcium permeability is not addressed. This seems like a particularly important characteristic to analyze given the importance of calcium in ASIC1a-mediated cell death.
- (3) Was the pH of the solution checked after the addition of glutamate and other compounds?
- (4) Previous work by at least one author of the current manuscript showed that there is cross talk between NMDA receptor and ASICs with activation of NMDA receptor potentiating ASIC1a currents through intracellular signaling cascades. Can the authors comment on these findings and how they relate to this study? Are there controls that could be done to rule-out the previously described mechanisms of potentiation?
- (5) The involvement of metabotropic glutamate receptors was not explored in this mechanism. Do these compounds affect metabotropic glutamate receptors?
- (6) Can the authors state that they tested the pH of the solution AFTER the addition of the compounds? Small changes in pH can occur with addition of some compounds.
- (7) Refer to data that show the inhibitor LK-1 and LK-2 do not affect "basal" ASIC1a currents in the extended data, but this is not included. The impact of these inhibitors of ASIC1a currents (their proton-dose response and desensitization) should be included.
- (8) Mutants used as control often show differential pH sensitivity of gating. This confounds the use of these mutants as sole experimental tests for specific interpretations.

Referee #4 (Remarks to the Author):

In this manuscript, Lai et al discovered that glutamate and its structural analogs robustly potentiated currents mediated by acid-sensing ion channels (ASICs) and revealed that glutamate acts by increasing proton affinity and open probability of ASIC.

The authors hypothesized that glutamate drives Ca²⁺ overload through direct action on ASICs leading to NMDAR-independent cell death. They validated this proposition by measuring the effects of glutamate on calcium concentrations, mitochondrial dysfunction, and cell death in visual and auditory cortex in vitro. They also confirmed the excessive glutamate release can cause ischemic brain damage in an MCAO mouse model in a ASIC1a-dependent manner.

They further performed computational docking, molecular dynamics simulations, and site-directed mutagenesis experiments to reveal the binding cavity of glutamate in the ectodomain of ASIC1a. Finally, the authors proposed the development of small molecules that attenuate neuronal damage by targeting the glutamate-binding site on ASIC1a. Electrophysiological, biological assays, and structure-based computational drug screening experiments were performed, and a small molecule

LK-2 was identified as a competitive antagonist of the glutamate-binding site on ASICs, providing neuroprotection by specifically abolishing glutamate-induced potentiation of IASICs, independent of its action on NMDARs.

The work is competently executed, analyzed, and well described. The finding presents a novel regulation on ASIC that could have major impact for neuroprotection drug discovery.

- In line 63, the authors described "it is unknown whether glutamate can act directly on ASICs ...". Shteinikov V et al have reported this finding in Synapse, and therefore a more precise description is needed here.

- In line 69, the authors describe "glutamate, but not glutamic acid monosodium, potently potentiated ASICs evoked by solutions with pH values ranging from 6.55 to 7.4". This description is confusing and needs to be more rigorously presented because the distinction between glutamate and glutamic acid monosodium is not clear, especially in a buffered solution environment.

- In line 181, the authors only applied mutation-based analysis to identification of binding surfaces of glutamate. It is suggested to perform an in vitro binding assay, such as SPR, ITC, etc., to confirm the binding between ASIC1A (extracellular domain) and glutamate.

- In line 202, the authors determined that the binding pocket of glutamate is close to the outer vestibule of the ion permeation pathway. More discussion is needed here, especially as to how this binding mechanistically activates the ion-permeation pathway and whether this activation differs from other ASIC agonist models.

Point-by-point rebuttal

General responses to all referees:

We would like to take the opportunity to thank all the reviewers for your extremely fair and constructive comments on our manuscript.

There are five major concerns/criticisms (highlighted in *blue*) that multiple referees have converged onto. I will address each of these major comments collectively before outlining our responses to specific comments by each referee:

(1) Inconsistency of the results in cell death, LDH release experiments and ambiguous contribution of NMDARs to neuronal injury (Raised by Referee 1, 2 & 3).

We acknowledge that these are important issues that may have taken away the confidence level of the reviewers in our results. Indeed, there appeared to be much higher levels of cell death by PI staining than that was reported by LDH. This inconsistency may be largely attributed to an overestimate of basal cell death in acute brain slices by mechanical sectioning itself rather than a result of glutamate and acidosis insults. Our images of PI staining were collected at the depth of 30 μm from the cut surface of the slice while the release of LDH from these unchallenged brain slices were likely underestimated in 10 ml extracellular solution which had also diluted local LDH concentration substantially. It is therefore evident that the model we originally used is faulty.

To resolve these issues, we abandoned slice preps for analyses of cell viability under different conditions. Instead, we opted to culture cells being subjected to the oxygen-glucose deprivation (OGD) paradigm. This model allowed us to simulate ischemic condition while reliably measuring cell death and LDH release in culture neurons (without basal cell death) and CHO cells expressing ASIC1a (without other confounding factors in neurons). As suggested by the referees, we have done so in the presence of NMDAR competitive blocker D-AP5, AMPAR blocker NBQX, Group I & II mGluR blockers MPEP, LY367385 and LY341495 (Figure 2) as well as LK2 at a much lower concentrations (10 μM) than previously used (i.e. 100 μM) to target ASICs while avoiding off-target effects on NMDARs (Figure 5). These experiments provided unequivocal evidence for strengthening our conclusion that glutamate-dependent enhancement of ASICs exacerbates cell death independent of glutamate receptors.

(2) Lack of effects by memantine on infarct volume in ASIC+/+ MCAO mice that is contradictory to previous study (Raised by Referee 1, 2 & 3).

We believe that the timing at which memantine was applied and the temperature of heat-pad set for surgery may have all influence the infarct volume. Previous study¹ reported that memantine (10 mg/kg) was injected following 60 min MCAO, and the heat-pad temperature was set at 37°C, which in our hands yielded high mortality and the ceiling effect of infarct volume, both of which precluded us from testing the enhancing effect of glutamate on ASICs. In our experiments, mice were subjected to moderate ischemic challenge (30 min MCAO) after which memantine was injected (20 mg/kg) whereas the heat-pad temperature was set at 33°C. Although memantine did not show statistically significant effect under our previous experimental conditions, there was a trend.

We performed new experiments with MCAO model using different dose of memantine (1, 10 and 20 mg/kg), and validated its effectiveness but only at 1 mg/kg. Surprisingly, higher doses produced opposite effects. Nevertheless, this observation replicates the results previously reported in the literature². These aspects of our findings imparted the confidence to the validity of our model and outcomes of experiments on *Selfotel* and LK1/2 with memantine as a positive control. We now

incorporated these new results into Figure 2 & 5 as well as Extended Figure 4. Our results on CGS19755 (*Selfotel*) and LK2 are now consolidated to further strengthen the main conclusion.

(3) Lack of clarity in the interaction of glutamate with the channel's extracellular domains determined by the known Ca²⁺ binding site (Raised by Referee 1, 2 & 3)

As rightly pointed out by the reviewers, we had inappropriately chosen pH 6.35-6.8 range to activate ASIC1a channel with mutations of Ca²⁺ binding sites (*hASIC1a*^{E427G/D434C}), resulting in minimal positive modulating effect of glutamate on the mutant channel. This was partially attributed to smaller currents of the mutant channel to proton than wild-type ASIC1a³ such that more acidic solutions were chosen to evoke larger currents, and consequentially the positive modulating effect of glutamate was masked.

We performed experiments on this mutant channel using a narrower pH range (6.5-7.4) in the absence and presence of glutamate, demonstrating that glutamate remains effective in significantly shifting proton dose-response curve (DRC) towards the left (Extended Data Figure 1f, g). Furthermore, reducing Ca²⁺ concentration from 1 mM to 0.1 mM in extracellular fluid (ECF) did not affect the magnitude of glutamate induced potentiation or the left-shift in EC₅₀ (Extended Data Figure 1h). Independent of patch-clamp recordings, we performed the Microscale Thermophoresis (MST) Assay⁴ to directly demonstrate a direct interaction between *hASIC1a*^{E427G/D434C} and glutamate (Extended Data Figure 1j). In addition, we have tested the effect of Zinc (Zn²⁺) by including 10 μM N,N,N',N'-tetrakis (2-pyridylmethyl)ethylenediamine (TPEN) in all recording solutions to chelate Zn²⁺. Again, we found that I_{ASICs} was potentiated by glutamate (Extended Data Figure. 1i). These results indicated that the potentiation of I_{ASICs} by glutamate is independent of the known binding sites for Ca²⁺ and Zn²⁺, and instead most likely engages a previously unknown site on the ASIC1a channel.

(4) Whether a reduction effect of glutamate on mutant channels is due to a reduction in binding or secondary to an effect on channel gating? (Raised by Referee 3, 4)

To distinguish these possible effects on binding and/or gating, we have performed in vitro binding assays using microscale thermophoresis (MST) to directly measure ligand-protein binding independent of the channel gating as can be seen in several figures (Figure 1d, 3j, 4j) and Extended Data Figures (1j, 2g-h, 3f, 7h-k). In all cases, we obtained MST results that corroborate those from patch-clamp recordings of currents mediated by ASICs. As exemplified in Figure 1d, the binding of glutamate to ASIC1a is highly dependent on proton concentrations, in good parallel to DRCs from patch-clamp analyses under the same experimental conditions (Figure 1c). In addition to glutamate, we have applied MST assays to quantitatively measuring direct binding of other ligands including AP5, CGS19755 and LK2 to wild-type and mutant ASIC1a channels (i.e. K380A mutant devoid of glutamate binding site; E427G/D434C mutant devoid of Ca²⁺ binding sites) with or without pore blocker amiloride. These experiments collectively have provided unequivocal evidence to strengthen our main conclusion that the binding of glutamate to ASIC1a itself gives rise to its positive modulating effects on ASICs, consistent with single channel recordings where *P_o* is the main variable being increased but not amplitude or conductance.

(5) A lack of effect of monosodium glutamate on ASIC1a is incomprehensible (Raised by Referee 1, 2 & 4).

We were equally puzzled by a lack of effects of monosodium glutamate (MSG) on ASIC1a currents or binding in MST assay while glutamate itself positively modulate ASICs. However, the chiral effect of L- vs D-AP5 on ASICs prompted us to consider if geometric configuration of MSG plays a critical role in differentiating their binding to ASIC1a. Indeed, we found the difference is due to the contradictory optical activities (i.e. geometry configuration) of glutamate/MSG and L-AP5/D-AP5 in ECF at pH 7.0. The optical activity of MSG was levorotatory with an optical rotation of -4° in water while glutamate was dextrorotatory with an optical rotation of $+12^\circ$ in water or $+32^\circ$ in HCl. These findings suggested that MSG and glutamate exhibit different geometry configuration when dissolved in ECF. Similarly, D- and L-AP5 display opposite geometric configurations (Table 1).

Compounds	Optical activity	Optical rotation	Solvent
L-MSG	levorotatory	-4	Water
L-Glutamate	dextrorotatory	+12	Water
	dextrorotatory	+32	HCl
L-AP5	dextrorotatory	+26	HCl
D-AP5	levorotatory	-22	HCl

Given that the optical rotation of amino acid could be regulated by ECF pH⁵, we manipulated the optical rotation of MSG with HCl and NaOH first, and then

added into ECF to test if their effects on I_{ASICs} could be altered (see below Figure. 1b). We found increasing the optical rotation of MSG [MSG (+OR)] enhances I_{ASIC1a} at pH 7.0, while decreasing the optical rotation of glutamate [Glu (-OR)] diminishes the potentiation (see below Figure. 1c-e). Collectively, these findings further demonstrated that co-activation of ASIC1a by glutamate is specific to the geometry configuration of the ligand. These results have now been included in Extended Data Figure 3, providing important insights into the structural feature of ligands that enable their binding to the cavity on ASIC1a to exert positive allosteric modulation of I_{ASICs} .

Referees' Specific comments:

Referee #1 (Remarks to the Author):

In this study, the authors made a very novel discovery: glutamate and its structural analogs, including NMDAR antagonist AP5, potentiated currents mediated by acid-sensing ion channels (ASICs). Site-directed mutagenesis and structure-based in silico molecular docking and simulations uncovered a glutamate binding cavity in the extracellular domain of ASIC1a. Computational drug screening of NMDAR competitive antagonist analogs identified a small molecule, LK-2, that binds to this cavity and abolishes glutamate-dependent potentiation of ASIC currents but largely spares NMDARs, providing clear neuroprotection in vitro and in vivo. The authors conclude that glutamate serves as a co-agonist for ASIC1a to exacerbate neurotoxicity, and that selective blockage of glutamate binding sites on ASICs without affecting NMDARs may be of strategic importance for developing effective stroke therapeutics devoid of the side effects of NMDAR antagonists. The findings are original and represent immediate interest to the audience in field of ASICs and stroke. Having said that, more scrutinized analysis and stringent data need to be provided for injury experiments to clearly separate the contribution of glutamate receptors per se and that of ASICs potentiated by glutamate.

Response: We sincerely thank the reviewer for these positive comments on our study. We have now included additional data from new cell injury experiments to separate the contribution of glutamate receptors per se and that of ASICs potentiated by glutamate (See **General Response #1**).

1. P3 line 69-70: "We found that glutamate, but not glutamic acid monosodium, potently potentiated IASICs". Source and purity of glutamate should be described and some explanation of why glutamate but not monosodium glutamate potentiates ASICs. Wouldn't we expect to have glutamate when monosodium glutamate is dissolved in the solution?

Response: The glutamic acid (powder) used in this study was purchased from Sigma-aldrich (G1251), purity >99%. It turned out that the optical activity of monosodium glutamate (MSG) was levorotatory with an optical rotation of -4° in water, however, glutamate was dextrorotatory with an optical rotation of $+12^\circ$ in water or $+32^\circ$ in HCl. This finding suggested that MSG and glutamate exhibit different geometry configuration when dissolved in ECF, similar to the situation of D-/L-AP5. Our detailed analyses are presented in **General Response #5** and included in Extended Data Figure 3.

2 & 3. Figure 2f and 4f. Baseline cell death over 60% (e.g., at 7.4) is unacceptably high. With this high baseline cell death, the system is unreliable for detecting small/moderate changes of cell death. In contrast, small baseline LDH release was detected but the changes in LDH release after LK-1 and LK-2 were marginal (e.g., 16% vs. 13%, Figure 4g), even though statistical analysis showed high significance. On the other hand, in Figure 2g, glutamate appears to potentiate LDH release by 50% in ASIC1a-/- cells (9% to 14%?) yet the difference was not significant. Therefore, more stringent data and scrutinized analysis are required in cell culture experiments to increase the confidence in these results.

Response: We agree with the reviewer that our slice prep may not be an ideal model for cell viability assay due to a high baseline of cell death due to mechanical slicing itself. We redesigned all the cell death experiment by subjecting primary cultured neurons and CHO cells to OGD paradigm, a classical model to mimic ischemia. Under such conditions, we found a striking parallel between cell death in calcein-PI staining and LDH release (Figure 2f-i and Figure 5a-c) above the basal cell death rate in WT (see **General Response #1**). These new results provided compelling evidence to demonstrate that the glutamate-induced cell injury is ASIC1a dependent.

4. Figure 2i: Memantine does not show any effect on infarct volume in ASIC^{+/+} mice. This is contradictory to the well-known contribution of glutamate toxicity to stroke damage. How is this finding compared to those of others who studied the effects of glutamate antagonists in stroke models? At what time was memantine applied?

Response: We agree with the reviewer that our results on memantine are somewhat confusing. We believe there were significant differences in our experimental conditions with respect to the timing at which memantine was applied and the temperature set for heat-pad, all of which might influence the infarct volume (**General Response #1**). To clarify the effect of memantine, we tested various dosage of memantine (1, 10, 20 mg/kg) in MCAO model, but surprisingly significant neuroprotective effect is only observed at the dosage of 1 mg/kg (Figure 2l, m and Extended Data Figure 4k). However, this is consistent with a previous work reported that high concentration of memantine tended to increase cell death while low concentration of memantine protected neurons from ischemic damage². Therefore, our model itself can serve as a reliable readout of the severity of ischemic injury in vivo.

5. P4, line 86-87: “Thus, glutamate appears to act as a co-agonist to allosterically potentiate ASICs, reminiscent of glycine as a co-agonist for NMDARs”. This description is not accurate since NMDAR does not function without the presence of a co-agonist but ASIC1a does not require a co-agonist to open.

Response: Agree, and we have removed the term “co-agonist” throughout the revised manuscript. Instead, we refer glutamate as a positive allosteric modulator for ASICs.

6. P4, line 93-94: “reversal potential (control: 33.15 mV; glutamate: 28.56 mV) (Extended Data Fig. 1h, i)”. Were these values expected for ASICs? In most whole-cell recordings, the reversal potential for ASICs is close to +60 mV (near the sodium equilibrium potential). The authors need to provide some explanation for the different reversal potentials they recorded.

Response: The reversal potential for ASICs is close to +60 mV when the intracellular solution contains Cs⁺⁶. This is because Na⁺ and Ca²⁺ influx through ASIC1a while K⁺ efflux influence the reversal potential. In our experiment, the intracellular solution contains K⁺, but not Cs⁺, which makes the reversal potential left shift relative to +60 mV⁷. These info and references are cited in the legend (Extended Data Figure 1).

7. P6, line 137-139: “We found that glutamate markedly increased the percentage of cell death in Asic1a^{+/+} slices, whilst in contrast, Asic1a^{-/-} slices had decreased cell death and seemed resistant to glutamate insults (Fig. 2e,f)”. Since glutamate can cause cell death through NMDA receptors, these experiments should be repeated in the presence of glutamate antagonist that does not block the binding site on ASIC1a but abolish the neuronal injury through NMDA receptors directly (e.g., MK801).

Response: In response to this suggestion, we have repeated the cell death experiments in the presence of D-AP5 (Figure 2f-i). Surprisingly, we found that addition of a cocktail of glutamate receptor blockers including D-AP5 did not block cell death in WT neurons while PcTX-1 is highly protective, consistent with the results in Asic1a^{-/-} neurons, suggesting OGD-dependent cell death can be largely accounted for by acidosis toxicity (Page 7-8).

8.P7, line 160-162: “Together, these data indicated that ischemic brain damage in an MCAO mouse model was NMDAR-independent and largely ASIC1a dependent”. Again, this finding is inconsistent with previous studies that showed neuroprotective effect of glutamate antagonists including those that do not interact with the glutamate binding site, e.g., MK801.

Response: As discussed in **General Response #1** and specific **Response #4**, we have established the reliability of MCAO model with memantine at 1 mg/kg (Figure 2l, m) as a positive control for our analyses. Like MK-801, memantine is a pore blocker and unlikely affects glutamate binding to either ASICs or NMDARs.

9. P11-12, line 299-302: “in cell death and LDH release assays under the acid condition with glutamate treatment in vitro, LK-2 remarkably reduced cell death and LDH release to an extent close to CGS19755, indicating glutamate-enhanced ASIC activity rather than NMDAR overactivation was the main contributor to cell injury in stroke conditions (Fig. 4e-g)”. There are three issues associated with the data presented in Fig4e-g. One: the reduction in cell death or LDH release by LK-2 is not remarkable (~10-15% in cell live/dead counting or LDH release). Two: the concentration used for LK-1 and LK-2 (100 uM) may not differentiate between the direct effect on NMDA reporters and indirect effect on ASICs. Three: MK801, an open channel blocker of NMDA receptor that does not interact with the glutamate binding site, shows a better or comparable effect in reducing LDH release compared with that of LK-1, LK-2, and CGS, arguing strongly against the claim that glutamate-enhanced ASIC activity rather than NMDAR overactivation was the main contributor to cell injury in stroke conditions!

Response: We have taken all these three important issues into serious consideration. Given the intrinsic limitations of high basal cell death rate with slice prep for cell viability assay, we have now clearly shown protective effects of 10 μ M LK-2 and 10 μ M CGS19755, close to their IC₅₀, on cell live/dead counting or LDH release using cultured cells with minimal basal cell death (Figure 5a-e). These results are consistent with the effect of memantine in vivo. Together, our study demonstrated that the glutamate-enhanced ASIC activity rather than NMDAR overactivation is the main contributor to cell injury in different ischemic models.

10. Extended data Figure 1g: the representative traces do not show appreciable potentiation of ASIC current by glutamate and the dose response curves do not show a significant shift/separation.

Response: We carefully repeated the experiments at pH range from 7.4 to 6.55, showing an apparent left-shift in dose response curve (Extended Data Figure 1f, g).

Referee #2 (Remarks to the Author):

Lai and colleagues have prevented an interesting and novel manuscript where they suggest that glutamate is a positive allosteric modulator of acid sensitive ion channel 1a with relevance to neuronal death during stroke. They use a breadth of techniques (electrophysiology, mutagenesis, molecular dynamics simulations and modelling). Perhaps most impactful is the discovery and evaluation of a new small molecule therapeutic, LK-2. In brief, they show that glutamate augments ASIC1a currents at acidic pH, but not at physiological pH, which is likely highly relevant to stroke. They demonstrate that relevant EC50 for glutamate and acidosis that occurs during stroke. It is demonstrated that glutamate enhances neuronal death at pH 7 in vitro and in vivo and describe the contribution of ASIC1a-mediated calcium influx to mitochondrial dysfunction. Using in silico molecular docking and site-directed mutagenesis, they identify the apparent binding pocket for glutamate and several NMDAR antagonists. The latter are used to rationally design an inhibitor with minimal activity at NMDARs and demonstrate neuroprotection with this compound. Statistical methods and experimental power are appropriately applied. Overall the conclusions are important and likely to be of importance to a broad range of neuroscience. Furthermore, the discovery of LK-2 as a starting point for a new class of putative neuroprotective drugs could greatly impact clinical stroke care. There are however some inconsistencies in the data as well as a conceptual issue that raise concerns:

Response: We thank the reviewer for these positive comments on our manuscript. We have performed additional experiments and provided new data in the revised manuscript to address the concerns.

1) Conceptually, I'm having difficulty understanding how ASIC1a channels make such a large contribution to Ca²⁺ overload. The channels inactivate rapidly as shown by others and in the whole-cell recordings presented in this manuscript. The channels are closed within about 1s and yet the Ca²⁺ dysregulation persists for minutes (figure 2). Are the channels causing release from stores? For example, the loss of mitochondrial membrane potential will likely cause these organelles to release Ca²⁺.

Response: We performed Ca²⁺ imaging under different conditions to investigate the source of Ca²⁺. In the presence of CGP37157 and RO2959 in ECF, blockers for mitochondrial Na⁺/Ca²⁺ exchanger and Ca²⁺ release-activated Ca²⁺ (CRAC) channel in endoplasmic reticulum, respectively, glutamate still generated a long-lasting Ca²⁺ elevation in neurons. However, no significant change of fluorescence was observed following 0 Ca²⁺ ECF perfusion. In the presence of PcTX-1, but not mGluRs blocker LY341495, the elevation of [Ca²⁺]_i was suppressed (Extended Data Figure 4b, c). These results indicated that the persistent Ca²⁺ must originate from influx from the extracellular space through ASICs.

Our additional experiments indicated that glutamate attenuates both instantaneous and steady-state desensitization (Figures. 2d, e and 1b), leading to a persistent current mediated by ASIC1a. This current provides a continuous influx of Ca²⁺ through ASICs, propelling intracellular Ca²⁺ accumulation. This effect is evidently independent of other ionotropic and metabotropic glutamate receptors as a cocktail of blockers do not block this sustained current, further supporting that glutamate induced Ca²⁺ overload via ASIC1a channel.

2) The title, Pg 4 line 86, and line 196 it is stated that glutamate is a co-agonist for ASIC1a. This idea is compared to glycine for NMDARs. ASIC1a channels open in the absence of glutamate and glutamate alone does not cause channel gating. This is more consistent with a role for glutamate as a positive allosteric modulator, not a co-agonist. This does not detract from the impact of the

discovery, but should be corrected for consistency.

Response: Agree and we have removed the term “co-agonist” throughout the revised manuscript. Instead, we refer glutamate as a positive allosteric modulator for ASICs.

3) It is reported (line 69-72) that glutamate but not glutamic acid monosodium is a PAM at ASIC1a. This seems impossible to me. Glutamate and glutamic acid have the same structure and both are going to be protonated in the pH ranges used in the study. The only clear difference is the addition of Na+. Is Na+ inhibiting the binding of glutamate to ASIC1a? This could be evaluated with addition of equimolar NaCl to the glutamate condition. If not, why are these molecules behaving differently?

Response: It turned out that the optical activity of monosodium glutamate (MSG) was levorotatory with an optical rotation of -4° in water, however, glutamate was dextrorotatory with an optical rotation of $+12^\circ$ in water or $+32^\circ$ in HCl. This finding suggested that MSG and glutamate exhibit different geometry configuration when dissolved in ECF, similar to the situation of D-/L-AP5. Our detailed analyses are presented in **General Response #5** and included in Extended Data Figure 3.

4) Figure 1b, what pH was the dose-response curve determined at? Is the dose response the same at different pH levels? It could be predicted that pH increases affinity of glutamate for the channels to more negatively influence cell health.

Response: We apologize for this omission. We have now included glutamate dose-response curves at pH 7.0 and 6.8, respectively, showing different EC₅₀ values for two conditions (i.e. 462 μ M at pH 7.0 vs 382 μ M at pH 6.8) at pH 7.0 (Figure 1c). In addition, MST assay also revealed that glutamate binds to ASIC1a with a smaller K_d at pH 6.8 than that at pH 7.0 (Figure 1d). These results demonstrated that both ligands can bind to ASIC1a and reciprocally affect each other's affinity.

5) Part of understanding the interaction of glutamate with the channels extracellular domains was determining if glutamate interacts with the known Ca²⁺ binding site. However, the potentiation of hASIC1aE427G/D434C by glutamate, as suggested in Extended Fig 1g, is not convincing. The leftward shift in the curve is modest at best. How was potentiation in these conditions statistically supported? It does appear that mutagenesis of the Ca²⁺ binding site may be important in some way for glutamate binding, whether it is a general structural modification that influences the glutamate pocket.

Response: Mutagenesis of the Ca²⁺ binding sites appear to modulate the proton affinity for ASIC1a³. By repeating the experiments with a narrower pH range from 7.4 to 6.55, we can observe a more apparent left shift of dose-response curve by glutamate (Extended Data Fig 1f, g). Moreover, we performed MST assays to confirm that glutamate can bind to hASIC1aE427G/D434C (Extended Data Fig 1j). Together, we concluded that mutagenesis of the Ca²⁺ binding sites has little impact on glutamate-dependent potentiation of ASIC1a currents (also see **General Response #3**).

6) D-APV will potently inhibit NMDARs but not block the glutamate potentiation of ASIC. Memantine and MK801 require channel opening and could allow for significant calcium influx prior to block. Further, NMDARs are known to enhance ASIC1a via nitrosylation (Gao, Neuron 2005), it is possible that this Ca²⁺-dependent cross talk is occurring prior to block by MK801. Key experiments (i.e. cell death assay, mitochondrial assay) should be done in the presence of D-APV instead of MK801 to completely rule out contributions of NMDARs?

Response: We appreciate the reviewer's suggestion of completely ruling out contributions of NMDARs. To this end, we performed Calcein-PI staining and LDH release assay in the presence of glutamate receptor competitive antagonists including D-AP5 (Figure 2f-i). Interestingly, we found

that addition of glutamate receptor blockers including D-AP5 offered no protective effects. In contrast, ASIC1a pore blocker amiloride was highly effective in attenuating cell death, reminiscent of the results from *Asic1a*^{-/-} neurons. These observations reinforced the idea that glutamate acts on ASICs to promote cell death.

7) Calcein / PI staining. How was % cell death determined? The basal rate of (apparent) cell death is ~50% in the slices. This seems very high and raises concerns about overall health of the slices or that images were collected too close to the cut surface of the slice. At what depth were the cells imaged?

Response: The % of cell death was determined by dead cells (red)/[dead cells (red)+live cells (green)]*100 as described in METHOD. We recognize that the slice prep is faulty for cell-death assay due to slicing induced elevation in basal cell death. We have now redone cell viability analyses using cultured cells being subjected to the classical OGD paradigm. Under such conditions, we found a striking parallel between cell death in calcein-PI staining and LDH release (Figure 2f-i and Figure 5a-c) above the basal cell death rate in WT (see **General Response #1**). These new results provided compelling evidence to demonstrate the glutamate-induced cell injury is ASIC1a dependent.

8) It is very surprising that memantine showed no significant neuroprotection in the MCAO model. Numerous groups have shown potent protection by memantine. I see that body temperature was maintained at 33C, which is quite hypothermic and could account for the apparent lack of contribution by NMDARs.

Response: We agree with the reviewer that our results on memantine are somewhat confusing. We believe there were significant differences in our experimental conditions with respect to the timing at which memantine was applied and the temperature set for heat-pad, all of which might influence the infarct volume (**General Response #1**). To clarify the effect of memantine, we tested various dosages of memantine (1, 10, 20 mg/kg) in MCAO model, but surprisingly significant neuroprotective effect is only observed at the dosage of 1 mg/kg (Figure 2l, m and Extended Data Figure 4k). However, this was consistent with a previous report showing that high concentration of memantine tended to increase cell death while low concentration of memantine protected neurons from ischemic damage². Therefore, our model itself can serve as a reliable readout of the severity of ischemic injury in vivo.

9) Were MD simulations done using a pH7 or lower model? This is important because there is little to no potentiation by glutamate at pH7.4.

Response: MD simulations were performed at pH 7 as described in METHOD. Consistent with our electrophysiology results that there was little to no potentiation by glutamate at pH 7.4, MST assay did not detect direct binding between ASIC1a and glutamate at pH 7.4 either (Extended Data Figure 7j, k).

10) Line 240. Not clear why memantine was evaluated for blocking glutamate effects on ASIC1a. memantine is a non-competitive, pore blocker and does not interact with the glutamate / APV sites on NMDARs.

Response: We removed these data.

11) Test LK2 in the ASIC1a k.o animals in addition to selfotel. This could help address some potential off target effects.

Response: LK-2 and CGS19755 have been tested in both ASIC1a WT and KO mice (Figure 5f, g),

showing that LK2 can achieve outstanding neuroprotective effects, comparable to CGS19755.

12) Importantly, the effective concentration of LK-2 in the brain appears too low to have a significant effect on ASIC1a. The authors report a brain level of 235 ug/L. The formula weight of LK-2 (C12H17N2O5P) is 300.25 (this should be reported in the paper), which is a concentration of ~0.7uM. This would appear to inhibit the channels by about 10-20%. The neuroprotective effect in vivo is at odds with this.

Response: We appreciate the reviewer's comment. However, it is worth noting that the concentration of LK-2 was measured using the whole brain tissue but not the cerebrospinal fluid. If we consider a dilution factor of ~50 from wet tissue mass⁸, we would argue that it is likely that the maximal concentration can reach micromolar range and even higher if BBB around the infarct is locally damaged. Furthermore, there is added value for neuroprotective effect as LK2 displays favorable multiple pharmacokinetic parameters, such as terminal elimination half-life and clearance rate (Extended Data Table 5). The molecular weight of LK-2 has been added in Figure 4f. We have incorporated these changes into the manuscript (Page 14).

13) Reductions in lesion size are an interesting measure of neuroprotection, but do not always relate directly to behavioural protection or recovery. The LK-2 drug should be evaluated in behavioural assays relevant to stroke.

Response: Given that transient occlusion of middle cerebral artery (MCA) mainly results in infarction of the ipsilateral frontal cortex (including motor cortex) and part of the striatum in the MCA territory in our MCAO model⁹, which could damage the motor behaviors, we performed rotarod task, a commonly used rodent motor behavior paradigm¹⁰, 24 hours and 7 days post MCAO to evaluate the short-term and long-term effects of LK-2 on motor learning (a part of cognitive function) and movement coordination ability. These new results demonstrated that LK-2 significantly improves neurobehavioral outcomes one week after MCAO (Figure 5h-j and extended data Figure 10g).

Referee #3 (Remarks to the Author):

This manuscript describes a novel potentiation of ASIC1a proton-gated currents by glutamate. Specifically, the authors show pH 7.0 currents potentiated in whole cell and single channel configurations and enhanced ASIC1a-dependent changes in calcium and mitochondrial dysfunction. The authors confirm previous work that ASIC1a is involved in cell death in cultured slices and show potentiation of cell death with glutamate that is limited in ASIC1a knockout animals. The authors identify an area of ASIC1a that, when mutated appears to lower glutamate-induced potentiation. They also identify compounds which interact with this site that can apparently inhibit glutamate-induced potentiation of ASIC1a and (for some) the NMDA receptor and show these molecules limit MCAO-mediated death. The authors provide a compelling and linear interpretation of their data. The idea that there is a glutamate binding site in ASIC1a that potentiates currents is exciting. Further, the discovery of LK-2 as an inhibitor of this phenomenon and the initial effect of this compound in the MCAO model is extremely promising. However, numerous weaknesses related to interpretation of data, the limited characterization of the effect and compounds and mutant channels, and a disconnection between the written results section and data included (especially within the extended data suggestion) reduce enthusiasm for the manuscript.

We appreciate the reviewer's recognition of our study as significant and compelling. In the revised manuscript, we have performed additional experiments and revised the text to address the questions raised in the previous round of review. We hope these revisions have strengthened the overall convincingness of our work.

1. There have been multiple extracellular modulators or "co-agonists" reported for ASIC function such as lactate, spermine, calcium, zinc, peptides, ammonium, etc. Although the shift in the pH50 for ASIC1a activation with glutamate is not reported, it appears that glutamate acts like many of these compounds to shift the apparent proton response for ASIC1a activation. The authors imply that this is a new mechanism of modulation, and state that it is independent of calcium binding. This is important as several exciting reports of gating modulators and alternate agonists/co-agonists were later found to affect the apparent proton sensitivity of ASIC1a by altering the availability of zinc or calcium... or the channels' response to calcium and, thus, indirectly affect channel gating. The involvement of calcium and zinc (which are likely present in the recording solutions) was not investigated through the use of chelators or ion substitution. The authors do not investigate glutamate response in a mutant channel likely insensitive to calcium (with altered expression and proton response curve) and state that glutamate still potentiates the current. However, this potentiation is NOT evident in the extended data (Figure 1g) as stated in the results section. In fact, it appears that this channel is unaffected by glutamate in the proton concentration response curve. This is significant as it speaks to the mechanism of action of glutamate, its novelty, and the mechanism of action of the newly identified compounds.

Response: The concern that glutamate may act like other compounds to shift the apparent proton response for ASIC1a activation is valid. To address this issue, we performed a series of experiments to rule in/out potential complications that involve Ca²⁺ binding site mutations, Ca²⁺ and Zn²⁺ modulations of ASICs (Also see **General Response #3**).

We performed experiments on this mutant channel using a narrower pH range (6.5-7.4) in the absence and presence of glutamate, demonstrating that glutamate remains effective in significantly shifting the proton dose-response curve towards the left (Extended Data Figure 1f, g). Furthermore, reducing Ca²⁺ concentration from 1 mM to 0.1 mM in extracellular fluid (ECF) did not affect the effect size of glutamate-induced left-shift in EC₅₀ (Extended Data Figure 1h). Independent of patch-clamp recordings, we performed the Microscale Thermophoresis (MST) Assay⁴ to directly

demonstrate a direct interaction between glutamate and *h*ASIC1a^{E427G/D434C} (Extended Data Figure 1j). In addition, we have tested the effect of Zn²⁺ by including 10 μM N,N,N',N'-tetrakis (2-pyridylmethyl)ethylenediamine (TPEN) in all recording solutions to chelate Zn²⁺. Again, we found that I_{ASICs} was potentiated by glutamate (Extended Data Figure 1i). These results indicated that the potentiation of I_{ASICs} by glutamate is independent of the known binding sites for Ca²⁺ and Zn²⁺, and instead most likely engages a novel cavity on the ASIC1a channel as described in this manuscript (Page 4-5).

2. ASIC1 knockout animals have been shown to have altered signaling cascades and CamKII levels. The authors rely on the sole use of these animals to determine ASIC-specific effects for cell death and mitochondrial function. The use of ASIC1a-dependent inhibitors (such as PcTx1 and other venom peptides) was not tested in most experiments and models.

Response: In compliance with this suggestion, we have performed additional experiments with PcTx1 to complement the results from both ASIC1a WT and KO mice. The effect of PcTx1 on Ca²⁺ and cell-death are consistent with those from ASIC1a KO and now included both main and extended figures (Extended Data Figure 4b, c; Figure 2g-k).

3. The ASIC1 KO slices showed an increase in LDH release with pH 7.0 and glutamate (Figure 2G) suggesting there are ASIC1a-independent effects. This result is minimized in the text of the results.

Response: We have redesigned the cell death experiments and details are described in **General Response #3** and reflected in the revised text and figures.

4. Binding of glutamate to the channel is not shown, but inferred through modeling and the reduced effect on a mutant channel that also appears to have an alteration to the pH dependence of activation (which is not investigated or characterized). Further, solid measurements of the concentration response of glutamate in these models is sorely needed to better ensure that a reduction of effect is due to a reduction in binding (and not secondary to an effect on channel gating).

Response: To this end, we performed MST assays to confirm the concentration dependent binding between ASIC1a and glutamate at pH 7.0 and measured the K_d values between 1) ASIC1a and glutamate, 2) ASIC1aK380A and glutamate, 3) ASIC1a and glutamate in the presence of amiloride, a pore blocker of ASIC1a. Indeed, direct binding between ASIC1a and glutamate can be demonstrated with or without amiloride. On the contrary, glutamate association with ASIC1a K380A mutant is unstable, precluding an estimation of its K_d (Figure 1d; Figure 3j; Extended Data Figure 7h, i). Moreover, no conspicuous binding between ASIC1a and glutamate at pH 7.4 was detected (extended data Figure 7j, k). Together, these results suggested that a reduction of effect is due to ablated/weakened binding, but not secondary to an effect on channel gating in mutant channels.

5. The manuscript in multiple places states that glutamate causes NMDAR-independent neurotoxicity. Previously reports (Neuron 2005) show that activation of NMDA enhances ASIC1a toxicity through a mechanism involving signaling cascades. Further, glutamate can activate metabotropic glutamate receptors. Alternate mechanisms to the one proposed by the authors are not explored.

Response: We have tune down the narrative by removing or rewording “NMDAR-independent” neurotoxicity which probably is no properly used. However, we expanded our experiments by testing blockers of NMDAR (50 μM D-AP5) and metabotropic glutamate receptors (10 μM LY341495) in

cell death experiments (Figure 2g-i) and Ca²⁺ imaging (Extended Data Figure 4b, c) to highlight the mechanistic focus on the impact of glutamate binding to ASIC1a.

6. The authors show that MK801 did not prevent MCAO-mediated cortical injury and therefore conclude that all effects in the MCAO model are independent of the NMDA receptor. Yet, the methods reported are superficial and whether different doses were tested/used is not discussed. It would be great to see that the methods used could replicate the effect of MK801 in MCAO before stating that all effects are independent of the NMDA receptor.

Response: There is an error here because we only tested other pore blocker memantine but not MK801 in the MCAO model. To clarify the effect of memantine, we tested various dosages (1, 10, 20 mg/Kg) of memantine in MCAO model. We observed a significant neuroprotective effect only at dosage of 1 mg/kg (Figure 2l, m and Extended Data Figure 4k), consistent with that previously reported, in which high concentration of memantine tended to increase cell death while low concentration of memantine protected neurons from ischemic damage². Please also refer to **General Response #1**.

Other more minor Issues:

(1) The number of cells recorded is reported without reporting the number of independent transfections/cultures. The number of slices is reported, but not the number of animals. Thus, the number of biological replicates is uncertain. Further, are these nested data as well?

Response: The number of biological replicates has been added and reported as nested data (Method Statistics page 7).

(2) The change in the EC₅₀ for pH activation of ASIC with glutamate is not reported. The effect of glutamate on the channel's desensitization or calcium permeability is not addressed. This seems like a particular important characteristic to analyze given the importance of calcium in ASIC1a-mediated cell death.

Response: The EC₅₀ has been added in all related figures. The effect of glutamate on the desensitization of ASICs has been tested (Figure 1b). Glutamate shifted the mid-point of pH for desensitization from 7.303±0.008 to 7.216±0.008. Our additional experiments indicate that glutamate attenuates both instantaneous and steady-state desensitization (Figs. 2d, e & 1b), leading to a persistent current mediated by ASIC1a. This provides a continuous influx of Ca²⁺ through ASICs, propelling intracellular Ca²⁺ accumulation. This effect is evidently independent of other ionotropic and metabotropic glutamate receptors as a cocktail of blockers do not block this sustained current, further supporting that glutamate induces Ca²⁺ overload via ASIC1a channel (Figure 2d, e).

(3) Was the pH of the solution checked after the addition of glutamate and other compounds?

Response: The pH change of the solution after the addition of glutamate and other compounds was adjusted to no more than 0.02 as described in METHOD.

(4) Previous work by at least one author of the current manuscript showed that there is cross talk between NMDA receptor and ASICs with activation of NMDA receptor potentiating ASIC1a currents through intracellular signaling cascades. Can the authors comment on these findings and how they relate to this study? Are there controls that could be done to rule-out the previously described mechanisms of potentiation?

Response: The previous study (Neuron, 2005) indeed demonstrated that the activation of NMDAR potentiates ASIC1a currents through intracellular signaling cascades (i.e. after 5 min OGD),

implying acidosis neurotoxicity is secondary to NMDAR activation. Although cell death may be the end results of both studies, our current study would argue that glutamate and proton co-released from synaptic vesicles can directly activate both NMDARs and ASICs and their downstream cell-death signaling cascade at the very onset of ischemic insult. However, intracellular cross-talk between NMDARs and ASICs can be a part of the vicious cycle that amplifies progressive cell death in stroke. We have elaborated the conceptual difference between these two studies in our discussion (Page 16).

(5) The involvement of metabotropic glutamate receptors was not explored in this mechanism. Do these compounds affect metabotropic glutamate receptors?

Response: As suggested by the reviewer, we have tested 10 μ M LY341495, a non-selective blocker for metabotropic glutamate receptors, in both cell death experiments (Figure 2g-i) and Ca^{2+} imaging (Extended Data Figure 4b, c). We found no significant effect of mGluRs in this regard.

(6) Can the authors state that they tested the pH of the solution AFTER the addition of the compounds? Small changes in pH can occur with addition of some compounds.

Response: We are aware of this important point such that the pH change of the solution after the addition of glutamate and any other compounds was always checked and adjusted to no more than 0.02 as described in METHOD.

(7) Refer to data that show the inhibitor LK-1 and LK-2 do not affect “basal” ASIC1a currents in the extended data, but this is not included. The impact of these inhibitors of ASIC1a currents (their proton-dose response and desensitization) should be included.

Response: We have tested the impact of LK-1 and LK-2 to ASIC1a currents and found that these two compounds did not affect proton-dose response and desensitization of ASIC1a currents (Extended Data Figure 10c, d).

(8) Mutants use as control often show differential pH sensitivity of gating. This confounds the use of these mutants as sole experimental tests for specific interpretations.

Response: As discussed above in our response to major comment #4, we performed MST assays to independently confirm the concentration dependent ligand-ASIC1a binding at pH 7.0 between 1) ASIC1a and glutamate, 2) ASIC1aK380A and glutamate, 3) ASIC1a and glutamate in the presence of amiloride.

Referee #4 (Remarks to the Author):

In this manuscript, Lai et al discovered that glutamate and its structural analogs robustly potentiated currents mediated by acid-sensing ion channels (ASICs) and revealed that glutamate acts by increasing proton affinity and open probability of ASIC. The authors hypothesized that glutamate drives Ca²⁺ overload through direct action on ASICs leading to NMDAR-independent cell death. They validated this proposition by measuring the effects of glutamate on calcium concentrations, mitochondrial dysfunction, and cell death in visual and auditory cortex in vitro. They also confirmed the excessive glutamate release can cause ischemic brain damage in an MCAO mouse model in a ASIC1a-dependent manner. They further performed computational docking, molecular dynamics simulations, and site-directed mutagenesis experiments to reveal the binding cavity of glutamate in the ectodomain of ASIC1a. Finally, the authors proposed the development of small molecules that attenuate neuronal damage by targeting the glutamate-binding site on ASIC1a. Electrophysiological, biological assays, and structure-based computational drug screening experiments were performed, and a small molecule LK-2 was identified as a competitive antagonist of the glutamate-binding site on ASICs, providing neuroprotection by specifically abolishing glutamate-induced potentiation of ASICs, independent of its action on NMDARs. The work is competently executed, analyzed, and well described. The finding presents a novel regulation on ASIC that could have major impact for neuroprotection drug discovery.

We thank the reviewer for these positive comments on our manuscript.

- In line 63, the authors described “it is unknown whether glutamate can act directly on ASICs ...”. Shteinikov V et al have reported this finding in Synapse, and therefore a more precise description is needed here.

Response: We thank the reviewer for bring our attention to this paper which we were not aware of. In the context of this publication (cited), we have reworded this sentence in the Introduction, which reads “Aside from direct activation of NMDARs and their downstream intracellular signaling pathways, glutamate has recently been found to enhance ASIC1a currents in recombinant expression system¹¹. However, it is unknown whether glutamate directly or indirectly acts on native ASICs in neurons to mediate and/or aggravate ischemic brain injury”.

- In line 69, the authors describe “glutamate, but not glutamic acid monosodium, potently potentiated ASICs evoked by solutions with pH values ranging from 6.55 to 7.4”. This description is confusing and needs to be more rigorously presented because the distinction between glutamate and glutamic acid monosodium is not clear, especially in a buffered solution environment.

Response: It turned out that the optical activity of monosodium glutamate (MSG) was levorotatory with an optical rotation of -4° in water, however, glutamate was dextrorotatory with an optical rotation of +12° in water or +32° in HCl. This finding suggested that MSG and glutamate exhibited different geometry configuration when dissolved in ECF, similar to the situation of D-/L-AP5. Our detailed analyses are presented in **General Response #5** and included in Extended Data Figure 3.

- In line 181, the authors only applied mutation-based analysis to identification of binding surfaces of glutamate. It is suggested to perform an in vitro binding assay, such as SPR, ITC, etc., to confirm the binding between ASIC1A (extracellular domain) and glutamate.

Response: To this end, we performed MST assays to confirm the concentration dependent binding between ASIC1a and glutamate at pH 7.0 between 1) ASIC1a and glutamate, 2) ASIC1aK380A and glutamate, 3) ASIC1a and glutamate in the presence of pore blocker amiloride. Indeed, direct binding between ASIC1a and glutamate can be demonstrated with or without amiloride. On the contrary,

glutamate association with ASIC1a K380A mutant is unstable, precluding an estimation of its K_d (Figure 1d; Figure 3j; Extended Data Figure 7h, i). Moreover, no conspicuous binding between glutamate and ASIC1a-GFP at pH 7.4 was detected, nor is there binding between glutamate and GFP itself (Extended Data Figure 7j, k). Together, these results indicated that a reduction of glutamate effect is due to ablated/weakened binding, but not secondary to an effect on channel gating in mutant channels.

- In line 202, the authors determined that the binding pocket of glutamate is close to the outer vestibule of the ion permeation pathway. More discussion is needed here, especially as to how this binding mechanistically activates the ion-permeation pathway and whether this activation differs from other ASIC agonist models.

Response: The exact mechanistical conformational change of glutamate binding on ASIC1a can only be speculated at this stage without extensive work using Cryo-EM, but agree with the referee that this is an interesting topic for future work. However, we have expanded more discussion in the text. We stated on Page 10: “Similar and close to proton binding pocket, the putative binding pocket for glutamate is a solvent-exposed and electrostatically positive cavity (Extended Data Table 4) that formed at subunit interfaces stabilized by hydrophobic and polar contacts across the finger, thumb and palm domains. We postulate that this conformation may undergo rearrangements of thumb domain upon glutamate binding and transduced to the channel pore via the palm domain to enhance ion permeation”.

References cited for the rebuttal:

- 1 Xiong, Z. G. *et al.* Neuroprotection in ischemia: blocking calcium-permeable acid-sensing ion channels. *Cell* **118**, 687-698, doi:10.1016/j.cell.2004.08.026 (2004).
- 2 Trotman, M., Vermehren, P., Gibson, C. L. & Fern, R. The dichotomy of memantine treatment for ischemic stroke: dose-dependent protective and detrimental effects. *J Cereb Blood Flow Metab* **35**, 230-239, doi:10.1038/jcbfm.2014.188 (2015).
- 3 Paukert, M., Babini, E., Pusch, M. & Grunder, S. Identification of the Ca²⁺ blocking site of acid-sensing ion channel (ASIC) 1: implications for channel gating. *The Journal of general physiology* **124**, 383-394, doi:10.1085/jgp.200308973 (2004).
- 4 Jerabek-Willemsen, M. *et al.* MicroScale Thermophoresis: Interaction analysis and beyond. *Journal of Molecular Structure* **1077**, 101-113, doi:10.1016/j.molstruc.2014.03.009 (2014).
- 5 Kundrat, M. D. & Autschbach, J. Computational modeling of the optical rotation of amino acids: a new look at an old rule for pH dependence of optical rotation. *J Am Chem Soc* **130**, 4404-4414, doi:10.1021/ja078257l (2008).
- 6 Chu, X. P., Close, N., Saugstad, J. A. & Xiong, Z. G. ASIC1a-specific modulation of acid-sensing ion channels in mouse cortical neurons by redox reagents. *The Journal of neuroscience : the official journal of the Society for Neuroscience* **26**, 5329-5339, doi:10.1523/JNEUROSCI.0938-06.2006 (2006).
- 7 Wu, L. J. *et al.* Characterization of acid-sensing ion channels in dorsal horn neurons of rat spinal cord. *The Journal of biological chemistry* **279**, 43716-43724, doi:10.1074/jbc.M403557200 (2004).
- 8 Rudick, R. A., Zirretta, D. K. & Herndon, R. M. Clearance of albumin from mouse subarachnoid space: a measure of CSF bulk flow. *J Neurosci Methods* **6**, 253-259, doi:10.1016/0165-0270(82)90088-7 (1982).
- 9 Li, Y. & Zhang, J. Animal models of stroke. *Animal Model Exp Med* **4**, 204-219, doi:10.1002/ame2.12179 (2021).
- 10 Cao, V. Y. *et al.* Motor Learning Consolidates Arc-Expressing Neuronal Ensembles in Secondary Motor Cortex. *Neuron* **86**, 1385-1392, doi:10.1016/j.neuron.2015.05.022 (2015).
- 11 Shteinikov, V., Evlanenkov, K., Bolshakov, K. & Tikhonov, D. Glutamate potentiates heterologously expressed homomeric acid-sensing ion channel 1a. *Synapse* **76**, e22227, doi:10.1002/syn.22227 (2022).

Reviewer Reports on the First Revision:

Referees' comments:

Referee #1 (Remarks to the Author):

The authors have addressed most of the previous questions reasonable well with additional new data. Remaining or new questions should be adequately addressed:

- (1) What were pH values for 0.2 -1.0 mM glutamic acid solutions used in the studies? For patch-clamp recording and cell injury assays, pH7.0 was mostly used. It is near the threshold of ASIC1a activation. Any slight change in pH would dramatically affect the current and the underlying cell injury.
- (2) In ischemic conditions, brain pH is commonly reduced to 6.5 or lower. In this study, only pH 7.0 was used for cell injury study which is not supposed to cause cell injury on its own without OGD. It is important to know whether the current finding can be extended to lower pH, e.g., 6.5 or 6.0, which is more relevant to ischemic conditions in vivo.
- (3) Both glutamic acid and sodium glutamate have been commonly used for various studies. In most cases they produce the same effect. The situation in the current study is completely different and surprising because glutamic acid and sodium glutamate are not doing the same thing. For this reason, it raises a critical question: which form is more relevant to the physiological condition? If sodium glutamate, which should become glutamate and sodium in the solution, is more relevant to endogenously released neurotransmitter glutamate, then the finding with glutamic acid is irrelevant to physiological and pathological conditions. This should be seriously addressed.
- (4) Zinc has profound effect on ASIC1a current as chelation of zinc has been shown to dramatically potentiates the ASIC1a current and a shift in dose-response curve (Chu et al., J Neurosci. 2004). Amino acid such as aspartic acid can chelate zinc. It might be important to know whether glutamic acid has a similar effect. The authors showed new data that glutamate still potentiated the current and caused a shift in ASIC dose-response curve. Since 10 μ M TPEN can dramatically potentiate the ASIC1a current, one may expect that glutamate would not cause the same degree of potentiation and the shift in dose-response curve in the presence of TPEN. Showing raw current traces and dose-response curve for control, glutamate alone and glutamate + TPEN would be very interesting and helpful.
- (5) Figure 2g: For ASIC+/+ neurons, example raw image for pH 7.0 shows no difference compared to that of pH 7.0 + GluR-B, which is not consistent with the data in Figure 2h that show changes of cell death from 50% to 70%. In fact, pH 7.0 image shows more than 90% cell death (>30 PI positive cells and 3 calcein positive cells) instead of ~50%.
- (6) Data in Figure 2h and 2i which show that GluR-B (50 μ M D-AP5, 10 μ M NBQX and 10 μ M LY341495) dramatically potentiates the pH7.0 cell injury do not make sense at all. Since D-AP5 is known to reduce glutamate potentiation of ASIC current (Extended Figure 2b), why addition of GluR-B increased the injury by Glu?

Referee #2 (Remarks to the Author):

in the revised manuscript by Lai et al., the authors demonstrate an exciting finding that glutamate is acting as a positive allosteric modulator of ASIC1a channels during acidification and ischemia. The work is impressive and comprehensive. Most impactful, is the identification of the glutamate binding site and the discovery of a new small molecule, LK-2 that can block glutamate/ASIC1a interaction, thus mitigating cell death during ischemic stroke. The work is highly novel and important for stroke research and treatment, but may also imply a broader role of ASIC1a in plasticity and other neuronal functions. At this time, I am satisfied with the author's responses to my original concerns. Great care has been taken to address each concern and add additional data to support the rebuttal. Excellent work overall.

Referee #4 (Remarks to the Author):

This revised paper is much clearer than the original version. The authors have largely addressed my previous comments in a satisfactory manner except for the question about the description of monosodium glutamate and glutamate.

According to the authors' results, monosodium glutamate and glutamate behave differently on ASIC1a due to different optical activities. However, under buffer conditions, monosodium glutamate should be able to ionize to form glutamate. The lack of effect of monosodium glutamate on ASIC1a still remains unreasonable in science. Could the effects of glutamate be modulated by high concentrations of sodium ions? This problem must be carefully solved. Suggested experiments include: 1. Use mixture of glutamate and sodium chloride to do the test. 2. Use potassium salt of glutamate to do the test. 3. Use ion exchange to convert monosodium glutamate to glutamate and test it.

BTW, what glutamate did the authors use? The authors should provide detailed structural formulas as the term glutamate is not a precise description.

Author Rebuttals to First Revision:

Rebuttal

We sincerely thank three referees for their insightful comments (*in blue*) which help strengthen this work. Our detailed responses are presented below.

Referee #1 (Remarks to the Author):

(1) The authors have addressed most of the previous questions reasonable well with additional new data. Remaining or new questions should be adequately addressed: (1) What were pH values for 0.2 -1.0 mM glutamic acid solutions used in the studies? For patch-clamp recording and cell injury assays, pH 7.0 was mostly used. It is near the threshold of ASIC1a activation. Any slight change in pH would dramatically affect the current and the underlying cell injury.

Response: We are aware of the critical importance of accurate pH values when the slope of dose-response curves is so steep. Throughout this study, we have carefully monitored the pH changes of solutions after addition of glutamate and any other compounds, and always readjusted to no more than 0.02 of designated pH values as described in METHOD. The pH values were 7.0 and 6.8 as shown in **Fig.1c** for dose-response curves.

(2) In ischemic conditions, brain pH is commonly reduced to 6.5 or lower. In this study, only pH 7.0 was used for cell injury study which is not supposed to cause cell injury on its own without OGD. It is important to know whether the current finding can be extended to lower pH, e.g., 6.5 or 6.0, which is more relevant to ischemic conditions in vivo.

Response: In compliance with the Referee's suggestion, we performed additional cell death experiments to test whether our current finding can be extended to lower pH, e.g., 6.5 and 6.0. As shown in **Fig.2g-i** and **Extended Data Fig. 4h-j**, we found that both cell death and LDH release at pH 6.5 were comparable to those at pH 7.0. Surprisingly, cell death was substantially alleviated at pH 6.0 compared to that at pH 7.0.

To find an answer for these unexpected results, we recorded and compared I_{ASICs} evoked by pH 7.0, 6.5 and 6.0 ECF perfusate onto cultured neurons, and found that I_{ASICs} was fully

desensitized in response to perfusate at all three pH values. However, co-application of glutamate generates a sustained steady-state inward tonic current at pH 7.0 and 6.5 but fails to do so at pH 6.0 (**Extended Data Fig. 4k or Rebuttal Figure 1a**). Although the magnitude of instantaneous I_{ASICs} grows as pH value decreases, the amplitude of tonic currents shows an inverse relationship to acidity, being the largest at pH 7.0 and smallest at pH 6.0 as evidently seen in **Rebuttal Figure 1b** (also see **Fig. 2d,e**).

We also measured glutamate release from neurons with or without OGD and found that OGD at pH 7.0 significantly elevates extracellular concentration of glutamate (**Fig. 2j**). Under these conditions, glutamate likely exerts positive allosteric effects on ASICs to facilitate a tonic leak current and Ca^{2+} inflow into the cell, ultimately exacerbating the cell death at mild pH environment (e.g. ~pH 7.0). In contrast, reduced cell death at pH 6.0 may be attributed to full desensitization of ASICs and minimal tonic current, despite elevated glutamate during OGD (**Extended Data Fig. 4k,i**). In our view, this set of results adds another layer of novelty to this work because it implicates the unappreciated risk of mild acidosis.

We believe that the positive allosteric effect of glutamate on ASIC1a at mild values (e.g. pH 7.0) is highly relevant to clinical ischemic conditions *in vivo*. It may very well be true that pH at the ischemic core of infarct is highly acidic (e.g. pH 6.0-6.5) to cause irreversible cell death. But brain damage typically expands from the ischemic core into the penumbra after stroke and/or during reperfusion, mild/moderate acidity in combination with elevated glutamate can aggravate brain damage and consequentially cause disability. This argument is supported by our measurements of cell death and LDH assays *in vitro* and comparisons of infarct volume *in vivo* (**Fig. 2g-n**) We have added this argument in the Discussion to underscore the importance of blocking glutamate binding to ASIC1a channels for constraining infarct volume and promoting recovery.

(3) Both glutamic acid and sodium glutamate have been commonly used for various studies. In most cases they produce the same effect. The situation in the current study is completely different and surprising because glutamic acid and sodium glutamate are not doing the same thing. For this reason, it raises a critical question: which form is more relevant to the physiological condition? If sodium glutamate, which should become glutamate and sodium in the solution, is more relevant to endogenously released neurotransmitter glutamate, then the finding with glutamic acid is irrelevant to physiological and pathological conditions. This should be seriously addressed.

Response:

There is no doubt that endogenous neurotransmitter glutamate is **L-isoform** (dextrorotatory) under physiological condition. But we agree with the referee that a lack of effect of the L-monosodium glutamate (L-MSG, also known as an additive for enhancing savory flavor of foods) on ASIC1a is truly puzzling, especially given that L-glutamic acid and L-MSG have been widely used to activate glutamate receptors in the literature.

We suspected that the L-monosodium glutamate (L-MSG, Sigma-Aldrich, batch#BCBK6359V, 2013) may have degraded or partially transformed into racemic mixture of D- and L-isoforms over the time span of >10 yrs at room temperature for storage. D-glutamate is known as a false neurotransmitter with much lower affinity for glutamate receptors. With these in mind, we just purchased two new batches of L-MSG (Sigma-

Aldrich 2022, batch#BCCH8156 and Macklin 2023, batch#C15097112). As seen from the figure below (**Rebuttal Figure 2**), we found that L-MSG (Sigma-Aldrich 2022 and Macklin 2023), but not L-MSG (Sigma-Aldrich 2013), significantly potentiated I_{ASIC1a} in the same cell at pH 7.0, suggesting that L-MSG (Sigma-Aldrich 2013) may have degraded.

As another test of quality control, we tested the effects of three batches of L-MSG on NMDAR currents and noted that L-MSG (Sigma-Aldrich 2013) generates much smaller currents compared with other two batches of L-MSG in the same cells. In addition, we observed that potassium glutamate could also enhance the peak amplitude and integral area of I_{ASIC1a} . Together, these results convinced us that a lack of potentiation of I_{ASIC1a} by L-MSG (Sigma-Aldrich 2013) presented in our previous version of the manuscript is flawed. Instead, it can be ascribed to a loss of agonist activity for both ASIC1a and NMDARs after nearly 10 years storage. Like L-glutamate, L-MSG and K-Glu are indeed capable of potentiating I_{ASIC1a} .

We sincerely apologize to the referee for having not validated the activity of L-MSG with functional measurements, leading to this major confusion in previous rounds of reviews.

Rebuttal Figure 2. The effect of different batches of L-MSG and monopotassium glutamate (K-Glu) on ASIC1a and NMDAR. **a**, Different batches of L-MSG (Sigma-Aldrich 2013, Sigma-Aldrich 2022, Macklin 2023). **b**, The effect of different batches of L-MSG (500 μ M) on I_{ASIC1a} of the same cell at pH 7.0. $n=8$ cells. **c**, The effect of different batches of L-MSG (200 μ M) on NMDAR (GluN1+GluN2A) currents of the same cell in the presence of 1 μ M glycine. $n=10$ cells. **d**, K-Glu potentiate the peak amplitude and integral area of I_{ASIC1a} at pH 7.0. $n=11$ cells. Three replicated cultures for patch-clamp recordings were tested. Data are mean \pm s.e.m.; one-way ANOVA with Tukey post hoc correction (**b,c**); two-tailed paired t-test (**d**). P values are indicated.

(4) Zinc has profound effect on ASIC1a current as chelation of zinc has been shown to dramatically potentiates the ASIC1a current and a shift in dose-response curve (Chu at al., J Neurosci. 2004). Amino acid such as aspartic acid can chelate zinc. It might be important to know whether glutamic acid has a similar effect. The authors showed new data that glutamate still potentiated the current and caused a shift in ASIC dose-response curve. Since 10 μ M TPEN can dramatically potentiate the ASIC1a current, one may expect that glutamate would not cause the same degree of potentiation and the shift in dose-response curve in the presence of TPEN. Showing raw current traces and dose-response curve for control, glutamate alone and glutamate + TPEN would be very interesting and helpful.

Response: Under our experimental conditions, we found that the ASIC1a current did not change after 2 min exposure of 10 or 30 μ M TPEN at pH 6.5 or 7.0 (**Extended Data Fig. 1j-l**). As a positive control, we added 50 nM zinc in all solution, and found that the ASIC1a current was significantly reduced (amplitude: \sim 30 pA), but addition of 10 μ M TPEN at pH 7.0 showed a dramatical potentiation of I_{ASIC1a} (**Extended Data Fig. 1m**). These results indicated that our ECF contained very little, if any, zinc, excluding the possibility that potentiation of ASIC1a current by glutamate was via chelating background zinc in ECF.

(5) Figure 2g: For ASIC+/+ neurons, example raw image for pH 7.0 shows no difference compared to that of pH 7.0 + GluR-B, which is not consistent with the data in Figure 2h that show changes of cell death from 50% to 70%. In fact, pH 7.0 image shows more than 90% cell death (>30 PI positive cells and 3 calcein positive cells) instead of \sim 50%.

Response: We have replaced this with a more representative raw image in **Fig. 2g**.

(6) Data in Figure 2h and 2i which show that GluR-B (50 μ M D-AP5, 10 μ M NBQX and 10 μ M LY341495) dramatically potentiates the pH7.0 cell injury do not make sense at all. Since D-AP5 is known to reduce glutamate potentiation of ASIC current (Extended Figure 2b), why addition of GluR-B increased the injury by Glu?

Response: D-AP5 has no effect on the glutamate potentiation of ASIC current ($P=0.3043$, not labelled in the panel in **Extended Data Fig. 2b**) and was used to specifically block glutamate binding site on NMDARs, but not that on ASIC1a.

The observation that glutamate receptor blockers, namely D-AP5, NBQX and LY341495 (GluR-B), exacerbate cell injury at pH 7.0 is indeed surprising. Our results showed that ASIC1a currents were not affected by individual blockers at pH 7.0 (**Extended Data Fig. 4m**). Considering the observation that there is a sustained ASIC1a tonic current when glutamate is co-applied, we speculated that glutamate release could be increased due to elevated excitability driven by such a tonic current under OGD condition. To this end, we measured the glutamate concentration of cultured neurons after 1 hour OGD treatment and found that glutamate concentration was remarkably elevated by \sim 5 folds in the presence of GluR-B (**Fig. 2j**). In addition, prolonged blockade of glutamate receptors is known to trigger homeostatic upregulation of the release of endogenous glutamate. We suggest that these synergistically potentiate ASIC1a mediated tonic current and the cell injury at pH 7.0. Contrary to the canonical view that GluR-B is neuroprotective, our findings indicate that OGD induced cell death at mild acidosis is largely mediated by ASICs. This is supported by the observation that PcTX-1 can preclude cell-death as effectively as ASIC1a knockout.

Referee #2 (Remarks to the Author):

in the revised manuscript by Lai et al., the authors demonstrate an exciting finding that glutamate is acting as a positive allosteric modulator of ASIC1a channels during acidification and ischemia. The work is impressive and comprehensive. Most impactful, is the identification of the glutamate binding site and the discovery of a new small molecule, LK-2 that can block glutamate/ASIC1a interaction, thus mitigating cell death during ischemic stroke. The work is highly novel and important for stroke research and treatment, but may also imply a broader role of ASIC1a in plasticity and other neuronal functions. At this time, I am satisfied with the author's responses to my original concerns. Great care has been taken to address each concern and add additional data to support the rebuttal. Excellent work overall.

Response: We appreciate the reviewer for these positive comments on our manuscript.

Referee #4 (Remarks to the Author):

This revised paper is much clearer than the original version. The authors have largely addressed my previous comments in a satisfactory manner except for the question about the description of monosodium glutamate and glutamate.

According to the authors' results, monosodium glutamate and glutamate behave differently on ASIC1a due to different optical activities. However, under buffer conditions, monosodium glutamate should be able to ionize to form glutamate. The lack of effect of monosodium glutamate on ASIC1a still remains unreasonable in science. Could the effects of glutamate be modulated by high concentrations of sodium ions? This problem must be carefully solved. Suggested experiments include: 1. Use mixture of glutamate and sodium chloride to do the test. 2. Use potassium salt of glutamate to do the test. 3. Use ion exchange to convert monosodium glutamate to glutamate and test it.

Response: We thank the reviewer for suggesting the experiments, and we agree with the referee that a lack of effect of the L-monosodium glutamate (L-MSG, also known as an additive for enhancing savory flavor of foods) on ASIC1a is truly puzzling, especially given that L-glutamic acid and L-MSG have been widely used to activate glutamate receptors in the literature.

We suspected that the L-monosodium glutamate (L-MSG, Sigma-Aldrich, batch#BCBK6359V, 2013) may have degraded or partially transformed into racemic mixture of D- and L-isoforms over the time span of >10 yrs at room temperature for storage. D-glutamate is known as a false neurotransmitter with much lower affinity for glutamate receptors. With these in mind, we just purchased two new batches of L-

MSG (Sigma-Aldrich 2022, batch#BCCH8156 and Macklin 2023, batch#C15097112). As seen from the figure below (**Rebuttal Figure 2**), we found that L-MSG (Sigma-Aldrich 2022 and Macklin 2023), but not L-MSG (Sigma-Aldrich 2013), significantly potentiated I_{ASIC1a} in

the same cell at pH 7.0, suggesting that L-MSG (Sigma-Aldrich 2013) may have degraded.

As another test of quality control, we tested the effects of three batches of L-MSG on NMDAR currents and noted that L-MSG (Sigma-Aldrich 2013) generates much smaller currents compared with other two batches of L-MSG in the same cells. In addition, we observed that potassium glutamate could also enhance the peak amplitude and integral area of I_{ASIC1a} . Together, these results convinced us that a lack of potentiation of I_{ASIC1a} by L-MSG (Sigma-Aldrich 2013) presented in our previous version of the manuscript is flawed. Instead, it can be ascribed to a loss of activity for both ASIC1a and NMDARs after nearly 10 years storage. Like L-glutamate, L-MSG and K-Glu are indeed capable of potentiating I_{ASIC1a} .

We sincerely apologize to the referee for having not validated the activity of L-MSG with functional measurements, leading to this major confusion in previous rounds of reviews.

BTW, what glutamate did the authors use? The authors should provide detailed structural formulas as the term glutamate is not a precise description.

Response: L-glutamic acid (Sigma-Aldrich, G1251) was used in the present study. The detailed structural formula was presented in **Fig. 3d**. The term “glutamate” has been defined as “L-glutamic acid” when it is denoted on the first occasion in the text.

Reviewer Reports on the Second Revision:

Referees' comments:

Referee #1 (Remarks to the Author):

The authors have now shown that both monosodium glutamate and L-glutamic acid produce the same effect on ASIC1a in contrast to their earlier claim that only L-glutamic acid but not monosodium glutamate can potentiate the ASIC1a response. This information is very important. It reconciles a major discrepancy and avoids publication of otherwise flawed data.

Abstract: "We conclude that glutamate serves as the first messenger for ASICs". The term "first messenger" may not be accurate because glutamate does not directly activate but only potentiates the ASIC1a response and proton serves as the first messenger. Consider revising.

Referee #4 (Remarks to the Author):

The authors satisfactorily addressed the question I made about MSG and glutamic acid in my previous review. Overall, the current iteration is suitable for publication.

Author Rebuttals to Second Revision:

Referees' comments: (*in italic*)

Referee #1 (Remarks to the Author):

The authors have now shown that both monosodium glutamate and L-glutamic acid produce the same effect on ASIC1a in contrast to their earlier claim that only L-glutamic acid but not monosodium glutamate can potentiate the ASIC1a response. This information is very important. It reconciles a major discrepancy and avoids publication of otherwise flawed data.

Response: We appreciate the comment by Referee #1 and are pleased with a satisfactory resolution to the issue.

Abstract: "We conclude that glutamate serves as the first messenger for ASICs". The term "first messenger" may not be accurate because glutamate does not directly activate but only potentiates the ASIC1a response and proton serves as the first messenger. Consider revising.

Response: We replaced "first messenger" with "positive allosteric modulator" which describes the role of glutamate in modulating ASICs precisely.

Referee #4 (Remarks to the Author):

The authors satisfactorily addressed the question I made about MSG and glutamic acid in my previous review. Overall, the current iteration is suitable for publication.

Response: We appreciate the comment and endorsement by Referee #4